# Improved Bayes Regret Bounds for Multi-Task Hierarchical Bayesian Bandit Algorithms

**Jiechao Guan[1]**          **Hui Xiong[1,2,∗]**

[1]AI Thrust , The Hong Kong University of Science and Technology (Guangzhou), China
[2]Department of Computer Science and Engineering, HKUST, China
`{jiechaoguan, xionghui}@hkust-gz.edu.cn`

## Abstract

Hierarchical Bayesian bandit refers to the multi-task bandit problem in which bandit tasks are assumed to be drawn from the same distribution. In this work, we provide improved Bayes regret bounds for hierarchical Bayesian bandit algorithms in the multi-task linear bandit and semi-bandit settings. For the multi-task linear bandit, we first analyze the preexisting hierarchical Thompson sampling (HierTS) algorithm, and improve its gap-independent Bayes regret bound from $O(m\sqrt{n\log n}\log{(mn)})$ to $O(m\sqrt{n\log n})$ in the case of infinite action set, with $m$ being the number of tasks and $n$ the number of iterations per task. In the case of finite action set, we propose a novel hierarchical Bayesian bandit algorithm, named hierarchical BayesUCB (HierBayesUCB), that achieves the logarithmic but gap-dependent regret bound $O(m\log{(mn)}\log n)$ under mild assumptions. All of the above regret bounds hold in many variants of hierarchical Bayesian linear bandit problem, including when the tasks are solved sequentially or concurrently. Furthermore, we extend the aforementioned HierTS and HierBayesUCB algorithms to the multi-task combinatorial semi-bandit setting. Concretely, our combinatorial HierTS algorithm attains comparable Bayes regret bound $O(m\sqrt{n}\log n)$ with respect to the latest one. Moreover, our combinatorial HierBayesUCB yields a sharper Bayes regret bound $O(m\log{(mn)}\log n)$. Experiments are conducted to validate the soundness of our theoretical results for multi-task bandit algorithms.

## 1 Introduction

A stochastic bandit [26, 6, 27] is a sequential decision-making problem where at each round, an agent has to choose an action, and receives a stochastic reward without knowing its expected value. The gap between the cumulative reward of optimal actions in hindsight and the cumulative reward of agent is defined as *regret*. The goal is to minimize regret, through a combination of exploring different actions and exploiting those with high rewards in the past. Typical applications of bandit algorithms include news article recommendation [28], computational advertisement [20], and dynamic pricing [24]. For example, in news article recommendation, the agent must choose a news article for a user. The actions in this bandit setting are articles and the reward could be an indicator of a click from user.

When the agent has to solve multiple bandit tasks, many machine learning researchers resort to multi-task learning/meta-learning paradigm [8, 34] to benefit task adaptation. The existing works focused on the multi-task bandit problem can be categorized into three main groups: **(1)** The first group attempts to learn a low-dimensional representation shared by different bandit tasks, to derive a sharper cumulative regret bound than that derived by learning each task independently [19, 10]. **(2)** The second group leverages the similarity of contexts (e.g. the feature of actions) in bandit tasks to improve agent's ability to predict rewards in a new task [14, 36]. **(3)** The third group chooses to

---

∗Corresponding Author

38th Conference on Neural Information Processing Systems (NeurIPS 2024).

maintain a meta-distribution over the hyper-parameters of within-task bandit algorithms (like Tsallis-INF [23], OFUL [9], and Thompson sampling [25, 7, 17]), and draws informative hyper-parameters from the meta-distribution for efficient regret minimization. Our work falls into the third group and formulates the problem of learning similar bandit tasks in a hierarchical Bayesian bandit model [17].

Specifically, in hierarchical Bayesian bandit setting, each bandit task is characterized by a task parameter. Different bandit task parameters are assumed to be independently and identically distributed according to the same distribution. At each round, the learning agent interacts with one or several bandit tasks, which correspond to the sequential and concurrent bandit settings respectively. Many existing works considered hierarchical Bayesian bandit problem, and proposed Thompson sampling [33] type algorithms to solve it [25, 7, 36]. The latest work [17] proposed hierarchical Thompson sampling (HierTS) algorithm and developed a gap-independent Bayes regret bound $O(m\sqrt{n \log n} \log (mn))$ in the Gaussian linear bandit setting, where $m$ is the number of bandit tasks and $n$ the number of iterations per task. However, it is still unclear for us whether we can derive sharper regret bounds or how to extend hierarchical Bayesian bandit algorithms to the more general multi-task bandit setting.

In this work, we attempt to tackle the above two issues, by providing improved Bayes regret bounds for hierarchical Bayesian bandit algorithms in the multi-task Gaussian linear bandit and semi-bandit setting. Firstly, in the linear bandit setting, we improve the multi-task Bayes regret bound of HierTS to $O(m\sqrt{n \log n})$ in the case of infinite action set, strengthening the latest bound in [17, Thm 3] by a factor of $O(\sqrt{\log (mn)})$. In the case of finite action set, we propose a novel hierarchical Bayesian bandit algorithm, named hierarchical BayesUCB (HierBayesUCB), that achieves the logarithmic but gap-dependent regret bound $O(m \log (mn) \log n)$ under mild assumptions. All of the above regret bounds for linear bandit hold in both the sequential and concurrent setting. Secondly, we extend the aforementioned HierTS and HierBayesUCB algorithms to the multi-task Gaussian combinatorial semi-bandit setting. Concretely, our combinatorial HierTS algorithm attains comparable Bayes regret bound $O(m\sqrt{n} \log n)$ with respect to the latest one in [7, Thm 6]. Moreover, our combinatorial HierBayesUCB yields a sharper but gap-dependent regret bound $O(m \log (mn) \log n)$. Extensive experiments in the Gaussian linear bandit setting are conducted to support our theoretical results.

Overall, our theoretical contributions are four-fold: **(1)** In the case of infinite action set, we provide a tighter Bayes regret bound $O(m\sqrt{n \log n})$ for HierTS. This bound improves the latest result by a factor of $O(\sqrt{\log (mn)})$. **(2)** In the case of finite action set, we propose a novel HierBayesUCB algorithm, and provide gap-dependent logarithmic Bayes regret bound $O(m \log (mn) \log n)$ for it. **(3)** We generalize the above regret bounds for linear bandit from sequential setting to the more challenging concurrent setting. **(4)** We extend both HierTS and HierBayesUCB algorithms to the more general multi-task combinatorial semi-bandit setting and derive improved Bayes regret bounds.

## 2   Related Work

**Frequentist Regret Bounds for Stochastic Linear Bandit**. In the frequentist stochastic bandit setting, we do not assume the bandit task parameter is sampled from a fixed distribution. The frequentist regret is thus for any fixed task parameter, without taking expectation over the distribution of task parameter. **(1)** In the case of finite action set: [5] for the first time investigated the stochastic linear bandit problem and proposed an algorithm with a frequentist regret of $O(\sqrt{dn} \log^{3/2} n)$, where $d$ is the dimension of action space and $n$ is the number of rounds. [29] developed a new algorithm and improved the regret bound to $O(\sqrt{dn \log n})$. [12] showed that the lower frequentist regret bound in the finite action set is $\Omega(\sqrt{dn})$. **(2)** In the case of infinite action set: Both [13] and [30] proposed algorithms that achieve $O(d\sqrt{n} \log^{3/2} n)$ regret. The regret bound was further improved in [1, 15] to $O(d\sqrt{n} \log n)$, by designing novel linear bandit algorithms or utilizing advanced martingale methods.

**Bayes Regret Bounds for Bayesian Linear Bandit**. In the Bayesian stochastic bandit setting, the Bayes regret is the expected cumulative regret whose expectation is taken over the draw of task parameter from a distribution. It is not difficult to see that the frequentist regret upper bound implies a Bayes regret upper bound, because the former holds for any task parameter. **(1)** When the action set is infinite: [30] showed that in the Gaussian linear bandit setting, the Bayes regret of any Bayesian bandit algorithm is lower bounded by $\Omega(\sqrt{n})$. [31] for the first time gave the Bayes regret bound of $O(d\sqrt{n} \log n)$ for both Thompson sampling (TS) and BayesUCB [22] algorithms. Recently, [21] provided an improved Bayes regret $O(d\sqrt{n \log n})$ for TS algorithm with a concise proof. **(2)** When the action set is finite: [32] derived a tight regret bound of $O(\sqrt{dn})$ for TS algorithm with

sub-Gaussian reward noise, via a novel information-theoretic approach. Recently, [3] developed a logarithmic Bayes regret bound $O(d^2 \log^2 n)$ for BayesUCB algorithm in the Gaussian bandit setting.

**Frequentist Regret Bounds for Multi-Task Linear Bandit Problems**. Under the representation learning paradigm, the frequentist regret bounds in [19, 10] for multi-task linear bandits scales as $O(m\sqrt{nk})$, where $k$ is the dimension of low-dimensional representation. The expected frequentist regret upper bound for multi-task adversarial linear bandit in [23] is $O(m\sqrt{n \log{(1 + nV)}})$, with $V$ being the similarity among multiple adversarial bandit tasks. Nevertheless, we should mention that all of these frequentist regret bounds for multi-task linear bandit problem are not tighter than $\Omega(m\sqrt{n})$.

**Bayes Regret Bounds for Multi-Task Bayesian Linear Bandit/Semi-Bandit**. The most related works to ours are [25, 7, 17], which provided hierarchical-type Thompson sampling algorithms for multi-task bandit and derived Bayes regret bounds in the Gaussian reward setting. We list these Bayes regret bounds in Table 1 for direct comparisons. Among them, the latest work [17] proposed the HierTS algorithm and obtained its regret bound of $O(m\sqrt{n \log n \log{(mn)}})$, [7] derived the first Bayes regret bound for multi-task hierarchical Bayesian semi-bandit algorithm. In this work, we provide for HierTS improved Bayes regret bound of $O(m\sqrt{n \log n})$ in Theorem 5.1. We also propose a novel HierBayesUCB algorithm that achieves logarithmic regret bound $O(m \log n \log{(nm)})$. We finally extend HierTS and HierBayesUCB to the semi-bandit setting and derive improved theoretical results. Other works utilized action features or structure information to derive Bayes regret bounds for multi-task bandit [36, 37], e.g. the Bayes regret bound in [36] is $O(m\sqrt{n \log n} + m \log^2{(mn)})$.

**Hierarchical Bayesian Bandit Algorithms**. Hierarchical Bayesian bandit algorithm was first proposed by [25] to solve multi-task bandit problems. More hierarchical-type Thompson sampling algorithms based on multi-task/meta learning frameworks were developed with improved theoretical guarantees and empirical performance [7, 17, 36, 37]. There also existed other works investigating hierarchical Bayesian bandit algorithms within the single-task bandit setting. For example, [16] extended the two-level hierarchical Bayesian bandit framework to the deeper multiple-level hierarchial Bayesian bandit framework. [2] generalized the single-effect-parameter HierTS algorithm (i.e. the action parameter is centered at a single latent variable) to the mixed-effect bandit framework where each action is associated with a parameter that depends upon one or multiple effect parameters.

## 3 Problem Setting

For any positive integer $n$, denote $[n] = \{1, 2, ..., n\}$ for brevity. For any square matrix $M \in \mathbb{R}^{d \times d}$, denote $\lambda_1(M)$, $\lambda_d(M)$ as its maximum and minimum eigenvalues respectively, denote $\kappa(M) = \lambda_1(M)/\lambda_d(M)$ as its condition number. The action set $\mathcal{A} \subseteq \overline{\mathcal{B}(0, B)} \subseteq \mathbb{R}^d$ is assumed to be compact for some positive constant $B > 0$, where $\overline{\mathcal{B}(0, B)}$ is the closed ball centered at the origin. We use $\langle, \rangle$ to denote the inner-product between vectors, use $\mathbf{w}(a)$ or $\mathbf{w}_a$ to denote the $a$-th element of vector $\mathbf{w}$.

**Single-Task Bandit**. A stochastic bandit problem is characterized by an unknown parameter $\theta$ with an action set $\mathcal{A}$. Each action $a \in \mathcal{A}$ under the bandit instance $\theta$ is associated with a reward distribution $\mathbb{P}(\cdot|a, \theta)$. The reward mean of action $a$ under $\theta$ is denoted as $r(a; \theta) = \mathbb{E}_{Y \sim \mathbb{P}(\cdot|a;\theta)}[Y]$, and the optimal action under $\theta$ is denoted as $A_* = \arg\max_{a \in \mathcal{A}} r(a; \theta)$. In the stochastic linear bandit setting, the mean reward of action $a \in \mathcal{A}$ is $r(a, \theta) = a^\top \theta$. In Bayesian bandit problem, we further assume that the task parameter $\theta$ is independently and identically distributed (i.i.d.) according to a task parameter distribution $\mathbb{P}(\cdot|\mu_*)$, which is characterized by an unknown hyper-parameter $\mu_*$.

**Single-Task Semi-Bandit**. In the semi-bandit setting, the action set $\mathcal{A} = [K]$ is a set of finite items. $\mathscr{A} = \{A \subseteq \mathcal{A} : |A| \leq L\}$ is a family of subsets of $\mathcal{A}$ with up to $L$ items, where $L \leq K$. $\mathbf{w} \in \mathbb{R}^K$ is a weight vector. The weight of a set $A \in \mathscr{A}$ is defined as $\sum_{a \in A} \mathbf{w}(a)$. We assume that the weights $\mathbf{w}$ are drawn i.i.d. from a distribution, and the mean weight is denoted as $\bar{\mathbf{w}} = \mathbb{E}[\mathbf{w}]$. Following previous work [38], we focus on the coherent case [39] which assumes that the agent knows a feature matrix $\Phi \in \mathbb{R}^{K \times d}$, such that $\bar{\mathbf{w}} = \Phi\theta$, where $\theta$ is the task parameter drawn from $\mathbb{P}(\cdot|\mu_*)$. The reward of a subset $A \in \mathscr{A}$ under the bandit instance $\theta$ is defined as $r(A; \theta) = \sum_{a \in A}(\Phi\theta)(a) = \sum_{a \in A}\langle \Phi_a, \theta \rangle$, where $\Phi_a$ is the transpose of the $a$-th row of matrix $\Phi$. We further assume that $\|\Phi_a\| \leq B, \forall a \in \mathcal{A}$.

**Hierarchical Bayesian Multi-Task Bandit/Semi-Bandit**. In this setting, the agent interacts with $m$ tasks sequentially or concurrently. First, sample the hyper-parameter $\mu_*$ from a hyper-prior $Q$. Then, for each task $s \in [m]$, sample the task parameter $\theta_{s,*}$ independently from distribution $\mathbb{P}(\cdot|\mu_*)$. The learning process can be detailed as follows. At round $t \geq 1$, the agent interacts with a set

of tasks $\mathcal{S}_t \subseteq [m]$, takes a series of actions $A_t = (A_{s,t})_{s \in \mathcal{S}_t}$, and receives a series of rewards $Y_t = (Y_{s,t})_{s \in \mathcal{S}_t}$. In the bandit setting, $Y_{s,t} \sim \mathbb{P}(\cdot|A_{s,t}; \theta_{s,*})$ is a stochastic reward obtained by taking action $A_{s,t}$ in task $s \in \mathcal{S}_t$; in the semi-bandit setting, $Y_{s,t} = \{\hat{\mathbf{w}}_{s,t}(a)\}_{a \in A_{s,t}}$ is a series of stochastic rewards, where $\hat{\mathbf{w}}_{s,t} = \bar{\mathbf{w}}_s + \eta_{s,t}$, $\bar{\mathbf{w}}_s = \Phi \theta_{s,*}$, and $\eta_{s,t}$ is a $K$-dimensional random noise. The full hierarchical Bayesian bandit/semi-bandit model in the $m$-task learning setting is exhibited as follow:

**(1)** $\mu_* \sim Q$; **(2)** $\theta_{s,*}|\mu_* \sim \mathbb{P}(\cdot|\mu_*), \forall s \in [m]$; **(3)** $Y_{s,t}|A_{s,t}, \theta_{s,*} \sim \mathbb{P}(\cdot|A_{s,t}; \theta_{s,*}), \forall t \geq 1, s \in \mathcal{S}_t$.

Therefore, the goal of the agent in hierarchical Bayesian multi-task bandit/semi-bandit setting is to interact with $m$ tasks efficiently and minimize the following cumulative *multi-task Bayes regret*:

$$\mathcal{BR}(m, n) = \mathbb{E}\Big[\sum_{t \geq 1}\sum_{s \in \mathcal{S}_t} r(A_{s,*}; \theta_{s,*}) - r(A_{s,t}; \theta_{s,*})\Big], \tag{1}$$

where $A_{s,*} = \arg\max_{a \in \mathcal{A}} r(a; \theta_{s,*})$ is the optimal action for task $s \in [m]$ in the bandit setting, and $A_{s,*} \in \arg\max_{A \in \mathscr{A}} r(A; \theta_{s,*})$ is the optimal subset for task $s \in [m]$ in the semi-bandit setting. The expectation is taken over $\mu_*$, all task parameters $(\theta_{s,*})_{s \in [m]}$, all actions $(A_t)_{t \geq 1}$, all stochastic rewards $(Y_t)_{t \geq 1}$. We further assume that the action set $\mathcal{A}$ is the same across different tasks for ease of exposition, and assume that the learning agent interacts with any task $s \in [m]$ for at most $n$ rounds for convenient comparison with exiting regret upper bounds for multi-task bandit/semi-bandit problem.

## 4 Algorithm

Denote $H_{s,t} = ((A_{s,\ell}, Y_{s,\ell}))_{\ell < t, s \in \mathcal{S}_t}$ as the history of all interactions of agent with task $s \in [m]$, and $H_t = (H_{s,t})_{s \in [m]}$ as the whole interaction history up to round $t$. We next introduce the specific form of Hierarchical Thompson Sampling (HierTS) and Hierarchical BayesUCB (HierBayesUCB) algorithms in the multi-task Bayesian linear bandit and semi-bandit settings, and instantiate these two algorithms to the multi-task Gaussian linear bandit (Algorithm 1) and semi-bandit (Algorithm 2) problems.

### 4.1 Hierarchical Thompson Sampling and Hierarchical BayesUCB

At round $t$, hierarchical Bayesian bandit algorithm samples a hyper-parameter $\mu_t$ from the hyper-posterior $Q_t$ defined as $Q_t(\mu) = \mathbb{P}(\mu_* = \mu|H_t)$, and then interacts with tasks $\mathcal{S}_t \subset [m]$. Next, we give details of bandit algorithms, and details of semi-bandit algorithms are deferred to Section 5.4.

**Hierarchical Thompson Sampling**. For any task $s \in \mathcal{S}_t$, HierTS samples task parameter $\theta_{s,t}$ from the distribution $\mathbb{P}_{s,t}(\theta|\mu_t) \triangleq \mathbb{P}(\theta_{s,*} = \theta|\mu_* = \mu_t, H_{s,t})$ and takes the action $A_{s,t} = \arg\max_{a \in \mathcal{A}} a^\top \theta_{s,t}$, where $\mathbb{P}_{s,t}(\theta|\mu_t)$ is only conditioned on $H_{s,t}$ due to the independence between task parameter $\theta_{s,*}$ and other task histories. This process clearly samples bandit instance $\theta_{s,t}$ from the true posterior $\mathbb{P}(\theta_{s,*} = \theta|H_t)$, which is equivalent to the form: $\int \mathbb{P}(\theta_{s,*} = \theta, \mu_* = \mu|H_t)\mathrm{d}\mu = \int \mathbb{P}_{s,t}(\theta|\mu)Q_t(\mu)\mathrm{d}\mu$, where $\mathbb{P}_{s,t}(\theta|\mu) \propto \mathcal{L}_{s,t}(\theta)\mathbb{P}(\theta|\mu)$ is the posterior probability, $\mathcal{L}_{s,t}(\theta) = \prod_{(a,y) \in H_{s,t}} \mathbb{P}(y|a; \theta)$ is the likelihood function, $\mathbb{P}(\theta|\mu)$ is the prior probability by Bayes rule.

**Hierarchical BayesUCB**. For any task $s \in \mathcal{S}_t$ in round $t$, HierBayesUCB computes the upper confidence bound $U_{t,s,a} = a^\top \hat{\mu}_{s,t} + \sqrt{2\log\frac{1}{\delta}}\|a\|_{\hat{\Sigma}_{s,t}}$ for any $a \in \mathcal{A}$, where $\hat{\mu}_{s,t}$ and $\hat{\Sigma}_{s,t}$ are the expectation and covariance of the distribution (i.e. $\mathbb{P}(\theta_{s,*} = \theta|H_t)$) of $\theta_{s,*}$ conditioned on the history $H_t$, and then takes action with the highest upper confidence bound : $A_{s,t} \leftarrow \arg\max_{a \in \mathcal{A}} U_{t,s,a}$.

### 4.2 Multi-Task Gaussian Linear Bandit and Semi-Bandit

The hierarchical Gaussian environment is generated as follow. In the multi-task linear bandit setting: **(1)** $\mu_* \sim \mathcal{N}(\mu_q, \Sigma_q)$, **(2)** $\theta_{s,*}|\mu_* \sim \mathcal{N}(\mu_*, \Sigma_0), \forall s \in [m]$, **(3)** $Y_{s,t}|A_{s,t}, \theta_{s,*} \sim \mathcal{N}(A_{s,t}^\top \theta_{s,*}, \sigma^2), \forall t \geq 1, s \in \mathcal{S}_t$; In the semi-bandit setting, the only difference lies in step **(3)** where $Y_{s,t,a}|A_{s,t}, \theta_{s,*} \sim \mathcal{N}(\langle \Phi_a, \theta_{s,*} \rangle, \sigma^2)$ for any $a \in A_{s,t}$. Here, $\mu_q, \mu_*, \theta_{s,*}$ are $d$-dimensional vectors; $\Sigma_q, \Sigma_0 \in \mathbb{R}^{d \times d}$ are positive semi-definite covariance matrices. In the above two settings, the reward noise can be regarded as $\mathcal{N}(0, \sigma^2)$. In the following theoretical analysis sections, we assume that all of $\mu_q, \Sigma_q, \Sigma_0$ and $\sigma$ are known by the agent to guarantee an analytically tractable posterior.

Concretely, using some basic algebraic computations in hierarchical Gaussian model (e.g. see [25, Appendix D]), we can obtain the closed-form hyper-posterior in round $t$ as $Q_t(\mu) = \mathcal{N}(\mu; \bar{\mu}_t, \bar{\Sigma}_t)$, where the expectation $\bar{\mu}_t$ and the covariance matrix $\bar{\Sigma}_t$ of $Q_t(\mu)$ have the following explicit forms:

$$\bar{\mu}_t = \bar{\Sigma}_t\Big(\Sigma_q^{-1}\mu_q + \sum_{s \in [m]}(\Sigma_0 + G_{s,t}^{-1})^{-1}G_{s,t}^{-1}B_{s,t}\Big), \qquad \bar{\Sigma}_t^{-1} = \Sigma_q^{-1} + \sum_{s \in [m]}(\Sigma_0 + G_{s,t}^{-1})^{-1}. \tag{2}$$

| **Algorithm 1** Hierarchical Bayesian Algorithms for Multi-Task Linear Bandit Setting | **Algorithm 2** Hierarchical Bayesian Algorithms for Multi-Task Combinatorial Semi-Bandit Setting |
|---|---|
| 1: **Input:** Hyper-prior $Q$ | 1: **Input:** Hyper-prior $Q$, features $\Phi \in \mathbb{R}^{K \times d}$ |
| 2: Initialize $Q_1 \leftarrow Q$ | 2: Initialize $Q_1 \leftarrow Q$ |
| 3: **for** $t = 1, 2, \dots$ **do** | 3: **for** $t = 1, 2, \dots$ **do** |
| 4:    Sample hyper-parameter $\mu_t \sim Q_t$ | 4:    Sample hyper-parameter $\mu_t \sim Q_t$ |
| 5:    Observe tasks $\mathcal{S}_t \subseteq [m]$ | 5:    Observe tasks $\mathcal{S}_t \subseteq [m]$ |
| 6:    **for** $s \in \mathcal{S}_t$ **do** | 6:    **for** $s \in \mathcal{S}_t$ **do** |
| 7:       **Option I (HierTS):** | 7:       **Option I (HierTS):** |
|         Compute $\mathbb{P}_{s,t}(\theta \mid \mu_t) \propto \mathcal{L}_{s,t}(\theta)\mathbb{P}(\theta \mid \mu_t)$ |         Compute $\mathbb{P}_{s,t}(\theta \mid \mu_t) \propto \mathcal{L}_{s,t}(\theta)\mathbb{P}(\theta \mid \mu_t)$ |
|         Sample task parameter $\theta_{s,t} \sim \mathbb{P}_{s,t}(\cdot \mid \mu_t)$ |         Sample task parameter $\theta_{s,t} \sim \mathbb{P}_{s,t}(\cdot \mid \mu_t)$ |
|         Take action $A_{s,t} \leftarrow \arg\max_{a \in \mathcal{A}} a^\top \theta_{s,t}$ |         Compute $A_{s,t} = \text{ORACLE}(\mathcal{A}, \mathscr{A}, \Phi\theta_{s,t})$ |
|         **Option II (HierBayesUCB):** |         **Option II (HierBayesUCB):** |
|         Set $U_{t,s,a} = a^\top \hat{\mu}_{s,t} + \sqrt{2\log\frac{1}{\delta}}\|a\|_{\hat{\Sigma}_{s,t}}$, for any $a \in \mathcal{A}$ |         Compute $U_{t,s}(A) = \sum_{a \in A}(a^\top\hat{\mu}_{s,t} + \sqrt{2\log\frac{1}{\delta}}\|a\|_{\hat{\Sigma}_{s,t}})$, for all $A \in \mathscr{A}$ |
|         Take action $A_{s,t} \leftarrow \arg\max_{a \in \mathcal{A}} U_{t,s,a}$ |         Compute $A_{s,t} = \arg\max_{A \in \mathscr{A}} U_{t,s}(A)$ |
| 8:       Observe reward $Y_{s,t}$ | 8:       Chooose $A_{s,t}$ and observe $\{\hat{\mathbf{w}}_{s,t}(a)\}_{a \in A_{s,t}}$ |
| 9:    **end for** | 9:    **end for** |
| 10:   Update $Q_{t+1}$ | 10:   Update $Q_{t+1}$ |
| 11: **end for** | 11: **end for** |

Here, in the bandit setting $G_{s,t} = \sigma^{-2}\sum_{\ell<t}\mathbf{1}\{s \in \mathcal{S}_\ell\}A_{s,\ell}A_{s,\ell}^\top$ and $B_{s,t} = \sigma^{-2}\sum_{\ell<t}\mathbf{1}\{s \in \mathcal{S}_\ell\}A_{s,\ell}Y_{s,\ell}$; in the semi-bandit setting $G_{s,t} = \sigma^{-2}\sum_{\ell<t}\mathbf{1}\{s \in \mathcal{S}_\ell\}(\sum_{a \in A_{s,\ell}}\Phi_a\Phi_a^\top)$ and $B_{s,t} = \sigma^{-2}\sum_{\ell<t}\mathbf{1}\{s \in \mathcal{S}_\ell\}(\sum_{a \in A_{s,\ell}}\Phi_a\hat{\mathbf{w}}_{s,t}(a))$. After the hyper-parameter $\mu_t$ is sampled from $Q_t(\mu)$, we sample task parameter $\theta_{s,t} \sim \mathcal{N}(\theta; \tilde{\mu}_{s,t}, \tilde{\Sigma}_{s,t})$ for task $s$, where $\tilde{\mu}_{s,t} = \tilde{\Sigma}_{s,t}(\Sigma_0^{-1}\mu_t + B_{s,t})$ is the posterior mean, $\tilde{\Sigma}_{s,t}^{-1} = \Sigma_0^{-1} + G_{s,t}$ the posterior covariance matrix. Such posterior of a linear model is obtained with a Gaussian prior $\mathcal{N}(\mu_t, \Sigma_0)$ and Gaussian observations $(Y_{s,\ell})_{\ell<t, s \in \mathcal{S}_\ell}$ by Bayes rule.

On the other hand, we also need to handle $\mathbb{P}(\theta_{s,*} = \theta | H_t)$. It is not difficult to see that, in the multi-task Gaussian linear bandit/semi-bandit setting, $\theta_{s,*}|H_t$ is Gaussian and denoted as $\mathbb{P}(\theta_{s,*} = \theta | H_t) = \mathcal{N}(\theta; \hat{\mu}_{s,t}, \hat{\Sigma}_{s,t})$. According to Lemma B.1, $\hat{\mu}_{s,t}$ and $\hat{\Sigma}_{s,t}$ have the following explicit forms:

$$\hat{\mu}_{s,t} = \tilde{\Sigma}_{s,t}(\Sigma_0^{-1}\bar{\mu}_t + B_{s,t}), \quad \hat{\Sigma}_{s,t} = \tilde{\Sigma}_{s,t} + \tilde{\Sigma}_{s,t}\Sigma_0^{-1}\bar{\Sigma}_t\Sigma_0^{-1}\tilde{\Sigma}_{s,t}. \tag{3}$$

## 5 Bayes Regret Bounds

In this section, we provide improved regret bounds of hierarchical Bayesian bandit algorithms for multi-task Gaussian linear bandit/semi-bandit problem. Concretely, we provide improved analysis for HierTS in the sequential linear bandit setting (Sections 5.1), propose a novel HierBayesUCB bandit algorithm with logarithmic regret guarantee (Section 5.2), develop regret bounds for these two algorithms in the concurrent linear bandit setting (Section 5.3), and finally extend these two algorithms to the semi-bandit setting (Section 5.4) with improved regret bounds. In the proof for our theoretical results, the most important step is to give an upper bound on the so-called *posterior variance* $\mathcal{V}_{m,n}$, which in the multi-task linear bandit setting is defined and upper bounded as follow:

$$\mathcal{V}_{m,n} \triangleq \mathbb{E}\Big[\sum_{t \geq 1}\sum_{s \in \mathcal{S}_t}\|A_{s,t}\|_{\hat{\Sigma}_{s,t}}^2\Big] \leq O\big(md\log(\frac{n}{d}) + d\log(\frac{m}{d})\big). \tag{4}$$

Although the above bound on $\mathcal{V}_{m,n}$ achieves the same order (w.r.t. $m, n$ and $d$) as that in the latest bound of [17, Sect B], our bound has a smaller multiplicative factor (see more details in Table 4). In the multi-task semi-bandit setting, the posterior variance is $\mathcal{V}_{m,n} \triangleq \mathbb{E}\sum_{t \geq 1}\sum_{s \in \mathcal{S}_t}\sum_{a \in A_{s,t}}\|\Phi_a\|_{\hat{\Sigma}_{s,t}}^2$ and can be bounded in a similar way. To finish the whole proof, our strategy consists of two main steps: **(1)** The first step is to transform the multi-task Bayes regret $\mathcal{BR}(m,n)$ into an intermediate regret upper bound that involves the posterior variance $\mathcal{V}_{m,n}$ as the dominant term. **(2)** The second step is to bound $\mathcal{V}_{m,n}$ with Eq. (4). Combining the results in steps **(1)** and **(2)** yields Bayes regret bound for multi-task hierarchical Bayesian bandit/semi-bandit algorithms. Detailed comparisons between our regret bounds and others in the bandit setting are shown in Table 1. Next, we define $c_1 = \sigma^2 + B^2\lambda_1(\Sigma_0)$, $c_2 = \sigma^2 + B^2\lambda_1(\Sigma_0) + B^2\lambda_1(\Sigma_q)\kappa(\Sigma_0)$ to be used through the whole Section 5.

Table 1: Different Bayes regret bounds for multi-task $d$-dimensional linear (or $K$-armed) bandit problem in the sequential setting. $m$ is the number of tasks, $n$ the number of iterations per task, $\mathcal{A}$ is the action set. **Bayes Regret Bound =Bound I + Bound II + Negligible Terms**, where **Bound I** is the regret bound for solving $m$ tasks, **Bound II** the regret bound for learning hyper-parameter $\mu_*$.

| Bayes Regret Bound | $|\mathcal{A}|$ | Bound I | Bound II |
|---|---|---|---|
| [25, Theorem 3] | Finite | $O\big(m\sqrt{Kn\log n}\big)$ | $O\big(n^2 K\sqrt{m\log{(n)}\log{(K)}}\big)$ |
| [7, Theorem 5] | Finite | $O\big(m\sqrt{dn(\log n)\log{(n^2|\mathcal{A}|)}}\big)$ | $O\big(\sqrt{dmn}(\log m)\log{(n|\mathcal{A}|)}\big)$ |
| [17, Theorem 3] | Infinite | $O\big(md\sqrt{n\log{(\frac{n}{d})}}\log{(mn)}\big)$ | $O\big(d\sqrt{mn}\log{(m)}\log{(mn)}\big)$ |
| Our Theorem 5.1 | Infinite | $O\big(md\sqrt{n\log{(\frac{n}{d})}}\big)$ | $O\big(d\sqrt{mn\log{(\frac{m}{d})}}\big)$ |
| Our Theorem 5.2 | Finite | $O\big(md\log{(\frac{n}{d})}\log{(mn)}\big)$ | $O\big(d\log{(\frac{m}{d})}\log{(mn)}\big)$ |

## 5.1 Improved Regret Bound for HierTS in the Sequential Bandit Setting

In the sequential bandit setting, $|\mathcal{S}_t| = 1$. Then, conditioned on $H_t$, it is not difficult to see that in Bayes regret, each term $\mathbb{E}[\theta_{s,*}^\top A_{s,*} - \theta_{s,*}^\top A_{s,t}|H_t] = \mathbb{E}\big[(\theta_{s,*} - \hat{\mu}_{s,t})^\top A_{s,*}\big|H_t\big]$, and we use a novel Cauchy-Schwartz type inequality from [21, Prop 2] to bound $\mathbb{E}\big[(\theta_{s,*} - \hat{\mu}_{s,t})^\top A_{s,*}\big|H_t\big]$, leading to $\mathcal{BR}(m,n) \leq \mathbb{E}\Big[\sum_{t\geq 1}\sum_{s\in\mathcal{S}_t}\sqrt{d\mathbb{E}\big[\big((\theta_{s,*} - \hat{\mu}_{s,t})^\top A_{s,t}\big)^2\big|H_t\big]}\Big]$. Expand the expression in the right hand side of the above inequality, we then have $\mathcal{BR}(m,n) \leq \mathbb{E}\sum_{t,s\in\mathcal{S}_t}\sqrt{dA_{s,t}^\top \hat{\Sigma}_{s,t} A_{s,t}} \leq \sqrt{dmn\mathcal{V}_{m,n}}$, reducing the Bayes regret bound to the posterior variance bound problem. Recalling Eq. (4) achieves our first improved Bayes regret upper bound in the sequential linear bandit setting.

**Theorem 5.1** *(Near-Optimal Sequential Regret) Let $|\mathcal{S}_t| = 1$ for any round t. Then in the multi-task Gaussian linear bandit setting, the Bayes regret upper bound of HierTS is as follow:*

$$\mathcal{BR}(m,n) \leq d\sqrt{2mn}\sqrt{mc_1\log{(1 + \frac{n}{d})} + c_2\log{(1 + \frac{m\operatorname{Tr}(\Sigma_q\Sigma_0^{-1})}{d})}}.$$

Our explanations for the above sequential regret bound are three-fold: **(1)** The term $md\sqrt{nc_1\log{(1 + n/d)}}$ represents the regret bound for solving $m$ bandit tasks, whose parameters $\theta_{s,*}$ are drawn i.i.d. from the prior distribution $\mathcal{N}(\mu_*, \Sigma_0)$. Under this assumption, no task provides information for any other task, and hence this bound is linear in $m$. Similar observation was also pointed out by [25, 7, 17]. **(2)** The term $d\sqrt{mnc_2\log{(1 + m\operatorname{Tr}(\Sigma_q\Sigma_0^{-1})/d)}}$ represents the regret bound for learning the hyper-parameter $\mu_*$. Such bound is sublinear in $m$ and is not a dominant term when $m$ is large. **(3)** For a large $m$, the averaged Bayes regret bound across $m$ tasks is of $\mathcal{BR}(m,n)/m = O(d\sqrt{n\log n})$, and strengthens the latest averaged bound $O(d\sqrt{n}\log n)$ in [17, Thm 3] by a factor $\sqrt{\log n}$. Besides, since the lower Bayes regret bound for any Bayesian bandit algorithm is $\Omega(d\sqrt{n})$ [30], our task-averaged Bayes regret bound is within $O(\sqrt{\log n})$ of optimality and hence is called 'Near-Optimal' sequential regret bound. We further make a detailed comparison between our regret bound in Theorem 5.1 and the regret bound [17, Thm 3] in the following remark.

**Remark 5.1** *(**Improvements of Our Theorem 5.1 over the Latest One**) Our sequential regret bound has two improvements over the latest one in [17, Thm 3, shown in Table 1]: **(1)** We remove the additional $\sqrt{\log{(mn)}}$ factor in both the regret bound for solving $m$ bandit tasks and the regret bound for learning the hyper-parameter $\mu_*$. **(2)** In the regret bound for learning hyper-parameter $\mu_*$, [17] has a multiplicative factor $\kappa^2(\Sigma_0)$, whereas our multiplicative factor is $\kappa(\Sigma_0)$. Such improvement is achieved by using technical matrix analysis proposed in Lemma C.1. and explained in Remark A.1.*

## 5.2 Logarithmic Regret Bound for HierBayesUCB in the Sequential Bandit Setting

In this section, we attempt to provide further improved Bayes regret bounds for hierarchical bandit algorithms in the sequential bandit setting. Because the task averaged Bayes regret bound in Theorem 5.1 is near optimal, it is not easy to derive improved Bayes regret bounds under the same assumptions. Therefore, we further assume that the action set $\mathcal{A}$ is finite, and propose a novel

hierarchical Bayesian bandit algorithm, named Hierarchical BayesUCB (HierBayesUCB), for multi-task linear bandit problem. The pseudo-code of our proposed algorithm is shown in Algorithm 1.

Next, we introduce some necessary notations. Let $\Delta_{s,t} = \theta_{s,*}^\top (A_{s,*} - A_{s,t})$, $\Delta_{s,\min} = \min_{a \in \mathcal{A} \setminus \{A_{s,*}\}} (\theta_{s,*}^\top A_{s,*} - \theta_{s,*}^\top a)$, $\Delta_{\min} = \min_{s \in [m]} \Delta_{s,\min}$. For any $\epsilon > 0$, let $\Delta_{\min}^\epsilon = \max\{\epsilon, \Delta_{\min}\}$. Define the event $E_{s,t} = \{\forall a \in \mathcal{A} : |a^\top(\theta_{s,*} - \hat{\mu}_{s,t})| \leq \sqrt{2\log\frac{1}{\delta}}\|a\|_{\hat{\Sigma}_{s,t}}\}$. Then, analogous to [3], we decompose the Bayes regret $\mathcal{BR}(m,n) = \mathbb{E}\sum_{t \geq 1}\sum_{s \in \mathcal{S}_t}\Delta_{s,t}$ into three terms: $\mathbb{E}\sum_{t \geq 1, s \in \mathcal{S}_t} \Delta_{s,t}\big[\mathbf{1}\{\Delta_{s,t} \geq \epsilon, E_{s,t}\} + \mathbf{1}\{\Delta_{s,t} < \epsilon, E_{s,t}\} + \mathbf{1}\{\bar{E}_{s,t}\}\big]$. We can bound the last two terms trivially with $mn[\epsilon + 2\delta(\max_{t,s}|\Delta_{s,t}|) \cdot |\mathcal{A}|]$. For the first term, we use the fact that $A_{s,t}|H_t \overset{\text{i.i.d.}}{\sim} A_{s,*}|H_t$, as well as the Upper Confidence Bound (UCB) technique to reduce it to an intermediate upper bound $\big(\sum_{t \geq 1, s \in \mathcal{S}_t}\|A_{s,t}\|_{\hat{\Sigma}_{s,t}}^2 \log\frac{1}{\delta}\big)/\min_{s,t}|\Delta_{s,t}|$. Combining the upper bound over $\mathcal{V}_{m,n}$ in Eq. (4), HierBayesUCB can achieve the following logarithmic Bayes regret bound in the sequential bandit setting (the logarithmic bound can be extended to the concurrent setting).

**Theorem 5.2** *(Logarithmic Sequential Regret of HierBayesUCB) Let $|\mathcal{S}_t| = 1$ for any round $t$, and the action set $\mathcal{A}$ is finite with $|\mathcal{A}| < \infty$. Then in the multi-task Gaussian linear bandit setting, for any $\delta \in (0,1)$, $\epsilon > 0$, the Bayes regret $\mathcal{BR}(m,n)$ of HierBayesUCB is upper bounded by*

$$mn\Big[\epsilon + 4B\delta\lambda_1^{\frac{1}{2}}(\Sigma_0 + \Sigma_q)\big(d^{\frac{1}{2}} + \|\mu_q\|_{\hat{\Sigma}_{s,1}^{-1}}\big)|\mathcal{A}|\Big] + \mathbb{E}\Big[\frac{16d\log\frac{1}{\delta}}{\Delta_{\min}^\epsilon}\Big]\Big[mc_1\log\big(1 + \frac{n}{d}\big) + c_2\log\big(1 + \frac{m\,\mathrm{Tr}(\Sigma_q\Sigma_0^{-1})}{d}\big)\Big].$$

We give more explanations for the above sequential regret in terms of the following five aspects:
**(1)** If let $\delta = 1/(mn)$, $\epsilon = 1/(mn)$ and $\Delta_{\min} >> \epsilon$, the above sequential regret bound is of $O\big(\log(mn)(md\log(\frac{n}{d}) + d\log(\frac{m}{d}))\big)$. The term $O(md\log(mn)\log(\frac{n}{d}))$ represents the regret bound for solving $m$ bandit tasks and is linear in $m$. Such bound is sharper than the corresponding bound $O(md\sqrt{n\log(\frac{n}{d})})$ in our Theorem 5.1 by a multiplicative factor $O(\sqrt{\log(n/d)/n}\log(mn))$, which is less than 1 especially when $m \leq n$. **(2)** The term $O(d\log(mn)\log(\frac{m}{d}))$ represents the regret bound for learning the hyper-parameter $\mu_*$, and its contribution to the Bayes regret bound can be negligible. Besides, this bound is sharper than the bound $O(d\sqrt{mn\log(m/d)})$ in our Theorem 5.1. **(3)** The averaged Bayes regret bound across $m$ tasks can be regarded as $\mathcal{BR}(m,n)/m = O(d\log(mn)\log n)$, which is logarithmic in $n$. Therefore, we call our regret bound as 'Logarithmic' sequential regret bound. Moreover, if there exists a fixed positive integer $i << n$, such that $m \leq n^i$, then our task-averaged Bayes regret $\mathcal{BR}(m,n)/m = O(d\mathbb{E}[\frac{1}{\Delta_{\min}^\epsilon}]\log^2 n)$ matches the latest single-task Bayes regret bound in [3, Thm 5] and is remarkably similar to the frequentist regret $O(d\Delta_{\min}^{-1}\log^2 n)$ in [1, Thm 5] . **(4)** We can obtain sharper bounds by setting $\delta, \epsilon$ as different values. For example, by setting $\delta = 1/n$, our regret bound becomes $O([mn\epsilon + m] + \frac{\log n}{\Delta_{\min}^\epsilon}m\log n)$, which is of order $O(m\log^2 n)$ if we set $\epsilon = 1/(mn)$ and the gap $\Delta_{\min} >> \epsilon$ is large. **(5)** We also need to point out that, the Bayes regret bound in Theorem 5.2 scales with $\mathbb{E}[\frac{1}{\Delta_{\min}^\epsilon}]$. If the gap $\Delta_{\min} \leq 1/(mn)$, then $\Delta_{\min}^\epsilon = 1/(mn)$ and this may cause a large Bayes regret upper bound.

## 5.3 Improved Regret Bounds of HierTS and HierBayesUCB in the Concurrent Bandit Setting

In the concurrent bandit setting, there exists a positive integer $L \leq m$, such that $1 \leq |\mathcal{S}_t| \leq L$. The concurrent bandit setting is thus more challenging than the sequential bandit setting, because the agent in the concurrent setting needs to interact with multiple bandit tasks in parallel at each round $t \geq 1$, and the hyper-posterior $Q_t$ will not be updated until the end of round $t$. Therefore, we need to make an additional assumption on the action space $\mathcal{A}$ as follow to facilitate our theoretical analysis.

**Assumption 5.1** *There exist actions $\{a_i\}_{i=1}^d \subseteq \mathcal{A}$, a constant $\beta > 0$, such that $\lambda_d(\sum_{i=1}^d a_i a_i^\top) \geq \beta$.*

This assumption is also used in previous works [7, 17] for hierarchical Bayesian linear bandit. It indicates that $\sum_{i=1}^d a_i a_i^\top$ is a positive definite matrix, and does not weaken the generality of our theoretical results. Actually, if $\mathbb{R}^d$ is not spanned by actions in $\mathcal{A}$, we can project $\mathcal{A}$ into a subspace where the assumption holds. We also need to modify the HierTS algorithm to let the agent take the basic actions $\{a_i\}_{i=1}^d$ for the first $d$ interactions in any task $s \in [m]$. This modification guarantees that the agent explores all directions within the task. Such exploration is very similar to the initialization method in UCB type $K$-arm bandit algorithms [6, 4], which choose to pull each arm in the first $K$ rounds. Define $c_3 = 1 + B^2\sigma^{-2}\kappa(\Sigma_0)\big[\lambda_1(\Sigma_0) + \sigma^2/\beta\big]$ that will be used throughout the concurrent

Table 2: Different Bayes regret bounds for multi-task semi-bandit problem. **Bayes Regret Bound =Bound I + Bound II + Negligible Terms**. $m$ is the number of tasks, $n$ the number of iterations per task, $K$ the size of action set, $L$ the number of pulled actions at each round ($1 \leq L \leq K$). **Bound I** is the regret bound for solving $m$ tasks, **Bound II** the regret bound for learning hyper-parameter $\mu_*$.

| Bayes Regret Bound | $\mathcal{A}$ | Bound I | Bound II |
|---|---|---|---|
| [7, Theorem 6] | $[K]$ | $O\big(m\sqrt{nKL}\log n \log{(nK)}\big)$ | $O\big(\sqrt{mnKL}\log m \log{(nK)}\big)$ |
| Our Theorem 5.4 | $[K]$ | $O\big(m\sqrt{nL}\log{(nL)}\log{(nK)}\big)$ | $O\big(L^{\frac{3}{2}}\sqrt{mn}\log m \log{(nK)}\big)$ |
| Our Theorem 5.5 | $[K]$ | $O\big(mL\log{(nL)}\log{(mnK)}\big)$ | $O\big(L^3 \log{(m)}\log{(mnK)}\big)$ |

setting. Then, analogous to the proof for Theorem 5.1, we bound $\sqrt{mn\mathcal{V}_{m,n}}$ with a more refined analysis, achieving the following improved Bayes regret bound for HierTS in the concurrent setting.

**Theorem 5.3** *Under Assumption 5.1, let $1 \leq |\mathcal{S}_t| \leq L$ for any round $t \geq 1$. Then in the multi-task Gaussian linear bandit setting, the Bayes regret $\mathcal{BR}(m,n)$ of HierTS is upper bounded by*

$$2Bmd\sqrt{\lambda_1(\Sigma_0 + \Sigma_q)}(\sqrt{d} + \|\mu_q\|_{\hat{\Sigma}_{s,1}^{-1}}) + d\sqrt{mn}\sqrt{2mc_1\log{(1+\frac{n}{d})} + 2c_2c_3\log{(1+\frac{m\operatorname{Tr}(\Sigma_q\Sigma_0^{-1})}{d})}}.$$

The concurrent regret bound in Theorem 5.3 achieves almost the same order (w.r.t. $m, n, d$) as the sequential regret bound in Theorem 5.1, but differs in two aspects: **(1)** The bound for learning $m$ i.i.d. bandit tasks has an additional term $Bmd\sqrt{\lambda_1(\Sigma_0 + \Sigma_q)}(\sqrt{d} + \|\mu_q\|_{\hat{\Sigma}_{s,1}^{-1}})$. This is due to the fact that we take the basic actions $\{a_i\}_{i=1}^d$ first for each task $s \in [m]$ in the modified HierTS algorithm. **(2)** The bound for learning the hyper-parameter $\mu_*$ has an additional multiplicative factor $c_3$. This is the price for deriving regret bounds in the concurrent setting. Nevertheless, when compared with the latest concurrent regret bound in [17, Thm 4] for HierTS, our concurrent regret bound in Theorem 5.3 removes the $\sqrt{\log{(mn)}}$ factor in both the regret bound for learning $m$ bandit tasks and the regret bound for learning hyper-parameter $\mu_*$. Detailed comparisons between different concurrent regret bounds for multi-task linear bandit setting are listed in Table 3. Furthermore, utilizing the proof strategy to demonstrate the logarithmic sequential regret for HierBayesUCB in our Theorem 5.2, we can analogously develop a logarithmic concurrent regret upper bound for HierBayesUCB algorithm, which is deferred to our Theorem C.2 in Appendix C due to the limited space of the main paper.

## 5.4 Improved Regret Bounds for HierTS and HierBayesUCB in the Semi-Bandit Setting

In this section, we extend the HierTS and HierBayesUCB algorithms to the multi-task Gaussian combinatorial semi-bandit setting. The pseudo-code of them is shown in Algorithm 2. Algorithm 2 is very similar to Algorithm 1 (i.e. the multi-task linear bandit algorithms), except that the combinatorial HierTS in Algorithm 2 uses the approximation/randomized algorithm ORACLE to solve combinatorial problem $A_* \in \arg\max_{A \in \mathscr{A}} \sum_{a \in A} \mathbf{w}(a)$ and denotes the solution as $A_* = \operatorname{ORACLE}(\mathcal{A}, \mathscr{A}, \mathbf{w})$. We adopt the ORACLE operator as in the seminal works [11, 38] to guarantee the efficiency of combinatorial HierTS semi-bandit algorithm. In this section, we only consider the sequential semi-bandit setting (i.e. $|\mathcal{S}_t| = 1$) for ease of presentation, and our results can be extended to the concurrent semi-bandit setting. Then, define $c_4 = \sigma^2 + B^2 L \lambda_1(\Sigma_0) + B^2 \lambda_1(\Sigma_q)\kappa(\Sigma_0)$, we first derive the Bayes regret upper bound for combinatorial HierTS algorithm in the sequential semi-bandit setting.

**Theorem 5.4** *Let $|\mathcal{S}_t| = 1$ for any $t \geq 1$. Let $c \geq \sqrt{2\ln{\big(\frac{nKB\lambda_1(\Sigma_0)}{\sqrt{2\pi}}\big)}}$, then in the multi-task Gaussian semi-bandit setting, the Bayes regret upper bound of combinatorial HierTS is:*

$$\mathcal{BR}(m,n) \leq m + c\sqrt{mnL}\sqrt{2c_1 m\log{(1+\frac{nL}{d})} + 2c_4 Ld\log(1+\frac{m\operatorname{Tr}(\Sigma_0^{-1}\Sigma_q)}{d})}.$$

Detailed comparisons between different Bayes regret bounds for multi-task semi-bandit problem are listed in Table 2. We can see that, in our Theorem 5.4, both the regret bound $O(m\sqrt{n}\log n)$ for learning $m$ tasks and the regret bound $O(\sqrt{mn\log m \log n})$ for learning hyper-parameter $\mu_*$ can achieve the same order (w.r.t. $m$ and $n$) when compared with the latest bound in [7, Thm 6]. Besides, our Bayes regret bound is logarithmic in the number $K$ of items, whereas the Bayes regret bound in

[7, Thm 6] is sublinear in $K$. Therefore, our regret bound becomes sharper when the size of action set is very large, e.g. $K >> L$. Next, we derive a gap-dependent logarithmic multi-task Bayes regret bound for our proposed combinatorial HierBayesUCB algorithm in the sequential semi-bandit setting.

**Theorem 5.5** *Let $|\mathcal{S}_t| = 1$ for any $t \geq 1$. Then for any $\epsilon > 0, \delta \in (0,1)$, in the multi-task Gaussian semi-bandit setting, the Bayes regret $\mathcal{BR}(m,n)$ of combinatorial HierBayesUCB is bounded by*

$$mn\big[\epsilon + 4LBK\delta\lambda_1^{\frac{1}{2}}(\Sigma_0+\Sigma_q)(d^{\frac{1}{2}}+\|\mu_q\|_{\hat{\Sigma}_{s,1}^{-1}})\big] + \mathbb{E}\big[\frac{8L\log\frac{1}{\delta}}{\Delta_{\min}^{\epsilon}}\big]\big[2c_1 m\log(1+\frac{nL}{d}) + 2c_4 Ld\log(1+\frac{m\operatorname{Tr}(\Sigma_0^{-1}\Sigma_q)}{d})\big].$$

In Theorem 5.5, if we set $\delta = 1/(mnK)$, $\epsilon = 1/(mn)$, and $\Delta_{\min} >> \epsilon$, then the regret bound $O\big(m\log n\log(mn)\big)$ for learning $m$ tasks is logarithmic in $n$. Such bound is sharper than the latest one $O(m\sqrt{n}\log n)$ in [7, Thm 6] for multi-task semi-bandit. The regret bound $O(\log m\log(mn))$ for learning hyper-parameter $\mu_*$ is also sharper than that of $O(\sqrt{mn\log m\log n})$ in [7, Thm 6]. Besides, since $\delta = 1/(mnK)$, the whole Bayes regret bound is also logarithmic in the number $K$ of items. Nevertheless, we should point out that our bounds hold for the multi-task semi-bandits with linear generalization, but [7] focuses on the multi-task $K$-arm semi-bandits without feature matrix $\Phi$.

### 5.5 Technical Novelties for Deriving Improved Regret Bounds

In this section, we summarize our technical novelties in terms of the following three aspects:

**(1)** For the improved regret bound for HierTS in Theorem 5.1: our proof has three novelties: **(i)** We apply a novel Cauchy-Schwartz type inequality in Lemma A.2 to bound $\mathbb{E}\big[(\theta_{s,*}-\hat{\mu}_{s,t})^\top A_{s,*}|H_t\big] \leq \sqrt{d\mathbb{E}\big[\big((\theta_{s,*}-\hat{\mu}_{s,t})^\top A_{s,t}\big)^2|H_t\big]}$, leading to a sharper bound without $\sqrt{\log(mn)}$ factor:

$$\mathcal{BR}(m,n) \leq \mathbb{E}\sum_{t,s\in\mathcal{S}_t}\sqrt{dA_{s,t}^\top\hat{\Sigma}_{s,t}A_{s,t}} \leq \sqrt{dmn\mathcal{V}_{m,n}} \leq O(m\sqrt{n\log n}).$$

**(ii)** We use a more technical positive semi-definite matrix decomposition analysis (i.e. our Lemma A.1) to reduce the multiplicative factor $\kappa^2(\Sigma_0)$ to $\kappa(\Sigma_0)$. **(iii)** Define a new matrix $\tilde{X}_{s,t}$ such that the denominator in the regret is $\sigma^2 + B^2\lambda_1(\Sigma_0)$, not just $\sigma^2$, avoiding the case that the variance serves alone as the denominator. Such technical novelties are also listed explicitly in Table 4.

**(2)** For the improved regret bound for HierBayesUCB in Theorem 5.2 in the sequential bandit setting: our novelty lies in decomposing the Bayes regret $\mathcal{BR}(m,n) = \mathbb{E}\sum_{t\geq 1}\sum_{s\in\mathcal{S}_t}\Delta_{s,t}$ into three terms:

$$\mathbb{E}\sum_{t\geq 1}\sum_{s\in\mathcal{S}_t}\Delta_{s,t} = \mathbb{E}\sum_{t\geq 1, s\in\mathcal{S}_t}\Delta_{s,t}\big[\mathbf{1}\{\Delta_{s,t}\geq\epsilon, E_{s,t}\} + \mathbf{1}\{\Delta_{s,t}<\epsilon, E_{s,t}\} + \mathbf{1}\{\bar{E}_{s,t}\}\big],$$

and bounding the first term with a new method as well as the property of BayesUCB algorithm as

$$\mathbb{E}\Delta_{s,t}\mathbf{1}\{\Delta_{s,t}\geq\epsilon, E_{s,t}\} = \mathbb{E}\frac{\Delta_{s,t}^2}{\Delta_{s,t}}\mathbf{1}\{\Delta_{s,t}\geq\epsilon, E_{s,t}\} \leq \mathbb{E}\frac{C_{t,s,A_{s,t}}^2}{\Delta_{\min}^\epsilon},$$

resulting in the final improved gap-dependent regret bound for HierBayesUCB as follows

$$\big(\sum_{t\geq 1, s\in\mathcal{S}_t}\|A_{s,t}\|_{\hat{\Sigma}_{s,t}}^2\log\frac{1}{\delta}\big)/\Delta_{\min}^\epsilon \leq O\big(m\log(n)\log\frac{1}{\delta}\big) \xrightarrow{\delta=1/mn} O\big(m\log(n)\log(mn)\big).$$

**(3)** For the improved regret bounds for HierTS and HierBayesUCB in the concurrent setting and in the semi-bandit setting: besides the aforementioned technical novelties in **(1)** and **(2)**, the additional technical novelty lies in leveraging more refined analysis (e.g. using Woodbury matrix identity) to bound the gap between matrices $\bar{\Sigma}_{t+1}^{-1}$ and $\bar{\Sigma}_t^{-1}$ (more details is shown in Lemma C.1 and Eq. (6)).

## 6 Experiments

In this section, we conduct experiments in the linear bandit setting to verify our theoretical results. Specifically, we show the influence of hyper-parameters (e.g. $m, n, L$) to the multi-task Bayes regret of HierTS and HierBayesUCB, to validate the consistency between their regret bounds and practical performance. Besides, we compare the performance between our algorithms and other baselines, to show the effectiveness of hierarchical Bayesian bandit algorithms in the multi-task bandit setting.

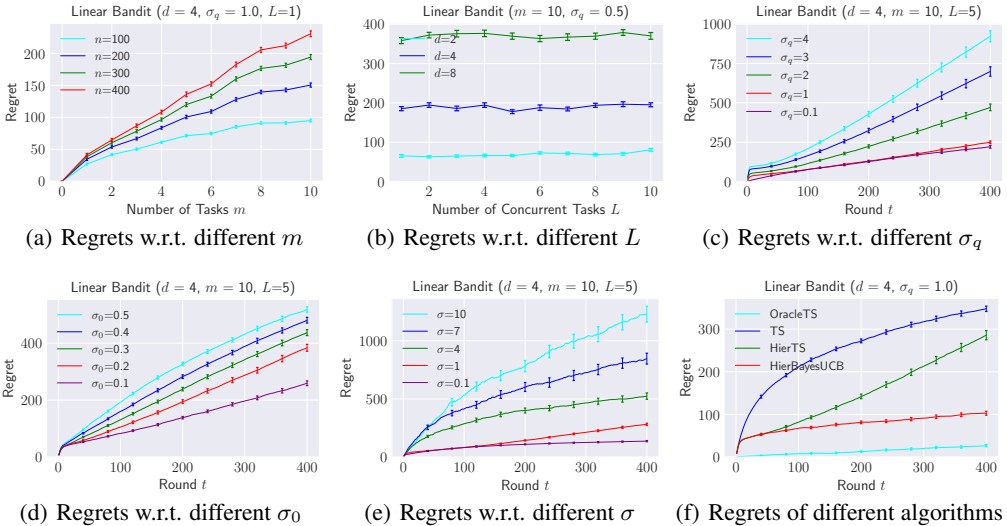

Figure 1: Regrets of HierTS algorithm with respect to (w.r.t.) different hyper-parameters.

**Experimental Setting.** We follow the same experimental setting as that in [7, 17]. Concretely, we conduct linear bandit experiments with Gaussian reward. The synthetic problem is defined as follows. In most experiments, we set the number of total tasks as $m = 10$, the dimension of action space as $d = 4$, the number of concurrent tasks as $L = 5$, the number of rounds as $n = 200m/L$. We focus on the finite action space with $|\mathcal{A}| = 10$, and each action is sampled uniformly from $[-0.5, 0.5]^d$. In hierarchical Bayesian model, we set the hyper-prior as zero-mean isotropic Gaussian distribution $\mathcal{N}(\mu_q, \Sigma_q) = \mathcal{N}(\mathbf{0}, \Sigma_q)$, where $\Sigma_q = \sigma_q^2 I_d$; and set the task variance $\Sigma_0 = \sigma_0^2 I_d$. Unless otherwise stated, we set $\sigma_q = 1$, $\sigma_0 = 0.1$, $\sigma^2 = 0.5$ for each task in most experiments. We exhibit the regret performance of HierTS algorithm with respect to five hyper-parameters $m, L, \sigma_q, \sigma_0, \sigma$ in Figure 1 (a)-(e) respectively. The regret performance of HierBayesUCB is shown in Figure 2 of Appendix F.

Besides, we compare HierTS/HierBayesUCB with other two TS type algorithms that do not learn the hyper-parameter $\mu^*$ in a hierarchical Bayesian model. The first baseline is the vanilla TS algorithm that samples task parameter $\theta_{s,*}$ from the marginal prior $\mathcal{N}(\mu_q, \Sigma_q + \Sigma_0)$. The second baseline is an idealized TS algorithm that knows $\mu_*$ exactly and uses the true prior $\mathcal{N}(\mu_*, \Sigma_0)$. We call the second baseline as OracleTS, since this TS algorithm accesses more information of $\mu_*$ than HierTS and vanilla TS algorithm. We show the regret performance of these four bandit algorithms in Figure 1 (f).

**Experimental Results.** From Figure 1, we can observe that: **(1)** In plot (a), the multi-task regret becomes larger with the increase of $m$ and $n$, which is consistent with our regret upper bound in Theorems 5.1. **(2)** In plot (b), the regret increases with a higher dimension $d$. The number $L$ of the concurrent tasks seems do not have a large impact on regret. **(3)** In plots (c)-(e), the regret decreases with a smaller variance (e.g. $\sigma_q$, $\sigma_0$ and $\sigma$) in hierarchical Bayesian model, validating the provable benefits of variance-reduction in regret minimization, which is revealed in our multi-task Bayes regret upper bounds. **(4)** The task-averaged regret of HierTS is tighter than that of single-task TS algorithm, empirically demonstrating the advantages of multi-task Bayesian bandit optimization paradigm over single-task bandit learning. **(5)** Our proposed HierBayesUCB achieves lower regret than HierTS.

## 7 Conclusions

This paper provides improved Bayes regret bounds for hierarchical Bayesian bandit algorithms in the multi-task Gaussian linear bandit and semi-bandit setting. For linear bandit problem: in the case of infinite action set, we strengthen the preexisting regret bound $O(m\sqrt{n \log n} \log (mn))$ of HierTS to $O(m\sqrt{n \log n})$ by a factor of $O(\sqrt{\log (mn)})$; in the case of finite action set, we propose a novel HierBayesUCB algorithm that achieves logarithmic regret bound $O(m \log (mn) \log n)$ under mild conditions. Our regret bounds in the bandit setting hold when the agent solves tasks sequentially or concurrently. Then, we extend the above HierTS and HierBayesUCB algorithms to the multi-task semi-bandit setting and derive improved regret bounds. The synthetic experiments further support our theoretical results. Our future work aims to extend our bounds to the sub-exponential bandit setting.

## Acknowledgments and Disclosure of Funding

Jiechao sincerely appreciates the financial support from the People's Government of Guangzhou Municipality for his postdoctoral project. We thank all reviewers for their constructive suggestions to improve the quality of this paper. This work was supported in part by the National Key R&D Program of China (Grant No.2023YFF0725001), in part by the National Natural Science Foundation of China (Grant No.92370204), in part by the guangdong Basic and Applied Basic Research Foundation(Grant No.2023B1515120057), in part by Guangzhou-HKUST(GZ) Joint Funding Program (Grant No.2023A03J0008), Education Bureau of Guangzhou Municipality.

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

# APPENDIX

## A Proofs for Regret Bound of HierTS in the Sequential Bandit Setting

We first give the following proposition to bound the posterior variance $\mathbb{E} \sum_{t \geq 1} \sum_{s \in \mathcal{S}_t} \|A_{s,t}\|^2_{\hat{\Sigma}_{s,t}}$ in the sequential setting. We choose to give the worst-case upper bound on $\sum_{t \geq 1} \sum_{s \in \mathcal{S}_t} \|A_{s,t}\|^2_{\hat{\Sigma}_{s,t}}$.

**Proposition A.1** *Let $c_1 = \sigma^2 + B^2 \lambda_1(\Sigma_0)$, $c_2 = \sigma^2 + B^2 \lambda_1(\Sigma_0) + B^2 \lambda_1(\Sigma_q)\kappa(\Sigma_0)$, then*

$$\sum_{t \geq 1} \sum_{s \in \mathcal{S}_t} \|A_{s,t}\|^2_{\hat{\Sigma}_{s,t}} \leq 2mdc_1 \log\left(1 + \frac{n}{d}\right) + 2dc_2 \log\left(1 + \frac{m \operatorname{Tr}\left(\Sigma_0^{-1}\Sigma_q\right)}{d}\right).$$

***Proof.*** Note that $\|A_{s,t}\|^2_{\hat{\Sigma}_{s,t}} = A_{s,t}^\top\left(\tilde{\Sigma}_{s,t} + \tilde{\Sigma}_{s,t}\Sigma_0^{-1}\bar{\Sigma}_t\Sigma_0^{-1}\tilde{\Sigma}_{s,t}\right)A_{s,t}$, then we bound $\sum_{t \geq 1} \sum_{s \in \mathcal{S}_t} \|A_{s,t}\|^2_{\tilde{\Sigma}_{s,t}}$ and $\sum_{t \geq 1} \sum_{s \in \mathcal{S}_t} A_{s,t}^\top\tilde{\Sigma}_{s,t}\Sigma_0^{-1}\bar{\Sigma}_t\Sigma_0^{-1}\tilde{\Sigma}_{s,t}A_{s,t}$ respectively.

**(1) Bounding $\sum_{t \geq 1} \sum_{s \in \mathcal{S}_t} \|A_{s,t}\|^2_{\tilde{\Sigma}_{s,t}}$.** Note that $\tilde{\Sigma}_{s,t} = (\Sigma_0^{-1} + G_{s,t})^{-1} \leq \Sigma_0$, then we have $A_{s,t}^\top\tilde{\Sigma}_{s,t}A_{s,t} \leq B^2\lambda_1(\Sigma_0) < B^2\lambda_1(\Sigma_0) + \sigma^2$. Accordingly, we define a new matrix $\tilde{X}_{s,t} \triangleq (\Sigma_0^{-1} + \frac{1}{B^2\lambda_1(\Sigma_0)+\sigma^2} \sum_{\ell < t} \mathbf{1}\{s \in \mathcal{S}_t\}A_{s,\ell}A_{s,\ell}^\top)^{-1}$, and notice that $\tilde{X}_{s,t} \geq \tilde{\Sigma}_{s,t} = (\Sigma_0^{-1} + \frac{1}{\sigma^2} \sum_{\ell < t} \mathbf{1}\{s \in \mathcal{S}_t\}A_{s,\ell}A_{s,\ell}^\top)^{-1}$, and that $\tilde{X}_{s,1} = \Sigma_0$. Then recall $c_1 = \sigma^2 + B^2\lambda_1(\Sigma_0)$, we have

$$\sum_{t \geq 1} \sum_{s \in \mathcal{S}_t} A_{s,t}^\top\tilde{\Sigma}_{s,t}A_{s,t}$$

$$= c_1 \sum_{t \geq 1} \sum_{s \in \mathcal{S}_t} \frac{A_{s,t}^\top\tilde{\Sigma}_{s,t}A_{s,t}}{\sigma^2 + B^2\lambda_1(\Sigma_0)}$$

$$\leq 2c_1 \sum_{t \geq 1} \sum_{s \in \mathcal{S}_t} \log\left(1 + \frac{A_{s,t}^\top\tilde{\Sigma}_{s,t}A_{s,t}}{\sigma^2 + B^2\lambda_1(\Sigma_0)}\right)$$

$$= 2c_1 \sum_{t \geq 1} \log\left(1 + \frac{A_{\mathcal{S}_t,t}^\top\tilde{\Sigma}_{\mathcal{S}_t,t}A_{\mathcal{S}_t,t}}{\sigma^2 + B^2\lambda_1(\Sigma_0)}\right)$$

$$= 2c_1 \sum_{t \geq 1} \sum_{s=1}^m \mathbf{1}\{\mathcal{S}_t = s\} \log\left(1 + \frac{A_{s,t}^\top\tilde{\Sigma}_{s,t}A_{s,t}}{\sigma^2 + B^2\lambda_1(\Sigma_0)}\right)$$

$$\leq 2c_1 \sum_{t \geq 1} \sum_{s=1}^m \mathbf{1}\{\mathcal{S}_t = s\} \log\left(1 + \frac{A_{s,t}^\top\tilde{X}_{s,t}A_{s,t}}{\sigma^2 + B^2\lambda_1(\Sigma_0)}\right)$$

$$= 2c_1 \sum_{s=1}^m \sum_{t \geq 1} \mathbf{1}\{\mathcal{S}_t = s\} \log\det\left(I + \frac{\tilde{X}_{s,t}^{\frac{1}{2}}A_{s,t}A_{s,t}^\top\tilde{X}_{s,t}^{\frac{1}{2}}}{\sigma^2 + B^2\lambda_1(\Sigma_0)}\right)$$

$$= 2c_1 \sum_{s=1}^m \sum_{t \geq 1} \mathbf{1}\{\mathcal{S}_t = s\}\left[\log\det\left(\tilde{X}_{s,t}^{-1} + \frac{A_{s,t}A_{s,t}^\top}{\sigma^2 + B^2\lambda_1(\Sigma_0)}\right) - \log\det\tilde{X}_{s,t}^{-1}\right]$$

$$= 2c_1 \sum_{s=1}^m \left[\log\det\left(\tilde{X}_{s,mn+1}^{-1}\right) - \log\det\tilde{X}_{s,1}^{-1}\right]$$

$$= 2c_1 \sum_{s=1}^m \log\det\left(I + \frac{1}{\sigma^2 + B^2\lambda_1(\Sigma_0)} \sum_{t \leq mn} \mathbf{1}\{s \in \mathcal{S}_t\}\Sigma_0^{\frac{1}{2}}A_{s,t}A_{s,t}^\top\Sigma_0^{\frac{1}{2}}\right)$$

$$\leq 2dc_1 \sum_{s=1}^m \log \frac{\operatorname{Tr}\left(I + \frac{1}{\sigma^2 + B^2\lambda_1(\Sigma_0)} \sum_{t \leq mn} \mathbf{1}\{s \in \mathcal{S}_t\}\Sigma_0^{\frac{1}{2}}A_{s,t}A_{s,t}^\top\Sigma_0^{\frac{1}{2}}\right)}{d}$$

$$=2dc_1 \sum_{s=1}^{m} \log \Big(1 + \frac{\sum_{t \le mn} \mathbf{1}\{s \in \mathcal{S}_t\} A_{s,t}^\top \Sigma_0 A_{s,t}}{d(\sigma^2 + B^2 \lambda_1(\Sigma_0))}\Big)$$

$$\le 2mdc_1 \log \Big(1 + \frac{n}{d}\Big),$$

where the first inequality holds because the basic inequality $x \le 2\log(1+x)$, $\forall x \in [0,1]$; the third inequality holds due to the mean-value inequality $(\prod_{i=1}^{d} \lambda_i)^{\frac{1}{d}} \le \frac{\sum_{i=1}^{d} \lambda_i}{d}$, for any $\lambda_i \ge 0$; the last inequality holds due to the fact that the agent interacts with each task $s \in [m]$ at most $n$ times.

Before bounding the remaining $\sum_{t \ge 1} \sum_{s \in \mathcal{S}_t} A_{s,t}^\top \tilde{\Sigma}_{s,t} \Sigma_0^{-1} \bar{\Sigma}_t \Sigma_0^{-1} \tilde{\Sigma}_{s,t} A_{s,t}$, we introduce the following lemma.

**Lemma A.1** *If the square matrices $A > 0, B \ge 0$, then $\lambda_1\big[\big((I + AB)(I + BA)\big)^{-1}\big] \le \frac{\lambda_1(A)}{\lambda_d(A)}$.*

***Proof.*** According to Theorem 7.6.1 in Page 485 of [18], there exists a non-singular matrix $S$, such that $A = SS^\top$, and $B = S^{-\top} \Lambda S^{-1}$, in which $\Lambda \ge 0$ is a diagonal matrix. Then we have $AB = S\Lambda S^{-1}$, $BA = S^{-\top} \Lambda S^\top$. Therefore, applying Weyl's inequality we have

$$\lambda_d\big((I + AB)(I + BA)\big)$$
$$=\lambda_d\big((I + S\Lambda S^{-1})(I + S^{-\top}\Lambda S^\top)\big)$$
$$=\lambda_d\big(S(I + \Lambda)S^{-1}S^{-\top}(I + \Lambda)S^\top\big)$$
$$=\lambda_d\big(S^\top S(I + \Lambda)S^{-1}S^{-\top}(I + \Lambda)\big)$$
$$\ge\lambda_d\big(S^\top S\big)\lambda_d\big((I + \Lambda)S^{-1}S^{-\top}(I + \Lambda)\big)$$
$$\ge\lambda_d\big(S^\top S\big)\lambda_d\big(S^{-1}S^{-\top}\big)\lambda_d\big((I + \Lambda)^2\big)$$
$$\ge\lambda_d\big(S^\top S\big)/\lambda_1\big(S^1 S^\top\big) = \lambda_d(A)/\lambda_1(A). \qquad \square$$

**Remark A.1** *(**Smaller multiplicative factor than the latest one in [17]**) The improvement lies in our sharper upper bound on $\lambda_1(\Sigma_0^{-1}\tilde{\Sigma}_{s,t}\tilde{\Sigma}_{s,t}\Sigma_0^{-1})$, and detailed explanations are two-fold:*
*(1) Previous work [17, Appendix B] directly used Weyl's inequality to upper bound*

$$\lambda_1(\Sigma_0^{-1}\tilde{\Sigma}_{s,t}\tilde{\Sigma}_{s,t}\Sigma_0^{-1}) \le \lambda_1^2(\Sigma_0^{-1})\lambda_1^2(\tilde{\Sigma}_{s,t}) \le \lambda_1^2(\Sigma_0^{-1})\lambda_1^2(\Sigma_0) = \kappa^2(\Sigma_0).$$

*(2) Instead of directly using Weyl's inequality, we first propose Lemma A.1 which uses positive semi-definite matrix diagonalization technique to bound*

$$\lambda_1\big[\big((I + AB)(I + BA)\big)^{-1}\big] \le \frac{\lambda_1(A)}{\lambda_d(A)}.$$

*Then we apply Lemma A.1 to upper bound*

$$\lambda_1(\Sigma_0^{-1}\tilde{\Sigma}_{s,t}\tilde{\Sigma}_{s,t}\Sigma_0^{-1}) = \lambda_1(\Sigma_0^{-1}\tilde{\Sigma}_{s,t}\tilde{\Sigma}_{s,t}\Sigma_0^{-1}) \le \lambda_1\big[\big((I + \Sigma_0\tilde{\Sigma}_{s,t})(I + \tilde{\Sigma}_{s,t}\Sigma_0)\big)^{-1}\big] \le \kappa(\Sigma_0),$$

*resulting in a smaller multiplicative factor than that in [17].* $\qquad \square$

**(2) Bounding $\sum_{t \ge 1} \sum_{s \in \mathcal{S}_t} A_{s,t}^\top \tilde{\Sigma}_{s,t} \Sigma_0^{-1} \bar{\Sigma}_t \Sigma_0^{-1} \tilde{\Sigma}_{s,t} A_{s,t}$.** First recall that

$$\bar{\mu}_t = \bar{\Sigma}_t\big(\Sigma_q^{-1}\mu_q + \sum_{s \in [m]} B_{s,t} - G_{s,t}(\Sigma_0^{-1} + G_{s,t})^{-1}B_{s,t}\big) = \bar{\Sigma}_t\big(\Sigma_q^{-1}\mu_q + \sum_{s \in [m]}(\Sigma_0 + G_{s,t}^{-1})^{-1}G_{s,t}^{-1}B_{s,t}\big),$$

$$\bar{\Sigma}_t^{-1} = \Sigma_q^{-1} + \sum_{s \in [m]} G_{s,t} - G_{s,t}(\Sigma_0^{-1} + G_{s,t})^{-1}G_{s,t} = \Sigma_q^{-1} + \sum_{s \in [m]}(\Sigma_0 + G_{s,t}^{-1})^{-1}.$$

Therefore $\bar{\Sigma}_t \le \Sigma_q$. Then applying Lemma A.1 and Weyl's inequality, we have

$$A_{s,t}^\top \tilde{\Sigma}_{s,t}\Sigma_0^{-1}\bar{\Sigma}_t\Sigma_0^{-1}\tilde{\Sigma}_{s,t}A_{s,t} \le B^2\lambda_1(\tilde{\Sigma}_{s,t}\Sigma_0^{-1}\bar{\Sigma}_t\Sigma_0^{-1}\tilde{\Sigma}_{s,t}) \le B^2\lambda_1(\Sigma_0^{-1}\tilde{\Sigma}_{s,t}\tilde{\Sigma}_{s,t}\Sigma_0^{-1})\lambda_1(\bar{\Sigma}_t)$$

$$= B^2\lambda_1\big[\big((I + \Sigma_0\tilde{\Sigma}_{s,t})(I + \tilde{\Sigma}_{s,t}\Sigma_0)\big)^{-1}\big]\lambda_1(\bar{\Sigma}_t) \le B^2\frac{\lambda_1(\Sigma_q)\lambda_1(\Sigma_0)}{\lambda_d(\Sigma_0)} \le B^2\frac{\lambda_1(\Sigma_q)\lambda_1(\Sigma_0)}{\lambda_d(\Sigma_0)} + B^2\lambda_1(\Sigma_0) + \sigma^2.$$

Meanwhile, we estimate the gap between matrix $\bar{\Sigma}_{t+1}^{-1}$ and matrix $\bar{\Sigma}_t^{-1}$ as follow

$$
\begin{aligned}
\bar{\Sigma}_{t+1}^{-1} - \bar{\Sigma}_t^{-1} &= (\Sigma_0 + (G_{s,t} + \sigma^{-2} A_{s,t} A_{s,t}^\top)^{-1})^{-1} - (\Sigma_0 + G_{s,t}^{-1})^{-1} \\
&= \Sigma_0^{-1} - \Sigma_0^{-1} (\tilde{\Sigma}_{s,t}^{-1} + \sigma^{-2} A_{s,t} A_{s,t}^\top)^{-1} \Sigma_0^{-1} - (\Sigma_0^{-1} - \Sigma_0^{-1} \tilde{\Sigma}_{s,t} \Sigma_0^{-1}) \\
&= \Sigma_0^{-1} \big[ \tilde{\Sigma}_{s,t} - (\tilde{\Sigma}_{s,t}^{-1} + \sigma^{-2} A_{s,t} A_{s,t}^\top)^{-1} \big] \Sigma_0^{-1} \\
&= \Sigma_0^{-1} \tilde{\Sigma}_{s,t}^{\frac{1}{2}} \big[ I - (I + \sigma^{-2} \tilde{\Sigma}_{s,t}^{\frac{1}{2}} A_{s,t} A_{s,t}^\top \tilde{\Sigma}_{s,t}^{\frac{1}{2}})^{-1} \big] \tilde{\Sigma}_{s,t}^{\frac{1}{2}} \Sigma_0^{-1} \\
&= \Sigma_0^{-1} \tilde{\Sigma}_{s,t}^{\frac{1}{2}} \big[ I - (I - \sigma^{-2} \frac{\tilde{\Sigma}_{s,t}^{\frac{1}{2}} A_{s,t} A_{s,t}^\top \tilde{\Sigma}_{s,t}^{\frac{1}{2}}}{1 + \sigma^{-2} A_{s,t}^\top \tilde{\Sigma}_{s,t} A_{s,t}}) \big] \tilde{\Sigma}_{s,t}^{\frac{1}{2}} \Sigma_0^{-1} \\
&= \frac{\Sigma_0^{-1} \tilde{\Sigma}_{s,t} A_{s,t} A_{s,t}^\top \tilde{\Sigma}_{s,t} \Sigma_0^{-1}}{\sigma^2 + A_{s,t}^\top \tilde{\Sigma}_{s,t} A_{s,t}} \\
&\geq \frac{\Sigma_0^{-1} \tilde{\Sigma}_{s,t} A_{s,t} A_{s,t}^\top \tilde{\Sigma}_{s,t} \Sigma_0^{-1}}{\sigma^2 + B^2 \lambda_1(\Sigma_0) + B^2 \lambda_1(\Sigma_q) \lambda_1(\Sigma_0)/\lambda_d(\Sigma_0)} ,
\end{aligned}
\tag{5}
$$

where the second equality holds due to the Woodbury matrix identity, and the fifth equality holds due to the Sherman-Morrison formula. Then analogous to the proof for **(1) Bounding** $\sum_{t\geq 1} \sum_{s\in\mathcal{S}_t} \|A_{s,t}\|_{\tilde{\Sigma}_{s,t}}^2$, recall $c_2 = \big[ \sigma^2 + B^2 \lambda_1(\Sigma_0) + B^2 \lambda_1(\Sigma_q) \kappa(\Sigma_0) \big]$ we have

$$
\begin{aligned}
&\sum_{t\geq 1} \sum_{s\in\mathcal{S}_t} A_{s,t}^\top \tilde{\Sigma}_{s,t} \Sigma_0^{-1} \bar{\Sigma}_t \Sigma_0^{-1} \tilde{\Sigma}_{s,t} A_{s,t} \\
\leq & 2c_2 \sum_{t\geq 1} \sum_{s\in\mathcal{S}_t} \log \big( 1 + \frac{A_{s,t}^\top \tilde{\Sigma}_{s,t} \Sigma_0^{-1} \bar{\Sigma}_t \Sigma_0^{-1} \tilde{\Sigma}_{s,t} A_{s,t}}{\sigma^2 + B^2 \lambda_1(\Sigma_0) + B^2 \lambda_1(\Sigma_q) \kappa(\Sigma_0)} \big) \\
= & 2c_2 \sum_{t\geq 1} \sum_{s\in\mathcal{S}_t} \log\det \big( I + \frac{\bar{\Sigma}_t^{\frac{1}{2}} \Sigma_0^{-1} \tilde{\Sigma}_{s,t} A_{s,t} A_{s,t}^\top \tilde{\Sigma}_{s,t} \Sigma_0^{-1} \bar{\Sigma}_t^{\frac{1}{2}}}{\sigma^2 + B^2 \lambda_1(\Sigma_0) + B^2 \lambda_1(\Sigma_q) \kappa(\Sigma_0)} \big) \\
= & 2c_2 \sum_{t\geq 1} \sum_{s\in\mathcal{S}_t} \big[ \log\det \big( \bar{\Sigma}_t^{-1} + \frac{\Sigma_0^{-1} \tilde{\Sigma}_{s,t} A_{s,t} A_{s,t}^\top \tilde{\Sigma}_{s,t} \Sigma_0^{-1}}{\sigma^2 + B^2 \lambda_1(\Sigma_0) + B^2 \lambda_1(\Sigma_q) \kappa(\Sigma_0)} \big) - \log\det \big( \bar{\Sigma}_t^{-1} \big) \big] \\
\leq & 2c_2 \sum_{t\geq 1} \sum_{s\in\mathcal{S}_t} \big[ \log\det \big( \bar{\Sigma}_{t+1}^{-1} \big) - \log\det \big( \bar{\Sigma}_t^{-1} \big) \big] \\
\leq & 2c_2 \big[ \log\det \big( \bar{\Sigma}_{mn+1}^{-1} \big) - \log\det \big( \bar{\Sigma}_1^{-1} \big) \big] \\
= & 2c_2 \log\det \big( I + \sum_{s\in[m]} \Sigma_q^{\frac{1}{2}} (\Sigma_0 + G_{s,mn+1}^{-1})^{-1} \Sigma_q^{\frac{1}{2}} \big) \\
\leq & 2dc_2 \log \frac{\mathrm{Tr}\,(I) + \mathrm{Tr}\,\big( \sum_{s\in[m]} \Sigma_q^{\frac{1}{2}} (\Sigma_0 + G_{s,mn+1}^{-1})^{-1} \Sigma_q^{\frac{1}{2}} \big)}{d} \\
\leq & 2dc_2 \log \big( 1 + \frac{m\,\mathrm{Tr}\,(\Sigma_0^{-1} \Sigma_q)}{d} \big),
\end{aligned}
$$

where the second inequality holds due to Eq. (5). Combining **(1)** and **(2)** finishes the whole proof. $\square$

**Remark A.2** *Actually, we can replace the term $\mathrm{Tr}(\Sigma_0^{-1} \Sigma_q)$ in the above regret bound with $O(n\lambda_1(\Sigma_q))$, at the cost of a slightly larger regret upper bound, by bounding $(\Sigma_0 + G_{s,mn+1}^{-1})^{-1} \leq G_{s,mn+1}$ in the last but one step in the above (2), instead of bounding $(\Sigma_0 + G_{s,mn+1}^{-1})^{-1} \leq \Sigma_0^{-1}$. Specifically, we have the following estimation:*

$$
\begin{aligned}
&\log \big( \frac{\mathrm{Tr}(I + \sum_{s\in[m]} \Sigma_q^{\frac{1}{2}} (\Sigma_0 + G_{s,mn+1}^{-1})^{-1} \Sigma_q^{\frac{1}{2}})}{d} \big) \\
\leq & \log \big( \frac{\mathrm{Tr}(I + \sum_{s\in[m]} \Sigma_q^{\frac{1}{2}} G_{s,mn+1} \Sigma_q^{\frac{1}{2}})}{d} \big)
\end{aligned}
$$

$$= \log \Big( 1 + \frac{\mathrm{Tr}(\sigma^{-2} \sum_{s \in [m]} \sum_{\ell < mn+1} \mathbf{1}[s \in \mathcal{S}_\ell] A_{s,\ell}^\top \Sigma_q A_{s,\ell})}{d} \Big)$$

$$\leq \log \Big( 1 + \frac{\sigma^{-2} mn \lambda_1(\Sigma_q)}{d} \Big),$$

*which is $O(\log(mn))$, slightly larger than the regret bound of $O(\log(m))$ in our Proposition A.1.*

Then we can begin proving our first Bayes regret bound for HierTS in the multi-task Gaussian linear bandit setting. We first give a lemma as follow, which is useful to prove our multi-task Bayes regret bound in the sequential setting.

**Lemma A.2** *(Proposition 2 in [21]) Let $X_1$ and $X_2$ be arbitrary i.i.d. $\mathbb{R}^m$ valued random variables and $f_1, f_2$ measurable maps such that $f_1, f_2 : \mathbb{R}^m \to \mathbb{R}^d$ with $\mathbb{E}\|f_1(X_1)\|_2^2$, $\mathbb{E}\|f_2(X_1)\|_2^2 < \infty$, then $|\mathbb{E}[f_1(X_1)^\top f_2(X_1)]| \leq \sqrt{d\mathbb{E}[(f_1(X_1)^\top f_2(X_2))^2]}$.*

**Theorem A.1** *(Theorem 5.1 in the main text). Let $|\mathcal{S}_t| = 1$ for all rounds $t \geq 1$. Then in the multi-task Gaussian linear bandit setting, the Bayes regret upper bound of HierTS is as follow:*

$$\mathcal{BR}(m,n) \leq \sqrt{mnd}\sqrt{2mdc_1 \log\Big(1 + \frac{n}{d}\Big) + 2dc_2 \log\Big(1 + \frac{m\,\mathrm{Tr}(\Sigma_q \Sigma_0^{-1})}{d}\Big)}$$

*Proof.* Recall that $H_t = (H_{s,\ell})_{\ell < t, s \in \mathcal{S}_\ell}$ is the history up to round $t$, then

$$\mathcal{BR}(m,n) = \mathbb{E}\Big[ \sum_{t \geq 1} \sum_{s \in \mathcal{S}_t} \mathbb{E}[\theta_{s,*}^\top A_{s,*} - \theta_{s,*}^\top A_{s,t}|H_t] \Big]$$

$$= \mathbb{E}\Big[ \sum_{t \geq 1} \sum_{s \in \mathcal{S}_t} \mathbb{E}\big[\theta_{s,*}^\top A_{s,*} - \mathbb{E}[\theta_{s,*}|H_t]^\top \mathbb{E}[A_{s,t}|H_t]\big|H_t\big] \Big]$$

$$= \mathbb{E}\Big[ \sum_{t \geq 1} \sum_{s \in \mathcal{S}_t} \mathbb{E}\big[(\theta_{s,*} - \hat{\mu}_{s,t})^\top A_{s,*}\big|H_t\big] \Big]$$

$$\leq \mathbb{E}\Big[ \sum_{t \geq 1} \sum_{s \in \mathcal{S}_t} \sqrt{d\mathbb{E}\big[\big((\theta_{s,*} - \hat{\mu}_{s,t})^\top A_{s,t}\big)^2\big|H_t\big]} \Big]$$

$$= \mathbb{E}\Big[ \sum_{t \geq 1} \sum_{s \in \mathcal{S}_t} \sqrt{d\mathbb{E}\big[A_{s,t}^\top(\theta_{s,*} - \hat{\mu}_{s,t})(\theta_{s,*} - \hat{\mu}_{s,t})^\top A_{s,t}\big|H_t\big]} \Big]$$

$$= \mathbb{E}\Big[ \sum_{t \geq 1} \sum_{s \in \mathcal{S}_t} \sqrt{d\mathbb{E}\big[A_{s,t}^\top \hat{\Sigma}_{s,t} A_{s,t}\big]} \Big]$$

$$\leq \sqrt{mnd}\sqrt{\mathbb{E} \sum_{t \geq 1} \sum_{s \in \mathcal{S}_t} \|A_{s,t}\|_{\hat{\Sigma}_{s,t}}^2},$$

where both the second and the fifth equality hold due to the independence between $A_{s,t}$ and $\theta_{s,*}$ conditioned on $H_t$; the first inequality holds by applying Lemma A.2 with functions $f_1(y_1, y_2) = y_1$, $f_2(y_1, y_2) = y_2$ for any $y_1, y_2 \in \mathbb{R}^d$, and the random variable $X_1 = (\theta_{s,*} - \hat{\mu}_{s,t}, A_{s,*})|H_t$, $X_2$ (with the second element as $A_{s,t}$) is the i.i.d. copy of $X_1$; the second inequality holds due to Jensen's inequality. Plugging the upper bound over $\mathbb{E} \sum_{t \geq 1} \sum_{s \in \mathcal{S}_t} \|A_{s,t}\|_{\hat{\Sigma}_{s,t}}^2$ in Proposition A.1 into the above result obtains the Bayes regret bound for HierTS. $\square$

# B Proofs for Regret Bound of HierBayesUCB in the Sequential Bandit Setting

**Lemma B.1** *Let $\theta \mid \mu \sim \mathcal{N}(\mu, \Sigma_0)$ and $H = (x_t, Y_t)_{t=1}^n$ be $n$ observations generated as $Y_t \mid \theta, x_t \sim \mathcal{N}(x_t^\top \theta, \sigma^2)$. Let $\mathbb{P}(\mu \mid H) = \mathcal{N}(\mu; \bar{\mu}, \bar{\Sigma})$, and $G = \sigma^{-2} \sum_{t=1}^n x_t x_t^\top$. Then*

$$\mathbb{E}[\theta \mid H] = (\Sigma_0^{-1} + G)^{-1}(\Sigma_0^{-1}\bar{\mu} + B), \quad \mathrm{cov}[\theta \mid H] = (\Sigma_0^{-1} + G)^{-1} + (\Sigma_0^{-1} + G)^{-1}\Sigma_0^{-1}\bar{\Sigma}\Sigma_0^{-1}(\Sigma_0^{-1} + G)^{-1}.$$

**Proof.** By definition, we have $\text{cov}[\theta \mid \mu, H] = (\Sigma_0^{-1} + G)^{-1}$, $\mathbb{E}[\theta \mid \mu, H] = \text{cov}[\theta \mid \mu, H](\Sigma_0^{-1}\mu + B)$ where $B = \sigma^{-2}\sum_{t=1}^n x_t Y_t$. Hence $\text{cov}[\theta \mid \mu, H]$ does not depend on $\mu$. Then we have

$$\mathbb{E}[\theta \mid H] = \mathbb{E}[\mathbb{E}[\theta \mid \mu, H] \mid H] = \text{cov}[\theta \mid \mu, H](\Sigma_0^{-1}\mathbb{E}[\mu \mid H] + B) = (\Sigma_0^{-1} + G)^{-1}(\Sigma_0^{-1}\bar{\mu} + B).$$

On the other hand, because $\text{cov}[\theta \mid \mu, H]$ does not depend on $\mu$, $\mathbb{E}[\text{cov}[\theta \mid \mu, H] \mid H] = \text{cov}[\theta \mid \mu, H]$. In addition, since $B$ is a constant conditioned on $H$, then according to [17, Lemma 2], we have the following result:

$$\text{cov}[\mathbb{E}[\theta \mid \mu, H] \mid H] = \text{cov}[\text{cov}[\theta \mid \mu, H]\Sigma_0^{-1}\mu \mid H] = (\Sigma_0^{-1} + G)^{-1}\Sigma_0^{-1}\bar{\Sigma}\Sigma_0^{-1}(\Sigma_0^{-1} + G)^{-1}. \quad \square$$

**Theorem B.1** (*Theorem 5.2 in the main text*). *Suppose the action set $\mathcal{A}$ is finite with $|\mathcal{A}| < \infty$. Let $|\mathcal{S}_t| = 1$ for all rounds $t \geq 1$. Then in the multi-task Gaussian linear bandit setting, the Bayes regret upper bound of Hierarchical BayesUCB is as follow:*

$$\mathcal{BR}(m,n) \leq mn\epsilon + 4B\sqrt{\lambda_1(\Sigma_0 + \Sigma_q)}\Big(\sqrt{d + \sqrt{8d\ln\frac{1}{\zeta}}} + \|\mu_q\|_{\hat{\Sigma}_{s,1}^{-1}}\Big)mn|\mathcal{A}|\delta$$

$$+ \mathbb{E}\Big[\frac{8\log\frac{1}{\delta}}{\Delta_{\min}^\epsilon}\Big]\Big\{2mdc_1\log\big(1 + \frac{n}{d}\big) + 2dc_2\log\big(1 + \frac{m\,\text{Tr}\,(\Sigma_0^{-1}\Sigma_q)}{d}\big)\Big\}.$$

*In Theorem 5.2 in the main text, we replace $\sqrt{d + \sqrt{8d\ln\frac{1}{\zeta}}}$ in the right hand side of the above inequality with $\sqrt{d}$ for ease of exposition.*

**Proof.** Define $\Delta_{s,t} = \theta_{s,*}^\top A_{s,*} - \theta_{s,*}^\top A_{s,t}$, the event $E_{s,t} = \{\forall a \in \mathcal{A} : |a^\top(\theta_{s,*} - \hat{\mu}_{s,t})| \leq \sqrt{2\log\frac{1}{\delta}}\|a\|_{\hat{\Sigma}_{s,t}}\}$, and $C_{t,s,a} = \sqrt{2\log\frac{1}{\delta}}\|a\|_{\hat{\Sigma}_{s,t}}$. Then we can rewrite the multi-task Bayes regret $\mathcal{BR}(m,n)$ as the following equivalent form:

$$\mathbb{E}\Big[\sum_{t\geq 1}\sum_{s\in\mathcal{S}_t}\Delta_{s,t}\Big] = \sum_{t\geq 1}\sum_{s\in\mathcal{S}_t}\mathbb{E}\big[\Delta_{s,t}\mathbf{1}\{\Delta_{s,t} \geq \epsilon, E_{s,t}\}\big] + \sum_{t\geq 1}\sum_{s\in\mathcal{S}_t}\mathbb{E}\big[\Delta_{s,t}\mathbf{1}\{\Delta_{s,t} < \epsilon, E_{s,t}\}\big] + \sum_{t\geq 1}\sum_{s\in\mathcal{S}_t}\mathbb{E}\big[\Delta_{s,t}\mathbf{1}\{\bar{E}_{s,t}\}\big].$$

Then we will bound the three terms in the RHS of the above equality respectively.

**(1) Bounding** $\sum_{t\geq 1}\sum_{s\in\mathcal{S}_t}\mathbb{E}\big[\Delta_{s,t}\mathbf{1}\{\Delta_{s,t} \geq \epsilon, E_{s,t}\}\big]$. Recall that $U_{t,s,a} = a^\top\hat{\mu}_{s,t} + \sqrt{2\log\frac{1}{\delta}}\|a\|_{\hat{\Sigma}_{s,t}}$, then we have

$$\sum_{t\geq 1}\sum_{s\in\mathcal{S}_t}\mathbb{E}\big[\Delta_{s,t}\mathbf{1}\{\Delta_{s,t} \geq \epsilon, E_{s,t}\}\big]$$

$$= \sum_{t\geq 1}\sum_{s\in\mathcal{S}_t}\mathbb{E}\Big[\mathbb{E}\Big[\frac{\big(\theta_{s,*}^\top A_{s,*} - \theta_{s,*}^\top A_{s,t}\big)^2}{\Delta_{s,t}}\mathbf{1}\{\Delta_{s,t} \geq \epsilon, E_{s,t}\}\Big| H_t\Big]\Big]$$

$$\leq \sum_{t\geq 1}\sum_{s\in\mathcal{S}_t}\mathbb{E}\Big[\mathbb{E}\Big[\frac{\big(\theta_{s,*}^\top A_{s,*} - U_{t,s,A_{s,*}} + U_{t,s,A_{s,t}} - \theta_{s,*}^\top A_{s,t}\big)^2}{\Delta_{s,t}}\mathbf{1}\{\Delta_{s,t} \geq \epsilon, E_{s,t}\}\Big| H_t\Big]\Big]$$

$$\leq \sum_{t\geq 1}\sum_{s\in\mathcal{S}_t}\mathbb{E}\Big[\mathbb{E}\Big[\frac{\big(U_{t,s,A_{s,t}} - \theta_{s,*}^\top A_{s,t}\big)^2}{\Delta_{s,t}}\mathbf{1}\{\Delta_{s,t} \geq \epsilon, E_{s,t}\}\Big| H_t\Big]\Big]$$

$$\leq \sum_{t\geq 1}\sum_{s\in\mathcal{S}_t}\mathbb{E}\Big[\frac{4C_{t,s,A_{s,t}}^2}{\Delta_{\min}^\epsilon}\Big]$$

$$= \sum_{t\geq 1}\sum_{s\in\mathcal{S}_t}\mathbb{E}\Big[\frac{(8\log\frac{1}{\delta})\|A_{s,t}\|_{\hat{\Sigma}_{s,t}}^2}{\Delta_{\min}^\epsilon}\Big]$$

$$\leq \mathbb{E}\Big(\frac{8\log\frac{1}{\delta}}{\Delta_{\min}^\epsilon}\Big)\Big\{2mdc_1\log\big(1 + \frac{n}{d}\big) + 2dc_2\log\big(1 + \frac{m\,\text{Tr}\,(\Sigma_0^{-1}\Sigma_q)}{d}\big)\Big\}.$$

where the first inequality holds due to the fact that $U_{t,s,A_{s,t}} \geq U_{t,s,A_{s,*}}$ in the BayesUCB algorithm; the second inequality holds because when event $E_{s,t}$ occurs, $\theta_{s,*}^\top A_{s,*} \leq U_{t,s,A_{s,*}}$ the third inequality

holds due to the definition of $U_{t,s,a}$; and the last inequality due to the result in Proposition A.1.

**(2) Bounding** $\sum_{t\geq 1}\sum_{s\in\mathcal{S}_t}\mathbb{E}\big[\Delta_{s,t}\mathbf{1}\{\Delta_{s,t} < \epsilon, E_{s,t}\}\big].$ We trivially have $\sum_{t\geq 1}\sum_{s\in\mathcal{S}_t}\mathbb{E}\big[\Delta_{s,t}\mathbf{1}\{\Delta_{s,t}<\epsilon,E_{s,t}\}\big] \leq mn\epsilon.$

**(3) Bounding** $\sum_{t\geq 1}\sum_{s\in\mathcal{S}_t}\mathbb{E}\big[\Delta_{s,t}\mathbf{1}\{\bar{E}_{s,t}\}\big].$

First we give an upper bound of $\Delta_{s,t} = \theta_{s,*}^\top(A_{s,*} - A_{s,t})$. Using Schwartz's inequality, we have

$$\theta_{s,*}^\top(A_{s,*} - A_{s,t}) \leq \|\theta_{s,*}\|_{\hat{\Sigma}_{s,1}^{-1}}\|A_{s,*} - A_{s,t}\|_{\hat{\Sigma}_{s,1}}$$

$$\leq 2B\sqrt{\lambda_1(\hat{\Sigma}_{s,1})}\|\theta_{s,*}\|_{\hat{\Sigma}_{s,1}^{-1}} \leq 2B\sqrt{\lambda_1(\hat{\Sigma}_{s,1})}\Big(\|\theta_{s,*} - \mu_q\|_{\hat{\Sigma}_{s,1}^{-1}} + \|\mu_q\|_{\hat{\Sigma}_{s,1}^{-1}}\Big).$$

Besides, we also have $\theta_{s,*} - \mu_q = \theta_{s,*} - \mu_* + \mu_* - \mu_q \sim \mathcal{N}(0, \Sigma_0 + \Sigma_q) = \mathcal{N}(0, \hat{\Sigma}_{s,1})$, then $\mathbb{E}\big[\|\theta_{s,*} - \mu_q\|_{\hat{\Sigma}_{s,1}^{-1}}\big] \leq \sqrt{\mathbb{E}\|\hat{\Sigma}_{s,1}^{-\frac{1}{2}}(\theta_{s,*} - \mu_q)\|_2^2} = \sqrt{d}$. According to [35, Exp 2.11], we have with probability $1 - \zeta$, $\|\theta_{s,*} - \mu_q\|_{\hat{\Sigma}_{s,1}^{-1}} \leq \sqrt{d + \sqrt{8d\ln\frac{1}{\zeta}}}$. Therefore, with probability $1 - \zeta$ over the draw of $\{\theta_{s,*}\}_{s\in[m]}$, we have

$$\sum_{t\geq 1}\sum_{s\in\mathcal{S}_t}\mathbb{E}\big[\Delta_{s,t}\mathbf{1}\{\bar{E}_{s,t}\}\big]$$

$$=\sum_{t\geq 1}\sum_{s\in\mathcal{S}_t}\mathbb{E}\big[\mathbb{E}\big[\Delta_{s,t}\mathbf{1}\{\bar{E}_{s,t}\}\big|H_t\big]\big]$$

$$\leq 2B\sqrt{\lambda_1(\Sigma_0 + \Sigma_q)}\Big(\sqrt{d + \sqrt{8d\ln\frac{1}{\zeta}}} + \|\mu_q\|_{\hat{\Sigma}_{s,1}^{-1}}\Big)\sum_{t\geq 1}\sum_{s\in\mathcal{S}_t}\mathbb{E}\big[\mathbf{1}\{\bar{E}_{s,t}\}\big|H_t\big]$$

$$=2B\sqrt{\lambda_1(\Sigma_0 + \Sigma_q)}\Big(\sqrt{d + \sqrt{8d\ln\frac{1}{\zeta}}} + \|\mu_q\|_{\hat{\Sigma}_{s,1}^{-1}}\Big)\sum_{t\geq 1}\sum_{s\in\mathcal{S}_t}\mathbb{P}\big(\bar{E}_{s,t}\big|H_t\big)$$

$$\leq 4B\sqrt{\lambda_1(\Sigma_0 + \Sigma_q)}\Big(\sqrt{d + \sqrt{8d\ln\frac{1}{\zeta}}} + \|\mu_q\|_{\hat{\Sigma}_{s,1}^{-1}}\Big)mn|\mathcal{A}|\delta,$$

where the last inequality holds because $\mathbb{P}\big(\bar{E}_{s,t}\big|H_t\big) \leq 2\delta$. Combining **(1)**, **(2)** and **(3)**, we achieve the final Bayes regret bound for any $\delta \in (0,1), \epsilon > 0, \zeta \in (0,1)$:

$$\mathcal{BR}(m,n) \leq mn\epsilon + 4B\sqrt{\lambda_1(\Sigma_0 + \Sigma_q)}\Big(\sqrt{d + \sqrt{8d\ln\frac{1}{\zeta}}} + \|\mu_q\|_{\hat{\Sigma}_{s,1}^{-1}}\Big)mn|\mathcal{A}|\delta$$

$$+ \mathbb{E}\Big[\frac{8\log\frac{1}{\delta}}{\Delta_{\min}^\epsilon}\Big]\Big\{2mdc_1\log\big(1 + \frac{n}{d}\big) + 2dc_2\log\big(1 + \frac{m\,\text{Tr}\,(\Sigma_0^{-1}\Sigma_q)}{d}\big)\Big\}. \qquad \square$$

# C Proofs for Regret Bounds of HierTS and HierBayesUCB in the Concurrent Bandit Setting

Let $\mathcal{C}_t = \{s \in \mathcal{S}_t : \lambda_d(G_{s,t}) \geq \beta/\sigma^2\}$ be the set of sufficiently-explored tasks at round $t$. We first give the following proposition to bound the posterior variance $\mathbb{E}\sum_{t\geq 1}\sum_{s\in\mathcal{S}_t}\mathbf{1}\{s \in \mathcal{C}_t\}\|A_{s,t}\|_{\hat{\Sigma}_{s,t}}^2$ in the concurrent setting. Analogous to the proof for Proposition A.1, we choose to give the worst-case upper bound on $\sum_{t\geq 1}\sum_{s\in\mathcal{S}_t}\mathbf{1}\{s \in \mathcal{C}_t\}\|A_{s,t}\|_{\hat{\Sigma}_{s,t}}^2$ as follow.

**Proposition C.1** *Let* $c_1 = \sigma^2 + B^2\lambda_1(\Sigma_0)$, $c_2 = \sigma^2 + B^2\lambda_1(\Sigma_0) + B^2\lambda_1(\Sigma_q)\kappa(\Sigma_0)$, $c_3 = 1 + B^2\sigma^{-2}\kappa(\Sigma_0)\big[\lambda_1(\Sigma_0) + \sigma^2/\beta\big]$, *then we have*

$$\sum_{t\geq 1}\sum_{s\in\mathcal{S}_t}\mathbf{1}\{s \in \mathcal{C}_t\}\|A_{s,t}\|_{\hat{\Sigma}_{s,t}}^2 \leq 2mdc_1\log\big(1 + \frac{n}{d}\big) + 2dc_2c_3\log\big(1 + \frac{m\,\text{Tr}\,(\Sigma_0^{-1}\Sigma_q)}{d}\big).$$

**Proof.** Note that we have

$$\sum_{t\geq 1}\sum_{s\in\mathcal{S}_t}\mathbf{1}\{s\in\mathcal{C}_t\}\|A_{s,t}\|^2_{\tilde{\Sigma}_{s,t}} = \sum_{t\geq 1}\sum_{s\in\mathcal{S}_t}\mathbf{1}\{s\in\mathcal{C}_t\}A_{s,t}^\top\big(\tilde{\Sigma}_{s,t} + \tilde{\Sigma}_{s,t}\Sigma_0^{-1}\bar{\Sigma}_t\Sigma_0^{-1}\tilde{\Sigma}_{s,t}\big)A_{s,t}.$$

On event $\{s\in\mathcal{C}_t\}$, the modified HierTS samples from the posterior and actually behaves the same as the original HierTS algorithm in Algorithm 1. Then we bound the two terms in the right hand side of the above equality respectively .

**(1) Bounding** $\sum_{t\geq 1}\sum_{s\in\mathcal{S}_t}\mathbf{1}\{s\in\mathcal{C}_t\}A_{s,t}^\top\tilde{\Sigma}_{s,t}A_{s,t}$

Similar to the proof for Theorem A.1, we have $A_{s,t}^\top\tilde{\Sigma}_{s,t}A_{s,t} \leq B^2\lambda_1(\Sigma_0) + \sigma^2$ and $\tilde{X}_{s,t} \triangleq (\Sigma_0^{-1} + \frac{1}{B^2\lambda_1(\Sigma_0)+\sigma^2}\sum_{\ell<t}\mathbf{1}\{s\in\mathcal{S}_t\}A_{s,\ell}A_{s,\ell}^\top)^{-1} \geq \tilde{\Sigma}_{s,t}$. Then we can analogously obtain

$$\sum_{t\geq 1}\sum_{s\in\mathcal{S}_t}\mathbf{1}\{s\in\mathcal{C}_t\}A_{s,t}^\top\tilde{\Sigma}_{s,t}A_{s,t}$$

$$\leq 2\big[\sigma^2 + B^2\lambda_1(\Sigma_0)\big]\sum_{t\geq 1}\sum_{s\in\mathcal{S}_t}\mathbf{1}\{s\in\mathcal{C}_t\}\log\big(1 + \frac{A_{s,t}^\top\tilde{\Sigma}_{s,t}A_{s,t}}{\sigma^2 + B^2\lambda_1(\Sigma_0)}\big)$$

$$= 2\big[\sigma^2 + B^2\lambda_1(\Sigma_0)\big]\sum_{t\geq 1}\sum_{s\in\mathcal{S}_t}\mathbf{1}\{s\in\mathcal{C}_t\}\big[\log\det\big(\tilde{X}_{s,t}^{-1} + \frac{A_{s,t}A_{s,t}^\top}{\sigma^2 + B^2\lambda_1(\Sigma_0)}\big) - \log\det\tilde{X}_{s,t}^{-1}\big]$$

$$\leq 2\big[\sigma^2 + B^2\lambda_1(\Sigma_0)\big]\sum_{t\geq 1}\sum_{s=1}^{m}\mathbf{1}\{s\in\mathcal{C}_t\}\big[\log\det\big(\tilde{X}_{s,t}^{-1} + \frac{A_{s,t}A_{s,t}^\top}{\sigma^2 + B^2\lambda_1(\Sigma_0)}\big) - \log\det\tilde{X}_{s,t}^{-1}\big]$$

$$\leq 2\big[\sigma^2 + B^2\lambda_1(\Sigma_0)\big]\sum_{s=1}^{m}\sum_{t\geq 1}\mathbf{1}\{s\in\mathcal{S}_t\}\big[\log\det\big(\tilde{X}_{s,t}^{-1} + \frac{A_{s,t}A_{s,t}^\top}{\sigma^2 + B^2\lambda_1(\Sigma_0)}\big) - \log\det\tilde{X}_{s,t}^{-1}\big]$$

$$= 2\big[\sigma^2 + B^2\lambda_1(\Sigma_0)\big]\sum_{s=1}^{m}\big[\log\det\big(\tilde{X}_{s,mn+1}^{-1}\big) - \log\det\tilde{X}_{s,1}^{-1}\big]$$

$$\leq 2d\big[\sigma^2 + B^2\lambda_1(\Sigma_0)\big]\sum_{s=1}^{m}\log\big(1 + \frac{\sum_{t\geq 1}\mathbf{1}\{s\in\mathcal{S}_t\}A_{s,t}^\top\Sigma_0 A_{s,t}}{d(\sigma^2 + B^2\lambda_1(\Sigma_0))}\big)$$

$$\leq 2md\big[\sigma^2 + B^2\lambda_1(\Sigma_0)\big]\log\big(1 + \frac{n}{d}\big) = 2mdc_1\log\big(1 + \frac{n}{d}\big),$$

where the second inequality holds due to the fact that, if square matrix $A \geq B \geq 0$, then $\det(A) \geq \det(B)$, and $|\mathcal{S}_t| \leq m$; the last inequality holds because the agent interact with each task at most $n$ times.

**(2) Bounding** $\sum_{t\geq 1}\sum_{s\in\mathcal{S}_t}\mathbf{1}\{s\in\mathcal{C}_t\}A_{s,t}^\top\big(\tilde{\Sigma}_{s,t}\Sigma_0^{-1}\bar{\Sigma}_t\Sigma_0^{-1}\tilde{\Sigma}_{s,t}\big)A_{s,t}$

**Analysis.** The real difference of the proof for the concurrent regret from the sequential regret lies in bounding $\sum_{t\geq 1}\sum_{s\in\mathcal{S}_t}\mathbf{1}\{s\in\mathcal{C}_t\}A_{s,t}^\top\big(\tilde{\Sigma}_{s,t}\Sigma_0^{-1}\bar{\Sigma}_t\Sigma_0^{-1}\tilde{\Sigma}_{s,t}\big)A_{s,t}$, because $|\mathcal{S}_t| \geq 1$ and the result in Eq. (5) (which only holds for the case $|\mathcal{S}_t| = 1$) does not hold. To tackle this difference, we reduce the concurrent setting to the sequential setting. Let $\mathcal{S}_t = \{I_{tj}\}_{j=1}^{|\mathcal{S}_t|}$ and define $\mathcal{S}_{t,1:i} = \{I_{tj}\}_{j=1}^{i}$. Then at round $t$, let $s = I_{ti}$ and define $\bar{\Sigma}_{s,t}^{-1} \triangleq \Sigma_q^{-1} + \sum_{z\in\mathcal{S}_{t,1:i-1}}(\Sigma_0 + G_{z,t+1}^{-1})^{-1} + \sum_{z\in[m]\backslash\mathcal{S}_{t,1:i-1}}(\Sigma_0 + G_{z,t}^{-1})^{-1}$, we estimate the gap between $\bar{\Sigma}_{s,t}^{-1}$ and $\bar{\Sigma}_t^{-1}$ as follow:

$$\bar{\Sigma}_{s,t}^{-1} - \bar{\Sigma}_t^{-1} = \sum_{z\in\mathcal{S}_{t,1:i-1}}\big[(\Sigma_0 + G_{z,t+1}^{-1})^{-1} - (\Sigma_0 + G_{z,t}^{-1})\big]$$

$$= \sum_{z\in\mathcal{S}_{t,1:i-1}}\big[(\Sigma_0 + (G_{z,t} + \frac{A_{z,t}A_{z,t}^\top}{\sigma^2})^{-1})^{-1} - (\Sigma_0 + G_{z,t}^{-1})\big]$$

$$= \sum_{z\in\mathcal{S}_{t,1:i-1}}\Sigma_0^{-1}\tilde{\Sigma}_{s,t}\frac{A_{z,t}A_{z,t}^\top}{\sigma^2 + A_{z,t}^\top\tilde{\Sigma}_{z,t}A_{z,t}}\tilde{\Sigma}_{s,t}\Sigma_0^{-1}.$$

Thus we can bound $\lambda_1(\bar{\Sigma}_{s,t}^{-1} - \bar{\Sigma}_t^{-1})$ and tackle $\bar{\Sigma}_{s,t}^{-1}$ instead of $\bar{\Sigma}_t^{-1}$ to reduce the concurrent setting to the sequential setting. We first give a useful lemma as follow.

**Lemma C.1** *For any fixed $t \geq 1$ and $i \in [L]$, suppose $\lambda_d(G_{s,t}) \geq \beta/\sigma^2$ and let $s = I_{t,i}$. Then*

$$\lambda_1(\bar{\Sigma}_{s,t}^{-1}\bar{\Sigma}_t) \leq 1 + \frac{B^2\lambda_1(\Sigma_0)}{\sigma^2\lambda_d(\Sigma_0)}\Big[\lambda_1(\Sigma_0) + \frac{\sigma^2}{\beta}\Big].$$

*Proof.* Applying Weyl's inequality, we have

$$\lambda_1\big((\bar{\Sigma}_{s,t}^{-1} - \bar{\Sigma}_t^{-1})\bar{\Sigma}_t + I\big) = \lambda_1\big(\bar{\Sigma}_t^{\frac{1}{2}}(\bar{\Sigma}_{s,t}^{-1} - \bar{\Sigma}_t^{-1})\bar{\Sigma}_t^{\frac{1}{2}} + I\big)$$

$$\leq 1 + \lambda_1(\bar{\Sigma}_{s,t}^{-1} - \bar{\Sigma}_t^{-1})\lambda_1(\bar{\Sigma}_t) = 1 + \frac{\lambda_1(\bar{\Sigma}_{s,t}^{-1} - \bar{\Sigma}_t^{-1})}{\lambda_d(\bar{\Sigma}_t^{-1})}$$

We first lower bound $\lambda_d(\bar{\Sigma}_t^{-1})$. According to Weyl's inequality, we have

$$\lambda_d(\bar{\Sigma}_t^{-1}) \geq \lambda_d(\Sigma_q^{-1}) + \sum_{z \in [m]} \lambda_d\big((\Sigma_0 + G_{z,t}^{-1})^{-1}\big) \geq \lambda_d(\Sigma_q^{-1}) + \sum_{z \in [m]} \frac{1}{\lambda_1(\Sigma_0) + \lambda_1(G_{z,t}^{-1})}$$

$$= \lambda_d(\Sigma_q^{-1}) + \sum_{z \in [m]} \frac{1}{\lambda_1(\Sigma_0) + \frac{1}{\lambda_d(G_{z,t})}} \geq \lambda_d(\Sigma_q^{-1}) + \frac{i-1}{\lambda_1(\Sigma_0) + \sigma^2/\beta},$$

where the last inequality holds because the tasks $\mathcal{S}_{t,1:i-1}$ have been sufficiently explored. On the other hand, using our Lemma A.1 we can bound $\lambda_1(\bar{\Sigma}_{s,t}^{-1} - \bar{\Sigma}_t^{-1})$ as follow

$$\lambda_1(\bar{\Sigma}_{s,t}^{-1} - \bar{\Sigma}_t^{-1}) \leq \sum_{z \in \mathcal{S}_{t,1:i-1}} \lambda_1\Big(\frac{A_{z,t}A_{z,t}^\top}{\sigma^2 + A_{z,t}^\top\tilde{\Sigma}_{z,t}A_{z,t}}\Big)\lambda_1(\Sigma_0^{-1}\tilde{\Sigma}_{s,t}\tilde{\Sigma}_{s,t}\Sigma_0^{-1}) \leq (i-1)\frac{B^2}{\sigma^2}\frac{\lambda_1(\Sigma_0)}{\lambda_d(\Sigma_0)}.$$

Combining the above results, we have

$$\lambda_1(\bar{\Sigma}_{s,t}^{-1}\bar{\Sigma}_t) \leq 1 + \frac{(i-1)\frac{B^2}{\sigma^2}\frac{\lambda_1(\Sigma_0)}{\lambda_d(\Sigma_0)}}{\lambda_d(\Sigma_q^{-1}) + \frac{i-1}{\lambda_1(\Sigma_0)+\sigma^2/\beta}} \leq 1 + \frac{B^2}{\sigma^2}\frac{\lambda_1(\Sigma_0)}{\lambda_d(\Sigma_0)}\Big[\lambda_1(\Sigma_0) + \frac{\sigma^2}{\beta}\Big]. \qquad \square$$

Then, recall $c_3 = 1 + B^2\sigma^{-2}\kappa(\Sigma_0)\big[\lambda_1(\Sigma_0) + \sigma^2/\beta\big]$, we can bound $\sum_{t \geq 1}\sum_{s \in \mathcal{S}_t}\mathbf{1}\{s \in \mathcal{C}_t\}A_{s,t}^\top(\tilde{\Sigma}_{s,t}\Sigma_0^{-1}\bar{\Sigma}_t\Sigma_0^{-1}\tilde{\Sigma}_{s,t})A_{s,t}$ as follows:

$$\sum_{t \geq 1}\sum_{s \in \mathcal{S}_t}\mathbf{1}\{s \in \mathcal{C}_t\}A_{s,t}^\top\big(\tilde{\Sigma}_{s,t}\Sigma_0^{-1}\bar{\Sigma}_t\Sigma_0^{-1}\tilde{\Sigma}_{s,t}\big)A_{s,t}$$

$$= \sum_{t \geq 1}\sum_{s \in \mathcal{S}_t}\mathbf{1}\{s \in \mathcal{C}_t\}A_{s,t}^\top\tilde{\Sigma}_{s,t}\Sigma_0^{-1}\bar{\Sigma}_{s,t}^{-\frac{1}{2}}\big(\bar{\Sigma}_{s,t}^{-\frac{1}{2}}\bar{\Sigma}_t\bar{\Sigma}_{s,t}^{-\frac{1}{2}}\big)\bar{\Sigma}_{s,t}^{\frac{1}{2}}\Sigma_0^{-1}\tilde{\Sigma}_{s,t}A_{s,t}$$

$$\leq \sum_{t \geq 1}\sum_{s \in \mathcal{S}_t}\mathbf{1}\{s \in \mathcal{C}_t\}\lambda_1\big(\bar{\Sigma}_{s,t}^{-\frac{1}{2}}\bar{\Sigma}_t\bar{\Sigma}_{s,t}^{-\frac{1}{2}}\big)A_{s,t}^\top\tilde{\Sigma}_{s,t}\Sigma_0^{-1}\bar{\Sigma}_{s,t}^{\frac{1}{2}}\bar{\Sigma}_{s,t}^{\frac{1}{2}}\Sigma_0^{-1}\tilde{\Sigma}_{s,t}A_{s,t}$$

$$\leq \Big\{1 + \frac{B^2}{\sigma^2}\frac{\lambda_1(\Sigma_0)}{\lambda_d(\Sigma_0)}\Big[\lambda_1(\Sigma_0) + \frac{\sigma^2}{\beta}\Big]\Big\}\mathbb{E}\sum_{t \geq 1}\sum_{s \in \mathcal{S}_t}\mathbf{1}\{s \in \mathcal{C}_t\}A_{s,t}^\top\tilde{\Sigma}_{s,t}\Sigma_0^{-1}\bar{\Sigma}_{s,t}\Sigma_0^{-1}\tilde{\Sigma}_{s,t}A_{s,t}$$

$$\leq 2c_3c_2\big[\log\det\big(\bar{\Sigma}_{mn+1}^{-1}\big) - \log\det\big(\bar{\Sigma}_1^{-1}\big)\big]$$

$$\leq 2dc_3c_2\log\big(1 + \frac{m\operatorname{Tr}\big(\Sigma_0^{-1}\Sigma_q\big)}{d}\big),$$

where the second inequality holds due to Lemma C.1, the third and the fourth inequality hold in the same way as that in the proof of **Bounding (2)** in Proposition A.1. Combining the results in **(1)** and **(2)** finishes the whole proof. $\qquad \square$

**Remark C.1** *In the last step of proof for Lemma C.1, we bound*

$$\lambda_1(\bar{\Sigma}_{s,t}^{-1}\bar{\Sigma}_t) \leq 1 + \frac{(i-1)\frac{B^2}{\sigma^2}\frac{\lambda_1(\Sigma_0)}{\lambda_d(\Sigma_0)}}{\lambda_d(\Sigma_q^{-1}) + \frac{i-1}{\lambda_1(\Sigma_0)+\sigma^2/\beta}} \leq 1 + \frac{(i-1)\frac{B^2}{\sigma^2}\frac{\lambda_1(\Sigma_0)}{\lambda_d(\Sigma_0)}}{\frac{i-1}{\lambda_1(\Sigma_0)+\sigma^2/\beta}}.$$

*Thus our upper bound is independent of the number $L$ of the concurrent tasks. Actually, $\forall i \in [L]$:*

$$\lambda_1(\bar{\Sigma}_{s,t}^{-1}\bar{\Sigma}_t) \leq 1 + \frac{\frac{B^2}{\sigma^2}\frac{\lambda_1(\Sigma_0)}{\lambda_d(\Sigma_0)}}{\frac{\lambda_d(\Sigma_q^{-1})}{(i-1)} + \frac{1}{\lambda_1(\Sigma_0)+\sigma^2/\beta}} \leq 1 + \frac{\frac{B^2}{\sigma^2}\frac{\lambda_1(\Sigma_0)}{\lambda_d(\Sigma_0)}}{\frac{\lambda_d(\Sigma_q^{-1})}{L} + \frac{1}{\lambda_1(\Sigma_0)+\sigma^2/\beta}},$$

the sharper bound $1 + \frac{\frac{B^2}{\sigma^2} \frac{\lambda_1(\Sigma_0)}{\lambda_d(\Sigma_0)}}{\frac{\lambda_d(\Sigma_q^{-1})}{L} + \frac{1}{\lambda_1(\Sigma_0)+\sigma^2/\beta}}$ is L-dependent. If $\frac{\lambda_d(\Sigma_q^{-1})}{L} << \frac{1}{\lambda_1(\Sigma_0)+\sigma^2/\beta}$, the influence of L to the regret may be large; otherwise, the influence of L to the regret may be negligible.

Next, we prove the concurrent regret bound for HierTS and HierBayesUCB.

**Theorem C.1** *(Theorem 5.3 in the main text). Let $|\mathcal{S}_t| \leq L \leq m$ for all rounds $t \geq 1$. Then in the multi-task Gaussian linear bandit setting, the Bayes regret bound of HierTS is as follow:*

$$\mathcal{BR}(m,n) \leq 2Bmd\sqrt{\lambda_1(\Sigma_0 + \Sigma_q)}(\sqrt{d} + \|\mu_q\|_{\hat{\Sigma}_{s,1}^{-1}}) + md\sqrt{2nc_1 \log\left(1 + \frac{n}{d}\right)}$$

$$+ \sqrt{mnd}\sqrt{2dc_2c_3 \log\left(1 + \frac{m \operatorname{Tr}\left(\Sigma_0^{-1}\Sigma_q\right)}{d}\right)}.$$

**Proof.** Recall that $\mathcal{C}_t = \{s \in \mathcal{S}_t : \lambda_d(G_{s,t}) \geq \beta/\sigma^2\}$ is the set of sufficiently-explored tasks at round $t$. Then, due to the modification of HierTS algorithm, we decompose Bayes regret $\mathcal{BR}(m,n)$ into two terms and bound them respectively:

$$\mathcal{BR}(m,n) = \mathbb{E}\sum_{t\geq 1}\sum_{s\in\mathcal{S}_t}\mathbf{1}\{s \notin \mathcal{C}_t\}\theta_{s,*}^\top(A_{s,*} - A_{s,t}) + \mathbb{E}\sum_{t\geq 1}\sum_{s\in\mathcal{S}_t}\mathbf{1}\{s \in \mathcal{C}_t\}\theta_{s,*}^\top(A_{s,*} - A_{s,t})$$

**(1) Bounding** $\mathbb{E}\sum_{t\geq 1}\sum_{s\in\mathcal{S}_t}\mathbf{1}\{s \notin \mathcal{C}_t\}\theta_{s,*}^\top(A_{s,*} - A_{s,t})$. Similar to the proof for Theorem B.1 **(3)**, we have

$$\theta_{s,*}^\top(A_{s,*} - A_{s,t}) \leq \|\theta_{s,*}\|_{\hat{\Sigma}_{s,1}^{-1}}\|A_{s,*} - A_{s,t}\|_{\hat{\Sigma}_{s,1}} \leq 2c\sqrt{\lambda_1(\hat{\Sigma}_{s,1})}\left(\|\theta_{s,*} - \mu_q\|_{\hat{\Sigma}_{s,1}^{-1}} + \|\mu_q\|_{\hat{\Sigma}_{s,1}^{-1}}\right),$$

and $\mathbb{E}\left[\|\theta_{s,*} - \mu_q\|_{\hat{\Sigma}_{s,1}^{-1}}\right] \leq \sqrt{\mathbb{E}\|\hat{\Sigma}_{s,1}^{-\frac{1}{2}}(\theta_{s,*} - \mu_q)\|_2^2} = \sqrt{d}$. Recalling the independence between $\theta_{s,*}$ and actions $A_{s,t}$ yields

$$\mathbb{E}\sum_{t\geq 1}\sum_{s\in\mathcal{S}_t}\mathbf{1}\{s \notin \mathcal{C}_t\}\theta_{s,*}^\top(A_{s,*} - A_{s,t})$$

$$\leq 2B\sqrt{\lambda_1(\Sigma_0 + \Sigma_q)}\mathbb{E}\sum_{t\geq 1}\sum_{s\in\mathcal{S}_t}\mathbf{1}\{s \notin \mathcal{C}_t\}\mathbb{E}\left(\|\theta_{s,*} - \mu_q\|_{\hat{\Sigma}_{s,1}^{-1}} + \|\mu_q\|_{\hat{\Sigma}_{s,1}^{-1}}\right)$$

$$\leq 2B\sqrt{\lambda_1(\Sigma_0 + \Sigma_q)}(\sqrt{d} + \|\mu_q\|_{\hat{\Sigma}_{s,1}^{-1}})md,$$

The last inequality holds because in the modified HierTS, event $\{s \notin \mathcal{C}_t\}$ occurs at most $d$ times for any task $s \in [m]$.

**(2) Bounding** $\mathbb{E}\sum_{t\geq 1}\sum_{s\in\mathcal{S}_t}\mathbf{1}\{s \in \mathcal{C}_t\}\|A_{s,t}\|_{\hat{\Sigma}_{s,t}^{-1}}^2$. It suffices to apply the upper bound in Proposition C.1.

Combining the upper bounds in steps **(1)** and **(2)** obtains the final Bayes regret bound for HierTS in the concurrent setting. $\qquad\square$

**Theorem C.2** *(Logarithmic Regret Bound for HierBayesUCB in the Concurrent Bandit Setting). Suppose the action set $\mathcal{A}$ is finite with $|\mathcal{A}| < \infty$. Let $|\mathcal{S}_t| \leq L \leq m$ for all rounds $t \geq 1$. Then in the multi-task Gaussian linear bandit setting, the Bayes regret bound of HierTS is as follow:*

$$\mathcal{BR}(m,n) \leq mn\epsilon + 4B\sqrt{\lambda_1(\Sigma_0 + \Sigma_q)}\left(\sqrt{d + \sqrt{8d\ln\frac{1}{\zeta}}} + \|\mu_q\|_{\hat{\Sigma}_{s,1}^{-1}}\right)mn|\mathcal{A}|\delta$$

$$+ 2B\sqrt{\lambda_1(\Sigma_0 + \Sigma_q)}(\sqrt{d} + \|\mu_q\|_{\hat{\Sigma}_{s,1}^{-1}})md + \mathbb{E}\left[\frac{8\log\frac{1}{\delta}}{\Delta_{\min}^\epsilon}\right]\left\{2mdc_1 \log\left(1 + \frac{n}{d}\right) + 2dc_3c_2 \log\left(1 + \frac{m \operatorname{Tr}\left(\Sigma_0^{-1}\Sigma_q\right)}{d}\right)\right\}.$$

**Proof.** Similar to the proof of Theorem C.1, we decompose the Bayes regret as $\mathcal{BR}(m,n) = \mathbb{E}\sum_{t\geq 1}\sum_{s\in\mathcal{S}_t}\mathbf{1}\{s \notin \mathcal{C}_t\}\theta_{s,*}^\top(A_{s,*} - A_{s,t}) + \mathbb{E}\sum_{t\geq 1}\sum_{s\in\mathcal{S}_t}\mathbf{1}\{s \in \mathcal{C}_t\}\theta_{s,*}^\top(A_{s,*} - A_{s,t})$. Then

Table 3: Different Bayes regret bounds for multi-task $d$-dimensional linear bandit problem in the concurrent setting. $m$ is the number of tasks, $n$ is the number of iterations per task, $\mathcal{A}$ is the action set. **Bayes Regret Bound =Bound I + Bound II + Negligible Terms**, where **Bound I** is the regret bound for solving $m$ tasks, **Bound II** the regret bound for learning hyper-parameter $\mu_*$.

| Bayes Regret Bound | $|\mathcal{A}|$ | Bound I | Bound II |
|---|---|---|---|
| [17, Theorem 4] | Infinite | $O\big(md\sqrt{n\log\big(\frac{n}{d}\big)}\log(mn)\big)$ | $O\big(d\sqrt{mn\log(m)}\log(mn)\big)$ |
| Our Theorem 5.3 | Infinite | $O\big(md\sqrt{n\log\big(\frac{n}{d}\big)}\big)$ | $O\big(d\sqrt{mn\log\big(\frac{m}{d}\big)}\big)$ |
| Our Theorem C.2 | Finite | $O\big(md\log\big(\frac{n}{d}\big)\log(mn)\big)$ | $O\big(d\log\big(\frac{m}{d}\big)\log(mn)\big)$ |

we bound the first term with the proof for Theorem C.1 **(1)**, bound the second term with the proof for our Theorem B.1. Then with probability $1-\zeta$ over the draw of $\{\theta_{s,*}\}_{s\in[m]}$,

$$\mathcal{BR}(m,n) \leq 2B\sqrt{\lambda_1(\Sigma_0+\Sigma_q)}(\sqrt{d}+\|\mu_q\|_{\hat{\Sigma}_{s,1}^{-1}})md + \mathbb{E}\Big[\frac{8\log\frac{1}{\delta}}{\Delta_{\min}^{\epsilon}}\Big]\sum_{t\geq 1}\sum_{s\in\mathcal{S}_t}\mathbb{E}\big[\mathbf{1}\{s\in\mathcal{C}_t\}\|A_{s,t}\|_{\hat{\Sigma}_{s,t}}^2\big]$$

$$+ mn\epsilon + 4B\sqrt{\lambda_1(\Sigma_0+\Sigma_q)}\Big(\sqrt{d+\sqrt{8d\ln\frac{1}{\zeta}}}+\|\mu_q\|_{\hat{\Sigma}_{s,1}^{-1}}\Big)mn|\mathcal{A}|\delta.$$

Plugging the upper bound on $\sum_{t\geq 1}\sum_{s\in\mathcal{S}_t}\mathbb{E}\big[\mathbf{1}\{s\in\mathcal{C}_t\}\|A_{s,t}\|_{\hat{\Sigma}_{s,t}}^2\big]$ in Proposition C.1 into the right hand side of the above inequality finishes the whole proof. $\qquad\square$

# D  Proofs for Regret Bounds of HierTS and HierBayesUCB in the Semi-Bandit Setting

We also choose to give the worst-case upper bound on $\sum_{t\geq 1}\sum_{s\in\mathcal{S}_t}\sum_{a\in A_{s,t}}\Phi_a^\top\hat{\Sigma}_{s,t}\Phi_a$ as follow.

**Proposition D.1** *Let* $c_1 = \sigma^2 + B^2\lambda_1(\Sigma_0)$, $c_4 = \sigma^2 + B^2 L\lambda_1(\Sigma_0) + B^2\lambda_1(\Sigma_q)\kappa(\Sigma_0)$, *then*

$$\sum_{t\geq 1}\sum_{s\in\mathcal{S}_t}\sum_{a\in A_{s,t}}\Phi_a^\top\hat{\Sigma}_{s,t}\Phi_a \leq 2c_1 m\log\big(1+\frac{nL}{d}\big) + 2c_4 Ld\log(1+\frac{m\,\mathrm{Tr}(\Sigma_0^{-1}\Sigma_q)}{d}).$$

**Proof.** Recall that $G_{s,t} = \sigma^{-2}\sum_{\ell<t}\mathbf{1}\{s\in\mathcal{S}_\ell\}(\sum_{a\in A_{s,t}}\Phi_a\Phi_a^\top)$, $B_{s,t} = \sigma^{-2}\sum_{\ell<t}\mathbf{1}\{s\in\mathcal{S}_\ell\}(\sum_{a\in A_{s,t}}\Phi_a\bar{\mathbf{w}}_s(a))$, $\tilde{\Sigma}_{s,t}^{-1} = \Sigma_0^{-1}+G_{s,t}$, $\bar{\Sigma}_t^{-1} = \Sigma_q^{-1}+\sum_{s\in[m]}(\Sigma_0+G_{s,t}^{-1})^{-1}$. Then, analogous to the proof for Proposition A.1, we introduce the matrix $\tilde{X}_{s,t} \triangleq \big(\Sigma_0^{-1} + \frac{1}{B^2\lambda_1(\Sigma_0)+\sigma^2}\sum_{\ell<t}\mathbf{1}\{s\in\mathcal{S}_\ell\}(\sum_{a\in A_{s,t}}\Phi_a\Phi_a^\top)\big)^{-1}$. We next bound $\sum_{t\geq 1}\sum_{s\in\mathcal{S}_t}\sum_{a\in A_{s,t}}\Phi_a^\top\tilde{\Sigma}_{s,t}\Phi_a$ and $\sum_{t\geq 1}\sum_{s\in\mathcal{S}_t}\sum_{a\in A_{s,t}}\Phi_a^\top\tilde{\Sigma}_{s,t}\Sigma_0^{-1}\bar{\Sigma}_t\Sigma_0^{-1}\tilde{\Sigma}_{s,t}\Phi_a$.

**(1) Bounding** $\sum_{t\geq 1}\sum_{s\in\mathcal{S}_t}\sum_{a\in A_{s,t}}\Phi_a^\top\tilde{\Sigma}_{s,t}\Phi_a$.

$$\sum_{t\geq 1}\sum_{s\in\mathcal{S}_t}\sum_{a\in A_{s,t}}\Phi_a^\top\tilde{\Sigma}_{s,t}\Phi_a$$

$$=[\sigma^2+B^2\lambda_1(\Sigma_0)]\sum_{t\geq 1}\sum_{s=1}^m\mathbf{1}\{\mathcal{S}_t=s\}\sum_{a\in A_{s,t}}\frac{\Phi_a^\top\tilde{\Sigma}_{s,t}\Phi_a}{\sigma^2+B^2\lambda_1(\Sigma_0)}$$

$$\leq 2[\sigma^2+B^2\lambda_1(\Sigma_0)]\sum_{t\geq 1}\sum_{s=1}^m\mathbf{1}\{\mathcal{S}_t=s\}\sum_{a\in A_{s,t}}\log\big(1+\frac{\Phi_a^\top\tilde{\Sigma}_{s,t}\Phi_a}{\sigma^2+B^2\lambda_1(\Sigma_0)}\big)$$

$$\leq 2[\sigma^2+B^2\lambda_1(\Sigma_0)]\sum_{t\geq 1}\sum_{s=1}^m\mathbf{1}\{\mathcal{S}_t=s\}\sum_{a\in A_{s,t}}\log\det(I+\frac{\tilde{\Sigma}_{s,t}^{\frac{1}{2}}\Phi_a\Phi_a^\top\tilde{\Sigma}_{s,t}^{\frac{1}{2}}}{\sigma^2+B^2\lambda_1(\Sigma_0)})$$

$$=2[\sigma^2 + B^2\lambda_1(\Sigma_0)]\sum_{s=1}^m \sum_{t\geq 1}\sum_{a\in A_{s,t}} \mathbf{1}\{\mathcal{S}_t = s\}\big[\log\det(\tilde{X}_{s,t}^{-1} + \frac{\Phi_a\Phi_a^\top}{\sigma^2 + B^2\lambda_1(\Sigma_0)}) - \log\det(\tilde{X}_{s,t}^{-1})\big]$$

$$=2[\sigma^2 + B^2\lambda_1(\Sigma_0)]\sum_{s=1}^m \big[\log\det(\tilde{X}_{s,mn+1}^{-1}) - \log\det(\tilde{X}_{s,1}^{-1})\big]$$

$$=2[\sigma^2 + B^2\lambda_1(\Sigma_0)]\sum_{s=1}^m \log\det\big(I + \frac{1}{\sigma^2 + B^2\lambda_1(\Sigma_0)}\sum_{t\leq mn}\mathbf{1}\{\mathcal{S}_t = s\}\sum_{a\in A_{s,t}}\hat{\Sigma}_{s,t}^{\frac{1}{2}}\Phi_a\Phi_a^\top\hat{\Sigma}_{s,t}^{\frac{1}{2}}\big)$$

$$=2[\sigma^2 + B^2\lambda_1(\Sigma_0)]\sum_{s=1}^m \log\frac{\mathrm{Tr}\big(I + \frac{1}{\sigma^2 + B^2\lambda_1(\Sigma_0)}\sum_{t\leq mn}\mathbf{1}\{\mathcal{S}_t = s\}\sum_{a\in A_{s,t}}\hat{\Sigma}_{s,t}^{\frac{1}{2}}\Phi_a\Phi_a^\top\hat{\Sigma}_{s,t}^{\frac{1}{2}}\big)}{d}$$

$$=2[\sigma^2 + B^2\lambda_1(\Sigma_0)]\sum_{s=1}^m \log\big(1 + \frac{\sum_{t\leq mn}\mathbf{1}\{\mathcal{S}_t = s\}\sum_{a\in A_{s,t}}\Phi_a^\top\hat{\Sigma}_{s,t}\Phi_a}{d(\sigma^2 + B^2\lambda_1(\Sigma_0))}\big)$$

$$\leq 2[\sigma^2 + B^2\lambda_1(\Sigma_0)]m\log\big(1 + \frac{nL}{d}\big) = 2c_1 m\log\big(1 + \frac{nL}{d}\big).$$

**(2) Bounding** $\sum_{t\geq 1}\sum_{s\in\mathcal{S}_t}\sum_{a\in A_{s,t}}\Phi_a^\top\tilde{\Sigma}_{s,t}\Sigma_0^{-1}\bar{\Sigma}_t\Sigma_0^{-1}\tilde{\Sigma}_{s,t}\Phi_a$.

$\forall t\geq 1, s\in\mathcal{S}_t, \forall a\in A_{s,t}$, we have $\Phi_a^\top\tilde{\Sigma}_{s,t}\Sigma_0^{-1}\bar{\Sigma}_t\Sigma_0^{-1}\tilde{\Sigma}_{s,t}\Phi_a \leq B^2\lambda_1(\Sigma_q)\kappa(\Sigma_0) + LB^2\lambda_1(\Sigma_0) + \sigma^2$. Meanwhile, define the matrix $M\triangleq \big(\tilde{\Sigma}_{s,t}^{\frac{1}{2}}\Phi_{a_1}, \tilde{\Sigma}_{s,t}^{\frac{1}{2}}\Phi_{a_2},\ldots,\tilde{\Sigma}_{s,t}^{\frac{1}{2}}\Phi_{a_{|A_{s,t}|}}\big)\in\mathbb{R}^{d\times|A_{s,t}|}$, we have $\sum_{a\in A_{s,t}}\tilde{\Sigma}_{s,t}^{\frac{1}{2}}\Phi_a\Phi_a^\top\tilde{\Sigma}_{s,t}^{\frac{1}{2}} = MM^\top$. Using the Wely's inequality, we further have

$$\lambda_1(I + \sigma^{-2}M^\top M) \leq \lambda_1(I) + \lambda_1(\sigma^{-2}MM^\top) \leq 1 + \sigma^{-2}\sum_{a\in A_{s,t}}\lambda_1(\tilde{\Sigma}_{s,t}^{\frac{1}{2}}\Phi_a\Phi_a^\top\tilde{\Sigma}_{s,t}^{\frac{1}{2}}) = 1 + \sigma^{-2}\sum_{a\in A_{s,t}}\Phi_a^\top\tilde{\Sigma}_{s,t}\Phi_a.$$

Then we can estimate the gap between matrix $\bar{\Sigma}_{t+1}^{-1}$ and $\bar{\Sigma}_t^{-1}$ as follow:

$$\bar{\Sigma}_{t+1}^{-1} - \bar{\Sigma}_t^{-1}$$
$$=\big(\Sigma_0 + (G_{s,t} + \sigma^{-2}\sum_{a\in A_{s,t}}\Phi_a\Phi_a^\top)^{-1}\big)^{-1} - \big(\Sigma_0 + G_{s,t}^{-1}\big)^{-1}$$
$$=\Sigma_0^{-1} - \Sigma_0^{-1}(\Sigma_0^{-1} + G_{s,t} + \sigma^{-2}\sum_{a\in A_{s,t}}\Phi_a\Phi_a^\top)^{-1}\Sigma_0^{-1} - [\Sigma_0^{-1} - \Sigma_0^{-1}(\Sigma_0^{-1} + G_{s,t})^{-1}\Sigma_0^{-1}]$$
$$=\Sigma_0^{-1}[\tilde{\Sigma}_{s,t} - (\tilde{\Sigma}_{s,t}^{-1} + \sigma^{-2}\sum_{a\in A_{s,t}}\Phi_a\Phi_a^\top)^{-1}]\Sigma_0^{-1}$$
$$=\Sigma_0^{-1}\tilde{\Sigma}_{s,t}^{\frac{1}{2}}\big[I - (I + \sigma^{-2}\sum_{a\in A_{s,t}}\tilde{\Sigma}_{s,t}^{\frac{1}{2}}\Phi_a\Phi_a^\top\tilde{\Sigma}_{s,t}^{\frac{1}{2}})^{-1}\big]\tilde{\Sigma}_{s,t}^{\frac{1}{2}}\Sigma_0^{-1}$$
$$=\Sigma_0^{-1}\tilde{\Sigma}_{s,t}^{\frac{1}{2}}\big[I - (I + \sigma^{-2}MM^\top)^{-1}\big]\tilde{\Sigma}_{s,t}^{\frac{1}{2}}\Sigma_0^{-1}$$
$$=\Sigma_0^{-1}\tilde{\Sigma}_{s,t}^{\frac{1}{2}}\big[\sigma^{-2}M(I + \sigma^{-2}M^\top M)^{-1}M^\top\big]\tilde{\Sigma}_{s,t}^{\frac{1}{2}}\Sigma_0^{-1}$$
$$\geq\Sigma_0^{-1}\tilde{\Sigma}_{s,t}^{\frac{1}{2}}\big[\sigma^{-2}M\lambda_d\big((I + \sigma^{-2}M^\top M)^{-1}\big)M^\top\big]\tilde{\Sigma}_{s,t}^{\frac{1}{2}}\Sigma_0^{-1}$$
$$=\Sigma_0^{-1}\tilde{\Sigma}_{s,t}^{\frac{1}{2}}\big[\sigma^{-2}\frac{1}{\lambda_1(I + \sigma^{-2}M^\top M)}MM^\top\big]\tilde{\Sigma}_{s,t}^{\frac{1}{2}}\Sigma_0^{-1}$$
$$\geq\Sigma_0^{-1}\tilde{\Sigma}_{s,t}^{\frac{1}{2}}\big[\sigma^{-2}\frac{1}{1 + \sigma^{-2}\sum_{a\in A_{s,t}}\Phi_a^\top\tilde{\Sigma}_{s,t}\Phi_a}MM^\top\big]\tilde{\Sigma}_{s,t}^{\frac{1}{2}}\Sigma_0^{-1}$$
$$=\frac{\Sigma_0^{-1}\tilde{\Sigma}_{s,t}\big(\sum_{a\in A_{s,t}}\Phi_a\Phi_a^\top\big)\tilde{\Sigma}_{s,t}\Sigma_0^{-1}}{\sigma^2 + \sum_{a\in A_{s,t}}\Phi_a^\top\tilde{\Sigma}_{s,t}\Phi_a}$$
$$\geq\frac{\Sigma_0^{-1}\tilde{\Sigma}_{s,t}\big(\sum_{a\in A_{s,t}}\Phi_a\Phi_a^\top\big)\tilde{\Sigma}_{s,t}\Sigma_0^{-1}}{\sigma^2 + B^2 L\lambda_1(\Sigma_0) + B^2\lambda_1(\Sigma_q)\kappa(\Sigma_0)}, \tag{6}$$

where the second and the sixth equality hold due to the Woodbury matrix identity. The proof for Eq. (6) is similar to the proof for Eq. (5), but requires more refined analysis (i.e. the first inequality in Eq. (6)) to estimate the lower bound of $\bar{\Sigma}_{t+1}^{-1} - \bar{\Sigma}_t^{-1}$. Then recall $c_4 = \sigma^2 + B^2 L\lambda_1(\Sigma_0) + B^2\lambda_1(\Sigma_q)\kappa(\Sigma_0)$ for brevity, we can bound $\sum_{t\geq 1}\sum_{s\in\mathcal{S}_t}\sum_{a\in A_{s,t}}\Phi_a^\top\tilde{\Sigma}_{s,t}\Sigma_0^{-1}\bar{\Sigma}_t\Sigma_0^{-1}\tilde{\Sigma}_{s,t}\Phi_a$ as follow:

$$\sum_{t\geq 1}\sum_{s\in\mathcal{S}_t}\sum_{a\in A_{s,t}}\Phi_a^\top\tilde{\Sigma}_{s,t}\Sigma_0^{-1}\bar{\Sigma}_t\Sigma_0^{-1}\tilde{\Sigma}_{s,t}\Phi_a$$

$$\leq 2c_4\sum_{t\geq 1}\sum_{s\in\mathcal{S}_t}\sum_{a\in A_{s,t}}\log\big(1 + \frac{\Phi_a^\top\tilde{\Sigma}_{s,t}\Sigma_0^{-1}\bar{\Sigma}_t\Sigma_0^{-1}\tilde{\Sigma}_{s,t}\Phi_a}{\sigma^2 + B^2 L\lambda_1(\Sigma_0) + B^2\lambda_1(\Sigma_q)\kappa(\Sigma_0)}\big)$$

$$= 2c_4\sum_{t\geq 1}\sum_{s\in\mathcal{S}_t}\sum_{a\in A_{s,t}}\log\det(I + \frac{\bar{\Sigma}_t^{\frac{1}{2}}\Sigma_0^{-1}\tilde{\Sigma}_{s,t}\Phi_a\Phi_a^\top\tilde{\Sigma}_{s,t}\Sigma_0^{-1}\bar{\Sigma}_t^{\frac{1}{2}}}{\sigma^2 + B^2 L\lambda_1(\Sigma_0) + B^2\lambda_1(\Sigma_q)\kappa(\Sigma_0)})$$

$$= 2c_4\sum_{t\geq 1}\sum_{s\in\mathcal{S}_t}\sum_{a\in A_{s,t}}\big[\log\det(\bar{\Sigma}_t^{-1} + \frac{\Sigma_0^{-1}\tilde{\Sigma}_{s,t}\Phi_a\Phi_a^\top\tilde{\Sigma}_{s,t}\Sigma_0^{-1}}{\sigma^2 + B^2 L\lambda_1(\Sigma_0) + B^2\lambda_1(\Sigma_q)\kappa(\Sigma_0)}) - \log\det(\bar{\Sigma}_t^{-1})\big]$$

$$\leq 2c_4 L\sum_{t\geq 1}\sum_{s\in\mathcal{S}_t}\big[\log\det(\bar{\Sigma}_{t+1}^{-1}) - \log\det(\bar{\Sigma}_t^{-1})\big]$$

$$= 2c_4 L\big[\log\det(\bar{\Sigma}_{mn+1}^{-1}) - \log\det(\bar{\Sigma}_1^{-1})\big]$$

$$= 2c_4 L\big[\log\det(I + \sum_{s\in[m]}\Sigma_q^{\frac{1}{2}}(\Sigma_0 + G_{s,mn+1}^{-1})^{-1}\Sigma_q^{\frac{1}{2}})\big]$$

$$\leq 2c_4 Ld\big[\log\frac{\mathrm{Tr}(I + \sum_{s\in[m]}\Sigma_q^{\frac{1}{2}}(\Sigma_0 + G_{s,mn+1}^{-1})^{-1}\Sigma_q^{\frac{1}{2}})}{d}\big]$$

$$\leq 2c_4 Ld\log(1 + \frac{m\,\mathrm{Tr}(\Sigma_0^{-1}\Sigma_q)}{d}),$$

where the second inequality holds due to Eq. (6). $\qquad\square$

**Lemma D.1** *If a Gaussian random variable $X \sim \mathcal{N}(\mu, \sigma^2)$, then $\mathbb{E}[X\mathbf{1}\{X \geq 0\}] = \mu\big[1 - \Phi_G(-\frac{\mu}{\sigma})\big] + \frac{\sigma}{\sqrt{2\pi}}\exp\{-\frac{\mu^2}{2\sigma^2}\}$. If further $\mu \leq 0$, then $\mathbb{E}[X\mathbf{1}\{X \geq 0\}] = \frac{\sigma}{\sqrt{2\pi}}\exp\{-\frac{\mu^2}{2\sigma^2}\}$.*

**Theorem D.1** *(Theorem 5.4 in the main text, Regret Bound of HierTS in the Semi-Bandit Setting).* Let $|\mathcal{S}_t| = 1$ for any $t \geq 1$. Let $c \geq \sqrt{2\ln\big(\frac{nKB\lambda_1(\Sigma_0)}{\sqrt{2\pi}}\big)}$, $c_1 = \sigma^2 + B^2\lambda_1(\Sigma_0)$, $c_4 = \sigma^2 + B^2 L\lambda_1(\Sigma_0) + B^2\lambda_1(\Sigma_q)\kappa(\Sigma_0)$, *then in the multi-task Gaussian semi-bandit setting, the Bayes regret bound of combinatorial HierTS is:*

$$\mathcal{BR}(m,n) \leq m + c\sqrt{mnL}\sqrt{2c_1 m\log\big(1 + \frac{nL}{d}\big) + 2c_4 Ld\log(1 + \frac{m\,\mathrm{Tr}(\Sigma_0^{-1}\Sigma_q)}{d})}.$$

*Proof.* Note that $\bar{\mathbf{w}}_s = \Phi\theta_{s,*}$, then define $g(A, \theta) = \sum_{a\in A}\langle\Phi_a, \theta\rangle$ for brevity, we have the following result:

$$\mathcal{BR}(m,n) = \mathbb{E}\sum_{t\geq 1}\sum_{s\in\mathcal{S}_t}\big[\sum_{a\in A_{s,*}}\bar{\mathbf{w}}_s(a) - \sum_{a\in A_{s,t}}\bar{\mathbf{w}}_s(a)\big]$$

$$= \mathbb{E}\sum_{t\geq 1}\sum_{s\in\mathcal{S}_t}\big[\sum_{a\in A_{s,*}}\langle\Phi_a, \theta_{s,*}\rangle - \sum_{a\in A_{s,t}}\langle\Phi_a, \theta_{s,*}\rangle\big]$$

$$= \mathbb{E}\sum_{t\geq 1}\sum_{s\in\mathcal{S}_t}\big[g(A_{s,*}, \theta_{s,*}) - g(A_{s,t}, \theta_{s,*})\big].$$

Define upper confidence bound $U_{t,s}(A) = \sum_{a\in A}\big[\langle\Phi_a, \hat{\mu}_{s,t}\rangle + c\sqrt{\Phi_a^\top\hat{\Sigma}_{s,t}\Phi_a}\big]$, where $c$ is a constant to be specified. Notice that $A_{s,*}|H_t \overset{\text{i.i.d.}}{\sim} A_{s,t}|H_t$ and $U_{t,s}(\cdot)$ is a deterministic function, thus

$\mathbb{E}[U_{t,s}(A_{s,*})|H_t] = \mathbb{E}[U_{t,s}(A_{s,t})|H_t]$. Then we can decompose Bayes regret $\mathcal{BR}(m,n)$ as follow:

$$\mathcal{BR}(m,n) = \mathbb{E}\sum_{t\geq 1}\sum_{s\in\mathcal{S}_t}\mathbb{E}\big[g(A_{s,*},\theta_{s,*}) - U_{t,s}(A_{s,*}) + U_{t,s}(A_{s,t}) - g(A_{s,t},\theta_{s,*})|H_t\big]$$
$$= \mathbb{E}\sum_{t\geq 1}\sum_{s\in\mathcal{S}_t}\big[g(A_{s,*},\theta_{s,*}) - U_{t,s}(A_{s,*})\big] + \mathbb{E}\sum_{t\geq 1}\sum_{s\in\mathcal{S}_t}\big[U_{t,s}(A_{s,t}) - g(A_{s,t},\theta_{s,*})\big].$$

**(1) Bounding** $\mathbb{E}\sum_{t\geq 1}\sum_{s\in\mathcal{S}_t}\big[g(A_{s,*},\theta_{s,*}) - U_{t,s}(A_{s,*})\big]$.

For any $t\geq 1$, $s\in\mathcal{S}_t$, $a\in\mathcal{A}$, define random variable $X_{t,s,a} = \langle\Phi_a,\theta_{s,*} - \hat{\mu}_{s,t}\rangle - c\sqrt{\Phi_a^\top\hat{\Sigma}_{s,t}\Phi_a}$,
then we have $X_{t,s,a}|H_t \sim \mathcal{N}(-c\sqrt{\Phi_a^\top\hat{\Sigma}_{s,t}\Phi_a}, \Phi_a^\top\hat{\Sigma}_{s,t}\Phi_a)$ since $\mathbb{E}[\theta_{s,*} - \hat{\mu}_{s,t}|H_t] = \mathbf{0}$. Then

$$\mathbb{E}\sum_{t\geq 1}\sum_{s\in\mathcal{S}_t}\big[g(A_{s,*},\theta_{s,*}) - U_{s,t}(A_{s,*})\big]$$
$$= \mathbb{E}\sum_{t\geq 1}\sum_{s\in\mathcal{S}_t}\sum_{a\in A_{s,*}} X_{t,s,a}$$
$$\leq \mathbb{E}\sum_{t\geq 1}\sum_{s\in\mathcal{S}_t}\sum_{a\in A_{s,*}} X_{t,s,a}\mathbf{1}\{X_{t,s,a}\geq 0\}$$
$$\leq \mathbb{E}\sum_{t\geq 1}\sum_{s\in\mathcal{S}_t}\sum_{a\in[K]} X_{t,s,a}\mathbf{1}\{X_{t,s,a}\geq 0\}$$
$$= \mathbb{E}\sum_{t\geq 1}\sum_{s\in\mathcal{S}_t}\sum_{a\in[K]}\mathbb{E}\big[X_{t,s,a}\mathbf{1}\{X_{t,s,a}\geq 0\}|H_t\big]$$
$$\leq \mathbb{E}\sum_{t\geq 1}\sum_{s\in\mathcal{S}_t}\sum_{a\in[K]}\frac{\sqrt{\Phi_a^\top\hat{\Sigma}_{s,t}\Phi_a}}{\sqrt{2\pi}}\exp\{-\frac{c^2}{2}\}$$
$$\leq \mathbb{E}\sum_{t\geq 1}\sum_{s\in\mathcal{S}_t}\sum_{a\in[K]}\frac{B\lambda_1(\hat{\Sigma}_{s,t})}{\sqrt{2\pi}}\exp\{-\frac{c^2}{2}\}$$
$$\leq nmK\frac{B\lambda_1(\Sigma_0)}{\sqrt{2\pi}}\exp\{-\frac{c^2}{2}\}.$$

If let $nmK\frac{B\lambda_1(\Sigma_0)}{\sqrt{2\pi}}\exp\{-\frac{c^2}{2}\}\leq m$, then $c\geq\sqrt{2\ln\big(\frac{nKB\lambda_1(\Sigma_0)}{\sqrt{2\pi}}\big)}$.

**(2) Bounding** $\mathbb{E}\sum_{t\geq 1}\sum_{s\in\mathcal{S}_t}\big[U_{t,s}(A_{s,t}) - g(A_{s,t},\theta_{s,*})\big]$.

$$\mathbb{E}\sum_{t\geq 1}\sum_{s\in\mathcal{S}_t}\big[U_{t,s}(A_{s,t}) - g(A_{s,t},\theta_{s,*})\big]$$
$$= \mathbb{E}\sum_{t\geq 1}\sum_{s\in\mathcal{S}_t}\sum_{a\in A_{s,t}}\langle\Phi_a,\hat{\mu}_{s,t} - \theta_{s,*}\rangle + c\sqrt{\Phi_a^\top\hat{\Sigma}_{s,t}\Phi_a}$$
$$= \mathbb{E}\sum_{t\geq 1}\sum_{s\in\mathcal{S}_t}\sum_{a\in[K]}\mathbb{E}\big[\mathbf{1}_{a\in A_{s,t}}|H_t\big]\mathbb{E}\big[\langle\Phi_a,\hat{\mu}_{s,t} - \theta_{s,*}\rangle|H_t\big] + c\mathbb{E}\sum_{t\geq 1}\sum_{s\in\mathcal{S}_t}\sum_{a\in A_{s,t}}\sqrt{\Phi_a^\top\hat{\Sigma}_{s,t}\Phi_a}$$
$$= c\mathbb{E}\sum_{t\geq 1}\sum_{s\in\mathcal{S}_t}\sum_{a\in A_{s,t}}\sqrt{\Phi_a^\top\hat{\Sigma}_{s,t}\Phi_a}$$
$$\leq c\sqrt{mnL}\sqrt{\mathbb{E}\sum_{t\geq 1}\sum_{s\in\mathcal{S}_t}\sum_{a\in A_{s,t}}\Phi_a^\top\hat{\Sigma}_{s,t}\Phi_a},$$

where the second equality holds because of the mutual independence between $A_{s,t}|H_t$ and $\theta_{s,*}|H_t$, and $\mathbb{E}[\hat{\mu}_{s,t} - \theta_{s,*}|H_t] = \mathbf{0}$; the last inequality holds due to the Jensen inequality. Then applying

Proposition D.1 to bound $\mathbb{E}\sum_{t\geq 1}\sum_{s\in\mathcal{S}_t}\sum_{a\in[A_{s,t}]}\Phi_a^\top\hat{\Sigma}_{s,t}\Phi_a$, we can obtain

$$\mathbb{E}\sum_{t\geq 1}\sum_{s\in\mathcal{S}_t}\big[U_{t,s}(A_{s,t})-g(A_{s,t},\theta_{s,*})\big]\leq c\sqrt{mnL}\sqrt{2c_1m\log\big(1+\frac{nL}{d}\big)+2c_4Ld\log(1+\frac{m\,\mathrm{Tr}(\Sigma_0^{-1}\Sigma_q)}{d})}.$$

Combining the above results finishes the whole proof. $\qquad\square$

**Theorem D.2** *(Theorem 5.5 in the main text, Regret Bound of HierBayesUCB in the Semi-Bandit Setting). Let $|\mathcal{S}_t|=1$ for all rounds $t\geq 1$. Let $c_1=\sigma^2+B^2\lambda_1(\Sigma_0)$, $c_4=\sigma^2+B^2L\lambda_1(\Sigma_0)+B^2\lambda_1(\Sigma_q)\kappa(\Sigma_0)$, Then for any $\epsilon>0,\delta\in(0,1),\zeta\in(0,1)$, in the multi-task Gaussian semi-bandit setting, the Bayes regret upper bound of combinatorial HierBayesUCB is as follow:*

$$\mathcal{BR}(m,n)\leq\mathbb{E}\big[\frac{8L\log\frac{1}{\delta}}{\Delta_{\min}^\epsilon}\big]\big[2c_1m\log\big(1+\frac{nL}{d}\big)+2c_4Ld\log(1+\frac{m\,\mathrm{Tr}(\Sigma_0^{-1}\Sigma_q)}{d})\big]+mn\epsilon$$

$$+4LB\sqrt{\lambda_1(\Sigma_0+\Sigma_q)}\big(\sqrt{d+\sqrt{8d\ln\frac{1}{\zeta}}}+\|\mu_q\|_{\hat{\Sigma}_{s,1}^{-1}}\big)mnK\delta.$$

*In Theorem 5.5 in the main text, we replace $\sqrt{d+\sqrt{8d\ln\frac{1}{\zeta}}}$ in the right hand side of the above inequality with $\sqrt{d}$ for ease of exposition.*

***Proof.*** Define the event $E_{s,t}=\{\forall a\in\mathcal{A}:|\Phi_a^\top(\theta_{s,*}-\hat{\mu}_{s,t})|\leq\sqrt{2\log\frac{1}{\delta}}\|\Phi_a\|_{\hat{\Sigma}_{s,t}}\}$, and the upper confidence bound $U_{t,s}(A)=\sum_{a\in A}\langle\Phi_a,\hat{\mu}_{s,t}\rangle+\sqrt{2\log\frac{1}{\delta}}\|\Phi_a\|_{\hat{\Sigma}_{s,t}}$. Let $\Delta_{s,t}=g(A_{s,*},\theta_{s,*})-g(A_{s,t},\theta_{s,*})$, then we decompose the Bayes regret into three parts as follow:

$$\mathbb{E}\sum_{t\geq 1}\sum_{s\in\mathcal{S}_t}\Delta_{s,t}$$

$$=\sum_{t\geq 1}\sum_{s\in\mathcal{S}_t}\mathbb{E}\big[\Delta_{s,t}\mathbf{1}\{\Delta_{s,t}\geq\epsilon,E_{s,t}\}\big]+\sum_{t\geq 1}\sum_{s\in\mathcal{S}_t}\mathbb{E}\big[\Delta_{s,t}\mathbf{1}\{\Delta_{s,t}<\epsilon,E_{s,t}\}\big]+\sum_{t\geq 1}\sum_{s\in\mathcal{S}_t}\mathbb{E}\big[\Delta_{s,t}\mathbf{1}\{\bar{E}_{s,t}\}\big].$$

**(1) Bounding** $\sum_{t\geq 1}\sum_{s\in\mathcal{S}_t}\mathbb{E}\big[\Delta_{s,t}\mathbf{1}\{\Delta_{s,t}\geq\epsilon,E_{s,t}\}\big]$.

$$\sum_{t\geq 1}\sum_{s\in\mathcal{S}_t}\mathbb{E}\big[\Delta_{s,t}\mathbf{1}\{\Delta_{s,t}\geq\epsilon,E_{s,t}\}\big]$$

$$=\sum_{t\geq 1}\sum_{s\in\mathcal{S}_t}\mathbb{E}\big[\frac{\big(g(A_{s,*},\theta_{s,*})-g(A_{s,t},\theta_{s,*})\big)^2}{\Delta_{s,t}}\mathbf{1}\{\Delta_{s,t}\geq\epsilon,E_{s,t}\}\big]$$

$$\leq\sum_{t\geq 1}\sum_{s\in\mathcal{S}_t}\mathbb{E}\big[\frac{\big(g(A_{s,*},\theta_{s,*})-U_{t,s}(A_{s,*})+U_{t,s}(A_{s,t})-g(A_{s,t},\theta_{s,*})\big)^2}{\Delta_{s,t}}\mathbf{1}\{\Delta_{s,t}\geq\epsilon,E_{s,t}\}\big]$$

$$\leq\sum_{t\geq 1}\sum_{s\in\mathcal{S}_t}\mathbb{E}\big[\frac{\big(U_{t,s}(A_{s,t})-g(A_{s,t},\theta_{s,*})\big)^2}{\Delta_{s,t}}\mathbf{1}\{\Delta_{s,t}\geq\epsilon,E_{s,t}\}\big]$$

$$=\sum_{t\geq 1}\sum_{s\in\mathcal{S}_t}\mathbb{E}\big[\frac{\big(\sum_{a\in A_{s,t}}\langle\Phi_a,\hat{\mu}_{s,t}-\theta_{s,*}\rangle+\sqrt{2\log\frac{1}{\delta}}\|\Phi_a\|_{\hat{\Sigma}_{s,t}}\big)^2}{\Delta_{s,t}}\mathbf{1}\{\Delta_{s,t}\geq\epsilon,E_{s,t}\}\big]$$

$$\leq\sum_{t\geq 1}\sum_{s\in\mathcal{S}_t}\mathbb{E}\big[\frac{\big(\sum_{a\in A_{s,t}}2\sqrt{2\log\frac{1}{\delta}}\|\Phi_a\|_{\hat{\Sigma}_{s,t}}\big)^2}{\Delta_{s,t}}\mathbf{1}\{\Delta_{s,t}\geq\epsilon,E_{s,t}\}\big]$$

$$\leq\sum_{t\geq 1}\sum_{s\in\mathcal{S}_t}\mathbb{E}\big[\frac{\big(\sum_{a\in A_{s,t}}8\log\frac{1}{\delta}\big)\big(\sum_{a\in A_{s,t}}\|\Phi_a\|_{\hat{\Sigma}_{s,t}}^2\big)}{\Delta_{s,t}}\mathbf{1}\{\Delta_{s,t}\geq\epsilon,E_{s,t}\}\big]$$

$$\leq \mathbb{E}\Big[\frac{8L\log\frac{1}{\delta}}{\Delta_{\min}^{\epsilon}}\sum_{t\geq 1}\sum_{s\in\mathcal{S}_t}\sum_{a\in A_{s,t}}\|\Phi_a\|_{\hat{\Sigma}_{s,t}}^2\Big],$$

where the first and the second inequality hold due to the definition of the upper confidence bound $U_{t,s}(A_{s,t})$, the fourth inequality holds due to the Cauchy-Schwartz inequality. Utilizing the upper bound on $\sum_{t\geq 1}\sum_{s\in\mathcal{S}_t}\sum_{a\in A_{s,t}}\|\Phi_a\|_{\hat{\Sigma}_{s,t}}^2$ in Proposition D.1 completes the proof for the first part.

**(2) Bounding $\sum_{t\geq 1}\sum_{s\in\mathcal{S}_t}\mathbb{E}\big[\Delta_{s,t}\mathbf{1}\{\Delta_{s,t}<\epsilon, E_{s,t}\}\big]$.**
We trivially have $\sum_{t\geq 1}\sum_{s\in\mathcal{S}_t}\mathbb{E}\big[\Delta_{s,t}\mathbf{1}\{\Delta_{s,t}<\epsilon, E_{s,t}\}\big]\leq mn\epsilon.$

**(3) Bounding $\sum_{t\geq 1}\sum_{s\in\mathcal{S}_t}\mathbb{E}\big[\Delta_{s,t}\mathbf{1}\{\bar{E}_{s,t}\}\big]$.**

Note that $\theta_{s,*}-\mu_q\sim\mathcal{N}(0,\hat{\Sigma}_{s,1})$, and $\mathbb{E}\big[\|\theta_{s,*}-\mu_q\|_{\hat{\Sigma}_{s,1}^{-1}}\big]\leq\sqrt{\mathbb{E}\|\hat{\Sigma}_{s,1}^{-\frac{1}{2}}(\theta_{s,*}-\mu_q)\|_2^2}=\sqrt{d}.$ Then according to [35, Exp 2.11], we have with probability $1-\zeta$, $\|\theta_{s,*}-\mu_q\|_{\hat{\Sigma}_{s,1}^{-1}}\leq\sqrt{d+\sqrt{8d\ln\frac{1}{\zeta}}}$ Therefore, with probability $1-\zeta$ over the draw of $\{\theta_{s,*}\}_{s\in[m]}$, we have

$$\begin{aligned}
\Delta_{s,t} &= g(A_{s,*},\theta_{s,*})-g(A_{s,t},\theta_{s,*})\\
&=\sum_{a\in A_{s,*}}\langle\Phi_a,\theta_{s,*}\rangle-\sum_{a\in A_{s,t}}\langle\Phi_a,\theta_{s,*}\rangle\\
&\leq\sum_{a\in A_{s,*}}\|\Phi_a\|\cdot\|\theta_{s,*}\|+\sum_{a\in A_{s,t}}\|\Phi_a\|\cdot\|\theta_{s,*}\|\\
&\leq 2LB\|\theta_{s,*}\|\\
&\overset{1-\zeta}{\leq} 2LB\sqrt{\lambda_1(\Sigma_0+\Sigma_q)}\Big(\sqrt{d+\sqrt{8d\ln\frac{1}{\zeta}}}+\|\mu_q\|_{\hat{\Sigma}_{s,1}^{-1}}\Big),
\end{aligned}$$

where the first inequality holds due to the Schwartz inequality. Then we have with probability $1-\zeta$:

$$\begin{aligned}
&\sum_{t\geq 1}\sum_{s\in\mathcal{S}_t}\mathbb{E}\big[\Delta_{s,t}\mathbf{1}\{\bar{E}_{s,t}\}\big]\\
&=\sum_{t\geq 1}\sum_{s\in\mathcal{S}_t}\mathbb{E}\big[\mathbb{E}\big[\Delta_{s,t}\mathbf{1}\{\bar{E}_{s,t}\}\big|H_t\big]\big]\\
&\leq 2LB\sqrt{\lambda_1(\Sigma_0+\Sigma_q)}\Big(\sqrt{d+\sqrt{8d\ln\frac{1}{\zeta}}}+\|\mu_q\|_{\hat{\Sigma}_{s,1}^{-1}}\Big)\sum_{t\geq 1}\sum_{s\in\mathcal{S}_t}\mathbb{E}\big[\mathbf{1}\{\bar{E}_{s,t}\}\big|H_t\big]\\
&=2LB\sqrt{\lambda_1(\Sigma_0+\Sigma_q)}\Big(\sqrt{d+\sqrt{8d\ln\frac{1}{\zeta}}}+\|\mu_q\|_{\hat{\Sigma}_{s,1}^{-1}}\Big)\sum_{t\geq 1}\sum_{s\in\mathcal{S}_t}\mathbb{P}\big(\bar{E}_{s,t}\big|H_t\big)\\
&\leq 4LB\sqrt{\lambda_1(\Sigma_0+\Sigma_q)}\Big(\sqrt{d+\sqrt{8d\ln\frac{1}{\zeta}}}+\|\mu_q\|_{\hat{\Sigma}_{s,1}^{-1}}\Big)mnK\delta.
\end{aligned}$$

Combining the results in **(1)** **(2)** and **(3)** finishes the whole proof. □

## E  Technical Overview and Limitations of this Work

In this section, we explain our technical novelties for deriving near-optimal sequential regret bound for HierTS and logarithmic sequential regret bound for HierBayesUCB as follow, when compared with the latest bound in [17] (More detailed explanations can be found in Table 4):
**(1) The Technical Overview for Deriving Near-Optimal Regret Bound in Theorem 5.1**. The biggest novelty lies in bounding each term $\mathbb{E}\big[(\theta_{s,*}-\hat{\mu}_{s,t})^{\top}A_{s,*}\big|H_t\big]$ in Bayes regret $\mathcal{BR}(m,n)$. Existing work [17] chose Cauchy-Schwartz inequality to directly bound $(\theta_{s,*}-\hat{\mu}_{s,t})^{\top}A_{s,*}\leq\|\theta_{s,*}-\hat{\mu}_{s,t}\|_{\hat{\Sigma}_{s,t}^{-1}}\|A_{s,*}\|_{\hat{\Sigma}_{s,t}}$, used UCB technique to bound $\|\theta_{s,*}-\hat{\mu}_{s,t}\|_{\hat{\Sigma}_{s,t}^{-1}}$ (which caused an additional multiplicative factor $\log\frac{1}{\delta}$), leveraged the fact that $A_{s,*}|H_t\overset{\text{i.i.d.}}{\sim}A_{s,t}|H_t$ to transform

$\sum_{t,s} \|A_{s,*}\|_{\hat{\Sigma}_{s,t}}$ into $\mathcal{V}_{m,n}$, and obtained an intermediate regret upper bound $\sqrt{mn\mathcal{V}_{m,n}\log(1/\delta)}$. Instead of using UCB technique, our Theorem 5.1 applies a novel Cauchy-Schwartz type inequality (i.e. Lemma A.2) to bound $\mathbb{E}(\theta_{s,*} - \hat{\mu}_{s,t})^\top A_{s,*} \leq \sqrt{d\mathbb{E}\big((\theta_{s,*} - \hat{\mu}_{s,t})^\top A_{s,t}\big)^2} \approx \sqrt{d}\|A_{s,t}\|_{\hat{\Sigma}_{s,t}}$, and finally achieves the regret bound $\sqrt{mn\mathcal{V}_{m,n}}$, removing the $\sqrt{\log(1/\delta)}$ factor. Besides, when bounding the posterior variance $\mathcal{V}_{m,n}$, we use a different matrix analysis to prevent variance terms (e.g. $\sigma^2, \lambda_1(\Sigma_0)$) solely appearing in the denominator of regret bound. Moreover, we employ a matrix decomposition technique (in our Lemma A.1) to reduce the multiplicative factor $\kappa^2(\Sigma_0)$ in [17, Theorems 3-4] to $\kappa(\Sigma_0)$ in our bounds (see more details in Table 4).

**(2) The Technical Overview for Deriving Logarithmic Regret Bound in Theorem 5.2**. To obtain sharper sequential regret bound than the near-optimal regret bound in Theorem 5.1, our Theorem 5.2 chooses the Bayes regret decomposition strategy shown above Theorem 5.2, uses UCB technique to bound the first term in the regret decomposition as $\sum_{t,s} \mathbb{E}\Delta_{s,t}\mathbf{1}\{\Delta_{s,t} \geq \epsilon, E_{s,t}\} \leq \mathcal{V}_{m,n}\log\frac{1}{\delta}$, and finally combines the upper bound on posterior variance $\mathcal{V}_{m,n}$ in Eq. (4) to achieve a logarithmic Bayes regret upper bound of $(\log\frac{1}{\delta})md\log\frac{n}{d}$.

Nevertheless, we also need to point out the limitations of our Bayes regret bounds:

**The Limitations of the Multi-Task Bayes Regret Bounds**. Honestly speaking, our regret bounds have two main limitations, because they are: **(i)** Not advantageous when compared with single-task regret bound. This is because our Bayes regret bounds (e.g. $O(m\sqrt{n\log n})$ in Theorem 5.1) for hierarchical Bayesian bandit problem are almost the same as the summation of regret bounds of learning $m$ Bayesian bandit task independently. This is also the limitation of existing bounds in this field (see [25, 7, 17]). **(ii)** Unable to shed more light on the advantages of multi-task bandit optimization. The existing regret bound $O(m\sqrt{nk})$ for multi-task representation which demonstrated that multi-task regret bound can be smaller for learning a low-dimensional representation (i.e. $k << d$) than the regret bound of $O(m\sqrt{nd})$ for learning each task independently, and the existing regret bound $O(m\sqrt{n\log(1 + nV)})$ for multi-task adversarial linear bandit which proved that the regret bound decreases with more similarity (i.e. smaller $V$) among bandit tasks. Our hierarchical Bayesian bandit model has assumed that different bandit instances are sampled the same meta-distribution, and hence fails to reveal the influence of task similarity to the multi-task Bayes regret.

**Remark E.1** *(The Underlying Causes for the Limitation of Multi-Task Bayes Regret Bound.)*
*The underlying causes for the shortcoming of the multi-task Bayes regret bound is that the upper bound on the posterior variance $\mathcal{V}_{m,n} = \mathbb{E}\sum_{t\geq 1}\sum_{s\in\mathcal{S}_t}\|A_{s,t}\|^2_{\hat{\Sigma}_{s,t}}$ may be not tight enough. Detailed explanations lie in the following three aspects:*

*(1) Recall that in the proof for our Theorem 5.1, we can upper bound the multi-task Bayes regret as $\mathcal{BR}(m,n) \leq \sqrt{mnd}\sqrt{\mathcal{V}_{m,n}}$. Then in Proposition A.1 we use a purely algebraic technique to bound the posterior-variance $V_{m,n} \leq O(m\log n)$, resulting in the final Bayes regret bound of*

$$\sqrt{mnd}\sqrt{V_{m,n}} = O(\sqrt{mnd}\sqrt{m\log n}) = O(m\sqrt{n\log n}),$$

*which is almost the same as the summation of the regret bounds for learning $m$ bandit tasks independently. Therefore, if we can upper bound the posterior-variance $V_{m,n}$ with a bound that is sublinear with respect to $m$ and logarithmic w.r.t. $n$ (e.g. a bound of $O(\sqrt{m}\log n)$), then the final Bayes regret bound will be much sharper. The upper bounds on the posterior-variance $V_{m,n}$ in existing works (e.g. see [25, 7, 17] in our Table 1 in the main text) are also obtained via purely algebraic techniques and are not sharp either (or even worse).*

*(2) In the proof for Proposition A.1, we only give the worst-case upper bound on $\sum_{t\geq 1}\sum_{s\in\mathcal{S}_t}\|A_{s,t}\|^2_{\hat{\Sigma}_{s,t}}$ via purely algebraic technique (thus leading to a worst-case upper bound on the posterior variance $\mathbb{E}\sum_{t\geq 1}\sum_{s\in\mathcal{S}_t}\|A_{s,t}\|^2_{\hat{\Sigma}_{s,t}}$). Such worst-case upper bound is obtained via purely algebraic techniques, ignoring the expectation over the randomness of $A_{s,t}$ and $\hat{\Sigma}_{s,t}$. Therefore, we may achieve sharper regret bound by considering the expectation over the randomness in the posterior variance.*

*(3) To derive a sharper and meaningful upper bound on the posterior-variance $\mathcal{V}_{m,n} = \mathbb{E}\sum_{t\geq 1}\sum_{s\in\mathcal{S}_t}\|A_{s,t}\|^2_{\hat{\Sigma}_{s,t}}$, we need to consider other bounding technique like concentration inequality, or more technical matrix analysis, to achieve an upper bound that is sublinear w.r.t. the*

*number $m$ of tasks and sublinear w.r.t. the number $n$ of iterations per task. Only in this way can we obtain a multi-task regret bound $o(mn)$ that is sublinear w.r.t. $m$ and sublinear w.r.t. $n$.*

*(4) On the other hand, we also consider finding the lower bound of posterior-variance $V_{m,n}$ to show that our upper bound on $V_{m,n}$ is tight, or finding the lower bound of multi-task Bayes regret $\mathcal{BR}(m,n)$ to show that our multi-task Bayes regret upper bound could not be improved. This serves as one of our ongoing research directions.*

# F   Additional Experiments and Computer Resources

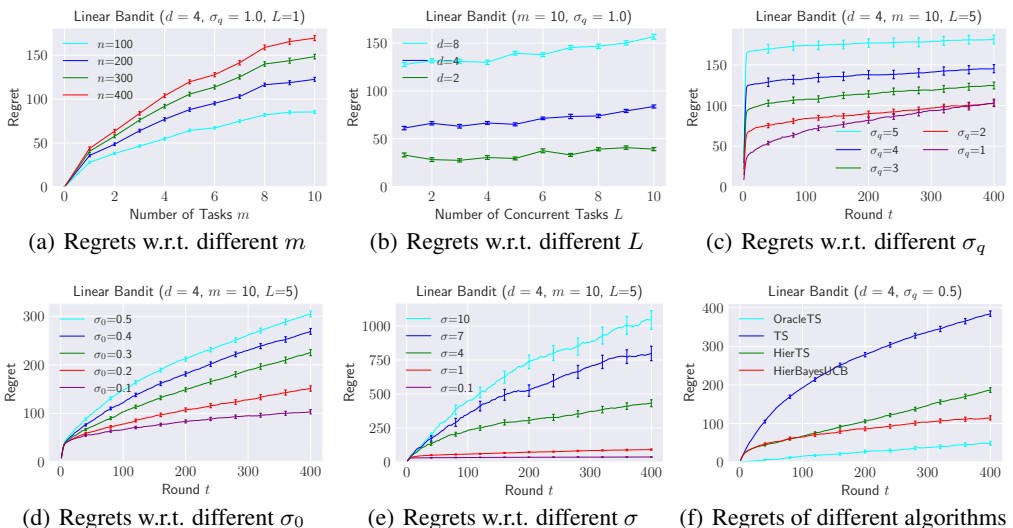

Figure 2: Regrets of HierBayesUCB algorithm with respect to (w.r.t.) different hyper-parameters.

**Experimental Results**. From Figure 2, we have the similar observations as that in Figure 1: **(1)** In plot (a), the multi-task regret of HierBayesUCB becomes larger with the increase of $m$ and $n$, which is consistent with our regret upper bound in Theorems 5.2. **(2)** In plot (b), the regret increases with a higher dimension $d$, and increases with a larger number $L$ of the concurrent tasks. **(3)** In plots (c)-(e), the regret decreases with a smaller variance (e.g. $\sigma_q$, $\sigma_0$ and $\sigma$) in hierarchical Bayesian model, validating the provable benefits of variance-reduction in Bayes regret minimization. **(4)** The task-averaged regret of our proposed HierBayesUCB is smaller than that of HierTS, and such improvement becomes larger with the increase of $\sigma_q$ (when compared with $\sigma_q = 1.0$ in Figure 1 (f)).

**Computer Resources**. Our implementations are based on Python. We run all bandit algorithms on a platform with 8 NVIDIA RTX 6000 GPUs and 2 AMD EPYC 7543 Processors. Each GPU has 48G memory, and each CPU has 64 cores. The CUDA version is 12.1, the Python version 3.7.16, the matplotlib version 3.5.3, and the tensorflow version 1.15. The source code for reproducing all experimental results of HierTS and HierBayesUCB is provided in the supplementary material.

Table 4: The technical novelties for deriving our improved sequential regret bound when compared with the latest regret bound in [17, Thm 3]. $m$ is the number of bandit tasks, $n$ is the number of iterations per task, and $d$ is the dimension of action $a \in \mathcal{A}$.

| Regret Bound | [17, Thm 3] | Existing Problems | Improvement Motivations | Our Theorem 5.1 | Our Improvements |
|---|---|---|---|---|---|
| **Bound I** | $dm\sqrt{n\log(mn)\log\left(1+\frac{n\lambda_1(\Sigma_0)}{d\sigma^2}\right)}$ $\times\sqrt{\dfrac{\lambda_1(\Sigma_0)}{\log\left(1+\frac{\lambda_1(\Sigma_0)}{\sigma^2}\right)}}$ | **(1)** There exists an additional factor $\sqrt{\log(mn)}$. | **(1)** Use a Shwartz-type inequality in Lemma A.2, instead of UCB strategy, to bound per-task regret to avoid additional term $\log\frac{1}{\delta}$ (where $\delta = nm$).  **(2)** Define a new matrix $\tilde{X}_{s,t}$ s.t. the denominator in the regret is $\sigma^2 + B^2\lambda_1(\Sigma_0)$, not only $\sigma^2$. Avoid the case that the variance serves alone as the denominator. | $dm\sqrt{n\log\left(1+\frac{n}{d}\right)}$ $\times\sqrt{(\sigma^2 + B^2\lambda_1(\Sigma_0))}$ | **(1)** Our regret bounds remove the $\sqrt{\log(mn)}$ factor.  **(2)** To minimize our bound, it suffices to decrease the variances $\sigma^2, \lambda_1(\Sigma_0), \lambda_1(\Sigma_q)$. |
| **Bound II** | $d\sqrt{mn\log(mn)\log\left(1+\frac{m\lambda_1(\Sigma_q)}{\lambda_d(\Sigma_0)}\right)}$ $\times\sqrt{\dfrac{\lambda_1^2(\Sigma_0)\lambda_1(\Sigma_q)\left(1+\frac{\lambda_1(\Sigma_0)}{\sigma^2}\right)}{\lambda_d^2(\Sigma_0)\log\left(1+\frac{\lambda_1^2(\Sigma_0)\lambda_1(\Sigma_q)}{\lambda_d^2(\Sigma_0)\sigma^2}\right)}}$ | **(1)** There also exists an additional factor $\sqrt{\log(mn)}$.  **(2)** There exists a paradox in this bound, i.e. variance $\sigma^2$ exists in both the denominator and numerator. Then whether we should increase or decrease $\sigma^2$ to minimize the regret bound? | **(3)** Give a more technical analysis in Lemma A.1 to improve $\frac{\lambda_1^2(\Sigma_0)}{\lambda_d^2(\Sigma_0)}$ to $\frac{\lambda_1(\Sigma_0)}{\lambda_d(\Sigma_0)}$ | $d\sqrt{mn\log\left(1+\frac{m\,\mathrm{Tr}(\Sigma_q\Sigma_0^{-1})}{d}\right)}$ $\times\sqrt{\sigma^2 + B^2\lambda_1(\Sigma_0) + B^2\frac{\lambda_1(\Sigma_0)\lambda_1(\Sigma_q)}{\lambda_d(\Sigma_0)}}$ | **(3)** Our regret bounds also show that we should decrease the condition number $\frac{\lambda_1(\Sigma_0)}{\lambda_d(\Sigma_0)}$ of the variance matrix $\Sigma_0$ to minimize the Bayes regret. |

