# OpenReview forum: "Improved Bayes Regret Bounds for Multi-Task Hierarchical Bayesian Bandit Algorithms"
_NeurIPS.cc/2024/Conference — NeurIPS 2024 poster_

### Official Review · Reviewer_rxXv · 2024-07-11

**Soundness:** 3
**Presentation:** 3
**Contribution:** 3
**Rating:** 6
**Confidence:** 2

**Summary:**

The manuscript discusses a multi-task learning problem and proposes a hierarchical Bayesian (HB) bandit approach. In this approach, the agent maintains a meta-posterior distribution over the hyperparameters of the within-task bandit problems. In the HB bandit, each bandit task is characterized by a task parameter. The paper examines both the sequential and concurrent settings for two types of bandit problems: 1) Gaussian Linear bandit and 2) Semi Bandit. In particular, for both sequential and concurrent bandit in the Gaussian Linear bandit setting:
1. The paper improves upon the existing gap-independent Bayes regret bound of HierTS from O(m \sqrt{n \log n  \log (mn)} ) to O(mn\log n) for infinite actions set where m is the number of tasks and n is the number of iterations per task.
2. The paper also extends the HierTS and proposes a new algorithm for finite action sets, HierBayesUCB, with gap-dependent O(m \log n  \log (mn)) regret bound.

Moreover, the paper extends the HierTS and HierBayesUCB for multi-task Gaussian combinatorial semi-bandit setting and derives regret bounds of O(m \sqrt{n} \log n) and O(m \log (mn \log n)).

**Strengths:**

This paper's contribution is important and novel because:
1. It improves upon the existing regret bound for HeirTS in the infinite action setting
2. Proposes a new algorithm to solve the finite action case.
3. They also extend these algorithms to semi-bandit settings with SOTA regret bounds, which is novel.

**Weaknesses:**

Many parts of the paper can be significantly improved:

1. [Novelty] I don’t think we need the last paragraph in the introduction. Instead, the authors should briefly describe the technical novelty they have introduced to improve the HierTS regret bound for both the sequential and concurrent bandit over the existing works. Why was the previous analysis not sufficient?   What is their novel and non-trivial contribution beyond BayesUCB? Please broadly discuss the novelties summarized in the appendix in the main paper, as this is your main contribution that led to improved bounds. Also, discuss the novelty in the techniques beyond using the improved inequality from [21].

2. 140-141: Why do we need this assumption of at least n-round interaction, and how is it ensured in the algorithm?

3. Alg1 and 2:  How do the algorithms choose the set \mathcal{S}_t of the tasks at each time t? Even the task sequence is also not input to the algorithm. It is not clear to me.


4. What is \hat \mu and \hat \simga in Alg 1 (HierBayesUCB)?  It is not defined yet. I found it to be defined later in (3).

5. display (4): If the order of V_m,n is the same as derived in [17] up to multiplicative factors, how does it help to improve the regret bound?  I cannot find the intuition to understand what improved the bound. The authors must discuss this in the main part of the paper.

**Questions:**

1. L228-229: Please elaborate more on how you have achieved this.

Minor:

- Single task semi-Bandit: L 121-122—The authors should mention in the introduction and abstract that they consider coherent cases of semi-bandit problems.

- L53: Latest

- L143:  Fix  ‘=‘

- L159: insert ‘and’

**Limitations:**

The nature of the work is theoretical.

---

> ### Author Rebuttal · Authors · 2024-08-05
>
> # Response to the Review by Reviewer rxXv (Part 1/2)
>
> **Q1. Please broadly discuss the novelties summarized in the appendix in the main paper, as this is your main contribution that led to improved bounds.**\
> A1: Thank you for this suggestion, we will broadly discuss the novelties summarized in the appendix in the main paper in the final version.
>
> **Q2. Also, discuss the novelty in the techniques beyond using the improved inequality from [21].**\
> A2: Thanks. Besides using the improved inequality from [21], our technical novelties lie in the following three aspects and we will clarify them in the revised version: \
> $\mathbf{(1)}$ For the improved regret bound for HierTS in Theorem 5.1 in the sequential bandit setting: our proof has two novelties: $\mathbf{(i)}$ We use a more technical positive semi-definite matrix decomposition analysis (i.e. our Lemma B.1) to reduce the multiplicative factor $\kappa^{2}(\Sigma _{0})$ to $\kappa(\Sigma _{0})$. $\mathbf{(ii)}$ Define a new matrix $\tilde{X} _{s,t}$ such that the denominator in the regret is $\sigma^{2}+B^{2}\lambda _{1}(\Sigma _{0})$, not just $\sigma^{2}$. Avoid the case that the variance serves alone as the denominator. Such technical novelties are also listed in our Table 4.\
> $\mathbf{(2)}$ For the improved regret bound for HierBayesUCB in Theorem 5.2 in the sequential bandit setting: our technical novelty lies in decomposing the Bayes regret $\mathcal{BR}(m,n)=\mathbb{E}\sum _{t\geq1}\sum _{s \in \mathcal{S} _{t}}\Delta _{s,t}$ into three terms:
> $$\mathbb{E}\sum _{t\geq1}\sum _{s \in \mathcal{S} _{t}}\Delta _{s,t}=\mathbb{E}\sum _{t \geq 1,s \in \mathcal{S} _{t}}\Delta _{s,t}\big[\mathbf{1}\lbrace\Delta _{s,t}\geq \epsilon, E _{s,t}\rbrace + \mathbf{1}\lbrace\Delta _{s,t}< \epsilon,E _{s,t}\rbrace +\mathbf{1}\lbrace \bar{E} _{s,t}\rbrace\big],$$ and bounding the first term with a new method as well as the specific property of BayesUCB algorithm as follow: $$ \mathbb{E}\Delta _{s,t} \mathbf{1}\lbrace\Delta _{s,t} \geq \epsilon, E _{s,t}\rbrace=\mathbb{E}\frac{\Delta _{s,t}^{2}}{\Delta _{s,t}} \mathbf{1}\lbrace\Delta _{s,t} \geq \epsilon, E _{s,t}\rbrace \leq \mathbb{E}\frac{C _{t,s, A _{s,t}}^{2}}{\Delta _{\min}^{\epsilon}},$$ resulting in the final improved gap-dependent regret bound for HierBayesUCB as follows $$ \big(\sum _{t\geq1,s\in \mathcal{S} _{t}}\lVert A _{s,t}\rVert _{\hat{\Sigma} _{s,t}}^{2}\log{\frac{1}{\delta}}\big)/\min _{s,t}\lvert\Delta _{s,t}\rvert \leq O\big(md\log{(n)}\log{\frac{1}{\delta}} \big),$$ which is of order $ O\big(md\log{(n)}\log{(mn)} \big)$ if we set $\delta = \frac{1}{mn}$.\
> $\mathbf{(3)}$ For the improved regret bounds for HierTS and HierBayesUCB in the concurrent setting and in the sequential semi-bandit setting: besides the aforementioned technical novelties in $\mathbf{(1)}$ and $\mathbf{(2)}$, the additional technical novelty lies in leveraging more refined analysis (e.g. using Woodbury matrix identity repeatedly) to bound the gap between the matrices $\bar{\Sigma} _{t+1}^{-1}$ and $\bar{\Sigma} _{t}^{-1}$ (more details can be found in Lemma D.1 and Equation (6) in Page 22).
>
> **Q3. L140-141: Why do we need this assumption of at least $n$-round interaction, and how is it ensured in the algorithm?**\
> A3: Thanks. Actually in L140-141 the assumption is at $\mathbf{MOST}$ $n$-round interactions. We adopt such assumption just for convenient comparison with exiting regret upper bounds (e.g. the bounds in Tables 1 and 2) for multi-task bandit/semi-bandit problem which directly assumes $m$ tasks and $n$ iterations per task. Such assumption is without loss generality and hence can be easily ensured in the algorithm (more implementation details can be confirmed in the code in our submitted zip file).
>
> **Q4. Alg1 and 2: How do the algorithms choose the set $\mathcal{S} _t$ of the tasks at each time t? Even the task sequence is also not input to the algorithm.**\
> A4: Thanks. The task set $\mathcal{S} _t$ is chosen randomly at each iteration $t$.
>
> **Q5. What is $\hat{\mu}$ and $\hat{\sigma}$ in Alg 1 (HierBayesUCB)? It is not defined yet. I found it to be defined later in (3).**\
> A5: Sorry for the confusion. $\hat{\mu}$ and $\hat{\sigma}$ in Alg 1 (HierBayesUCB) are actually the expectation and covariance of the conditional distribution (given the history $H$) of the true task parameter $\theta _{*}$ (including the form of hierarchical Gaussian bandit in Eq.(3)). We will clarify them when introducing HierBayesUCB algorithm in the revised version.

---

> ### Author Response · Authors · 2024-08-05
> **Response to the Review by Reviewer rxXv (Part 2/2)**
>
> **Q6. Display (4): If the order of $\mathcal{V} _{m,n}$ is the same as derived in [17] up to multiplicative factors, how does it help to improve the regret bound?**\
> A6: Thanks. Our explanations are two-fold: \
> $\mathbf{(1)}$ It is true that the order of $\mathcal{V} _{m,n}$ is the same (w.r.t. $m$ and $n$) as derived in [17], but our bound on $\mathcal{V} _{m,n}$ has a smaller multiplicative factor (i.e. using our proposed Lemma B.1 to reduce $\kappa^{2}(\Sigma _{0})$ to $\kappa(\Sigma _{0})$), this is where our first improvement lies in.\
> $\mathbf{(2)}$ The improvement of the order of regret bound (i.e. w.r.t. to $m$ and $n$) is actually attributed to the novel strategy to transform the multi-task Bayes regret $\mathcal{BR}(m,n)$ into an intermediate regret upper bound that involves the posterior variance $\mathcal{V} _{m,n}$ as the dominant term. Previous work [17, Lemma 1] used traditional UCB bounding technique (with the confidence factor $\delta \in (0,1)$) to derive the intermediate regret upper bound $\sqrt{mn\mathcal{V} _{m,n} \log{(1/\delta)}} + mn\delta$, resulting in the additional multiplicative factor $\sqrt{\log{(1/\delta)}}$ which is $\sqrt{\log{(mn)}}$ if we set $\delta = \frac{1}{mn}$. Our strategies differ from [17] in the following two aspects:\
> $\mathbf{(i)}$ For the improved regret bound of HierTS in Theorem 5.1: we leverage a novel Cauchy-Schwartz inequality from [21] to transform the multi-task Bayes regret $\mathcal{BR}(m,n)$ into an intermediate regret upper bound of $O(\sqrt{mn\mathcal{V} _{mn}})$, which is different from the UCB bounding technique and hence removes the additional $\sqrt{\log{(1/\delta)}}$ factor.\
> $\mathbf{(ii)}$ For the improved regret bound of HierBayesUCB in Theorem 5.2: we choose to decompose the Bayes regret $\mathcal{BR}(m,n)=\mathbb{E}\sum _{t\geq1}\sum _{s \in \mathcal{S} _{t}}\Delta _{s,t}$ into three terms:
> $$\mathbb{E}\sum _{t\geq1}\sum _{s \in \mathcal{S} _{t}}\Delta _{s,t}=\mathbb{E}\sum _{t \geq 1,s \in \mathcal{S} _{t}}\Delta _{s,t}\big[\mathbf{1}\lbrace\Delta _{s,t}\geq \epsilon, E _{s,t}\rbrace + \mathbf{1}\lbrace\Delta _{s,t}< \epsilon,E _{s,t}\rbrace +\mathbf{1}\lbrace \bar{E} _{s,t}\rbrace\big] ,$$ and bound the first term (the main term) with the property of BayesUCB algorithm (with confidence factor $\delta \in (0,1)$) as follow $$ \mathbb{E}\Delta _{s,t} \mathbf{1}\lbrace\Delta _{s,t} \geq \epsilon, E _{s,t}\rbrace=\mathbb{E}\frac{\Delta _{s,t}^{2}}{\Delta _{s,t}} \mathbf{1}\lbrace\Delta _{s,t} \geq \epsilon, E _{s,t}\rbrace \leq \mathbb{E}\frac{C _{t,s, A _{s,t}}^{2}}{\Delta _{\min}^{\epsilon}},$$ leading to the improved intermediate regret bound of $\sum _{t\geq 1, s\in \mathcal{S} _{t}}\frac{8\log{\frac{1}{\delta}}\lVert A _{s,t}\rVert _{\hat{\Sigma} _{s,t}}}{\Delta _{\min}^{\epsilon}}=O(\mathcal{V} _{m,n}\log{\frac{1}{\delta}})$, which is of $O(m\log{n}\log{(mn)})$ if we set $\delta = \frac{1}{mn}$.
>
> **Q7. L228-229: Please elaborate more on how you have achieved this.**\
> A7: Thanks. The improvement lies in our upper bound on $\lambda _{1}(\Sigma _{0}^{-1}\tilde{\Sigma} _{s,t}\tilde{\Sigma} _{s,t} \Sigma _{0}^{-1})$, and detailed explanations are two-fold:\
> $\mathbf{(1)}$ Previous work [17, Appendix B] directly used Weyl’s inequality to upper bound $$\lambda _{1}(\Sigma _{0}^{-1}\tilde{\Sigma} _{s,t}\tilde{\Sigma} _{s,t} \Sigma _{0}^{-1}) \leq \lambda _{1}^{2}(\Sigma _{0}^{-1}) \lambda _{1}^{2}(\tilde{\Sigma} _{s,t}) \leq \lambda _{1}^{2}(\Sigma _{0}^{-1}) \lambda _{1}^{2}(\Sigma _{0}) = \kappa^{2}(\Sigma _{0}).$$
> $\mathbf{(2)}$ Instead of directly using Weyl’s inequality, we first propose Lemma B.1 which uses positive semi-definite matrix diagonalization technique to bound $$\lambda _{1}\big[\big((I+AB)(I+BA)\big)^{-1}\big]\leq \frac{\lambda _{1}(A)}{\lambda _{d}(A)}.$$ Then we apply Lemma B.1 to upper bound
> $$\lambda _{1}(\Sigma _{0}^{-1}\tilde{\Sigma} _{s,t}\tilde{\Sigma} _{s,t} \Sigma _{0}^{-1}) = \lambda _{1}(\Sigma _{0}^{-1}\tilde{\Sigma} _{s,t}\tilde{\Sigma} _{s,t} \Sigma _{0}^{-1})\leq \lambda _{1}\big[\big((I+\Sigma _{0}\tilde{\Sigma} _{s,t})(I+\tilde{\Sigma} _{s,t}\Sigma _{0})\big)^{-1}\big] \leq \kappa(\Sigma _{0}),$$ resulting in a smaller multiplicative factor than that in [17].
>
> **Q8. Single task semi-Bandit: L 121-122—The authors should mention in the introduction and abstract that they consider coherent cases of semi-bandit problems.**\
> A8: Thanks for your suggestion, we will clarify in the introduction and abstract that we consider coherent cases of semi-bandit problems in the revised version.
>
> **References**\
> [17] Hierarchical Bayesian Bandits. AISTATS 2022.\
> [21] An Improved Regret Bound for Thompson Sampling in the Gaussian Linear Bandit Setting. ISIT 2021.

---

> > ### Comment · Reviewer_rxXv · 2024-08-13
> >
> > I thank the authors for their detailed response. I will maintain my score.

---

> > > ### Author Response · Authors · 2024-08-13
> > > **Thank you for the response.**
> > >
> > > Dear Reviewer rxXv,\
> > >      Thank you for the response. We benefit a lot from your reviews and kind suggestions. We will take them into the revision of this paper and give more explanations for our technical contributions. Thank you!\
> > > Best,\
> > > Authors

---

### Official Review · Reviewer_diEN · 2024-07-12

**Soundness:** 3
**Presentation:** 3
**Contribution:** 3
**Rating:** 6
**Confidence:** 4

**Summary:**

The paper improves the Bayesian regret bound for hierarchical Bayesian bandit algorithms in multi-task bandit and semi-bandit settings. Firstly, it improves the gap-independent bound by a factor of $\mathcal{O}(\sqrt{\log(mn)})$ for infinite action set, $m$ being number of tasks and $n$, the number of iterations per task. For finite action set, the authors propose and analyze HierBayesUCB algorithm and analyze its gap-dependent Bayesian regret. Finally, the paper extends these algorithms to multi-task combinatorial semi-bandit setting.

**Strengths:**

The paper deals with an interesting hierarchical bandit/semi-bandit setting, where the agent needs to optimize its policy for several tasks simultaneously based on semi-bandit feedback. For the Gaussian setting, the paper provides Bayesian regret bounds that either improve upon previous results, or provide new results (e.g., HierBayesUCB in the combinatorial setting).

**Weaknesses:**

The main weakness of the paper is the requirement of Gaussian distributions for each layer of the hierarchy including the noise. While such an assumption facilitates the development of closed-form posteriors to ease the learning task, it is highly restrictive since in many cases, the noise is not Gaussian (assuming sub-Gaussian would be better, many classes of noise distributions including uniform, truncated Gaussian or Rademacher could be accommodated). This should at least be mentioned in the abstract since the current abstract gives the sense of a more general theory being developed in the paper.

**Questions:**

Here are a few questions regarding the paper:
1. In the case of several tasks, what is the benefit of assuming a hierarchical structure rather than dealing with each task independently? In Bayesian hierarchical models, this kind of shared structure often allows borrowing of strength across the different groups (tasks here), so that the overall rate could be better -- in this case, the overall regret is linear in $m$, which seems to be similar to that of dealing with the tasks independently. Can the authors shed more light on this.
2. Is it possible to use this result to derive frequentist regret bounds (either high probability bounds or in expectation), given either some true $\theta_{s}^*$. To keep the tasks exchangeable, one can alternatively assume a true $Q^*$ and $\{\theta_s^*\}$ as latent variables. Furthermore, in the Gaussian case, is the lower bound $\Omega(d\sqrt{n})$ tight in this hierarchical (shared) setting (note that this lower bound is for a linear bandit learning a single task, while in this case, it is not immediately obvious whether or not the shared structure across the tasks could help in the regret bound)
3. I am concerned with the Gaussian assumptions required for all the theoretical results. While the distributions for $Q$ and $\theta_{s}^*|\mu_*$ are fine to assume Gaussian (equivalent to a Gaussian prior on $(\mu_{1}^*,\dots,\mu_{m}^*)$ with dependence structure), the issue is with the Gaussian noise, which is restrictive to apply the method to broader range of problems. Unlike this work, Thompson sampling for linear bandits have been studied where the Gaussianeity assumption in the likelihood is only taken for algorithmic convenience, whereas the theoretical results only require milder assumptions like sub-Gaussianeity (e.g., [Agrawal and Goyal]). Furthermore, the paper does not deal with the case where $\mu_q, \Sigma_q, \Sigma_0, \sigma$ are unknown -- although the experiments place further priors on them, but this essentially makes the model a mixture, whereby the theoretical results are no longer valid.
4. The term $\mathcal{V}_{m,n}$ is the posterior variance of which distribution?
5. Regarding novelty in technical contribution, it seems the application of the Cauchy Schwarz inequality from [C. Kalkanli and A. Özgür.] (line 204) results in the reduction of the extra factor of $\sqrt{\log(mn)}$ from both Theorems 5.1 and 5.3, which is the main improvement in the paper. For Theorem 5.2, it seems like the fact that $|\mathcal{A}|<\infty$ is the key to reducing the $\sqrt{n}$ factor in the averaged Bayes regret (compared to Theorem 5.1), otherwise, trivially $\Delta_{s,\min}=0$ for compact $\mathcal{A}$ -- however, the additional term of $\log(m)$ is intriguing, can the authors comment on this? The last comment in line 267 shows that if $\Delta_{\min}$ is small, then this bound is much worse, scaling as $m^{3/2}$ (also the additional $\sqrt{\log(mn)}$ reappears).
6. Can the authors comment on the technical difference to remove the polynomial dependence of $K$ to logarithmic in Theorem 5.4 for the combinatorial case?
7. The bound in Theorem 5.5 is stronger than that in [S. Basu, B. Kveton, M. Zaheer, and C. Szepesvári.] in the case $\Delta_{\min}$ is large, otherwise the comment in line 266 would result in a bound worse -- is this correct?
8. Algorithms 1 and 2: Apart from $Q$, the update step for $Q_{t+1}$ also requires the various other likelihood and model parameters (assumed known) like $\mu_q, \Sigma_q, \sigma$ which should be included as input. Also, $\delta$ is an input parameter for the HierBayesUCB algorithm.
9. The *ORACLE* used in the semi-bandit setting: In practice, this is a combinatorially hard problem (e.g., the work by Chen, Wang and Yuan considers a $(\alpha,\beta)-$approximate oracle for efficiency). No simulation examples were shown to demonstrate the performance and more importantly, the computational complexity, for this important step in both HierTS and HierBayesUCB algorithms.
10. What are the values of $m, n$ in Figure 1(f) -- it seems like HierTS surprisingly performs very poorly, in fact, with higher $T$, vanilla TS might have better regret than HierTS. Any insights as to why this is the case? This is also related to my first question.

Other small comments: In line 153, since $t$ is mentioned as a suffix in $P_{s,t}$, it might be better to use $\mu_t$, i.e. $\mathbb{P}(\theta_{s}^*=\theta|\mu_*=\mu_t, H_{s,t})$. Also, $\hat{\mu}_{s,t}$ in line 160 is not defined till the following page (equation 3).

**Limitations:**

See the questions.

---

> ### Author Rebuttal · Authors · 2024-08-05
>
> # Response to the Review by Reviewer diEN (Part 1/4)
>
> **Q1. The Gaussian distribution assumption for each layer of the hierarchy is highly restrictive since in many cases, the noise is not Gaussian (assuming sub-Gaussian would be better). This should at least be mentioned in the abstract since the current abstract gives the sense of a more general theory being developed in the paper.**\
> A1: Thank you for the suggestion, we will mention the Gaussian linear bandit setting in the abstract and introduction in the revised version to make the statement more rigorous. For the Gaussian assumption, our explanations are two-fold:\
> $\mathbf{(1)}$ It is true that the Gaussian distribution assumption for each layer of the hierarchy (especially the Gaussian noise) is restrictive. On the other hand, we also need to point out that the existing Bayes regret bounds [7,17,25] for hierarchical Bayesian bandit algorithms all adopt the Gaussian distribution for each layer of the hierarchy.\
> $\mathbf{(2)}$ Extending our results to the more general settings (e.g. sub-Gaussian noise) is also one of our future directions. However, the generalization is not easy, mainly because the non-Gaussian assumption could not lead to the closed-form posteriors and hence could not apply our proposed algebraic analysis (e.g. Lemma B.1, Propositions B.1, D.1 and E.1). Instead, we may need to choose other tools like information-theoretic analysis in [32] to derive Bayes regret bounds for hierarchical Bayesian bandit algorithms.
>
> **Q2. What is the benefit of assuming a hierarchical structure rather than dealing with each task independently? The overall regret is linear in $m$, which seems to be similar to that of dealing with the tasks independently.**\
> A2: Thanks. Our explanations are two-fold:\
> $\mathbf{(1)}$ It is true that, from the theoretical perspective, the current Bayes regret bound for multi-task hierarchical Bayesian bandit is similar to that of dealing with the tasks independently. This is not very surprising because the regret bound $O(\sqrt{n\log{n}})$ in [21] for single-task Bayesian bandit is also near-optimal. Besides, deriving more insightful regret bounds (e.g. the bound revealing the impact of task-similarity to the generalization) to illustrate the benefits of multi-task Bayes regret bound over the single-task regret bound for hierarchical Bayesian bandit is also one of our ongoing research directions.\
> $\mathbf{(2 )}$ From the algorithm perspective, in the hierarchical Bayesian bandit problem, using specifically-designed hierarchical Bayesian bandit algorithms can access more information of $\mu _{\star}$ via interacting with different tasks, than using traditional single-task Bayesian bandit algorithms. Such benefit can also be found in the lower regrets of HierTS/HierBayesUCB than that of vanilla TS in our Figures 1(f) and 2(f).
>
> **Q3. Is it possible to use this result to derive frequentist regret bounds (either high probability bounds or in expectation?**\
> A3: Probably not. Detailed explanations are two-fold:\
> $\mathbf{(1)}$ Notice that frequentist regret bound (for any fixed task instance) is the upper bound on the Bayesian regret. Besides, as shown in [31, Sect 3.1], only in some cases can Bayes regret bound be converted to frequentist regret bounds. Therefore it is not easy to use our Bayes regret bound to derive frequentist regret bounds. \
> $\mathbf{(2)}$ Obtaining frequentist regret bound may require more advanced techniques. To the best of our knowledge, deriving frequentist regret bounds always need to use advanced analysis tools such as anti-concentration inequality or martingale method (see [1,15]), while deriving Bayes regret bounds in our work just uses less technical algebraic bounding analysis.
>
> **Q4. In the Gaussian case, is the lower bound $\Omega(d\sqrt{n})$ tight in this hierarchical setting (note that this lower bound is for a linear bandit learning a single task, but it is not immediately obvious whether or not the shared structure across the tasks could help in the regret bound).**\
> A4:Thanks for pointing this out, we believe that the lower bound $\Omega(d\sqrt{n})$ is tight in the hierarchical Gaussian setting, due to the following reason. For any fixed hyper-parameter $\mu _{\star}$ (drawn from the hyper-posterior $Q$), [30, Theorem 2.1] shows that the lower Bayes regret bound for Gaussian bandit instance $\theta _{s,\star} \sim \mathbb{P}(\cdot | \mu _{\star})$ (for any policy) is $$ \mathbb{E} _{\theta _{s,\star} \sim  \mathbb{P}(\cdot | \mu _{\star})} \sum _{t=1}^{n}(A _{s,\star}^{\top}\theta _{s,\star} – A _{s,t}^{\top} \theta _{s,\star}) \geq 0.006d\sqrt{n}.$$ Taking expectation over $\mu _{\star} \sim Q$, we can obtain the lower Bayes regret bound for hierarchical Gaussian linear bandit: $$ \mathbb{E} _{\mu _{\star} \sim Q} \mathbb{E} _{\theta _{s,\star} \sim  \mathbb{P}(\cdot | \mu _{\star})} \sum _{t=1}^{n}(A _{s,\star}^{\top}\theta _{s,\star} – A _{s,t}^{\top} \theta _{s,\star}) \geq 0.006d\sqrt{n}.$$
>
> **Q5. While the distributions for $Q$ and $\theta _{s,\star}|\mu _{*}$ are fine to assume Gaussian, the issue is with the Gaussian noise, which is restrictive to apply the method to broader range of problems.**\
> A5: Thanks for pointing this out. It is true that the Gaussian noise makes our theoretical results restrictive to be applied to a broader range of problems. Extending our results to the more general setting (e.g. sub-Gaussian noise setting) is also one of future directions, and may require more advanced analysis tools rather than the algebraic bounding technique in the current work.

---

> ### Author Response · Authors · 2024-08-05
> **Response to the Review by Reviewer diEN (Part 2/4)**
>
> **Q6. The paper does not deal with the case where $\mu _{q}, \Sigma _{q}, \Sigma _{0}, \sigma$ are unknown.**\
> A6: Thanks for your comments. Our explanations are two-fold:\
> $\mathbf{(1)}$ It is true that the current version is unable to deal with the case where $\mu _{q}, \Sigma _{q}, \Sigma _{0}, \sigma$ are unknown, and this may be one limitation of the algebraic bounding technique (see more explanations in Remark F.1) to derive Bayes regret bounds (in [7,17,25] and ours) for hierarchical Bayesian bandit algorithms. \
> $\mathbf{(2)}$ To deal with the case where $\mu _{q}, \Sigma _{q}, \Sigma _{0}, \sigma$ are unknown, we believe we need to leverage other tools (like information-theoretic analysis) to analyze general cases.
>
> **Q7. The term $\mathcal{V} _{m,n}$ is the posterior variance of which distribution?.**\
> A7: Sorry for the confusion, our explanations are two-fold:\
> $\mathbf{(1)}$ The posterior variance $\mathcal{V} _{m,n} \leq \mathbb{E}\Big[\sum _{t\geq 1}\sum _{s \in \mathcal{S} _{t}}\lVert A _{s,t}\rVert _{\hat{\Sigma} _{s,t}}^{2}\Big]$ is just a notation, not the variance of certain random variable. And the expectation is taken over the randomness of the random variables $A _{s,t}$, $\hat{\Sigma} _{s,t}$.\
> $\mathbf{(2)}$ $\mathcal{V} _{m,n}$ measures the variation change of the Mahalanobis-norm of the actions $A _{s,t}$. Therefore, we call it posterior variance as in previous works [7,17].
>
> **Q8. For Theorem 5.2, it seems like the fact that $\lvert\mathcal{A}\rvert < \infty$ is the key to reducing the $\sqrt{n}$ factor in the averaged Bayes regret (compared to Theorem 5.1).**\
> A8: Thanks. Our explanations lie in the following two aspects:\
> $\mathbf{(1)}$ Actually the fact hat $\lvert\mathcal{A}\rvert < \infty$ is $\mathbf{not}$ the key to reducing the $\sqrt{n}$ factor, because $\lvert\mathcal{A}\rvert$ is the multiplicative factor in the regret bound of Theorem 5.2 and $\lvert\mathcal{A}\rvert < \infty$ is used to ensure that the regret bound will not be infinity.\
> $\mathbf{(2)}$ The key to reducing the $\sqrt{n}$ factor in Theorem 5.1 is actually the following strategy to decompose the Bayes regret into three terms: $$ \mathcal{BR}(m,n)=\mathbb{E}\sum_{t\geq1}\sum_{s \in \mathcal{S} _{t}}\Delta _{s,t}=\mathbb{E}\sum _{t\geq 1,s \in \mathcal{S} _{t}}\Delta _{s,t}\big[\mathbf{1}\lbrace\Delta _{s,t}\geq \epsilon, E _{s,t}\rbrace + \mathbf{1}\lbrace\Delta _{s,t}< \epsilon,E _{s,t}\rbrace +\mathbf{1}\lbrace \bar{E} _{s,t}\rbrace\big],$$ and we use the equal transformation $\Delta _{s,t}\mathbf{1}\lbrace\Delta _{s,t}\geq \epsilon, E _{s,t}\rbrace=\frac{\Delta _{s,t}^{2}}{\Delta _{s,t}} \mathbf{1}\lbrace\Delta _{s,t}\geq \epsilon, E _{s,t}\rbrace$ to bound the first term as $$ \mathbb{E}\sum _{t\geq 1,s \in \mathcal{S} _{t}}\Delta _{s,t}\big[\mathbf{1}\lbrace\Delta _{s,t}\geq \epsilon, E _{s,t}\rbrace\big] \leq  \mathbb{E}\sum _{t\geq 1,s \in \mathcal{S} _{t}}\frac{\Delta _{s,t}^{2}}{\Delta _{\min}^{\epsilon}} \leq \mathbb{E}\sum _{t\geq 1,s \in \mathcal{S} _{t}}\frac{(8\log{\frac{1}{\delta}})\lVert A _{s,t}\rVert _{\hat{\Sigma} _{s,t}}^{2}}{\Delta _{\min}^{\epsilon}} = O(\frac{\log{\frac{1}{\delta}}\mathcal{V} _{m,n}}{\Delta _{\min}^{\epsilon}}),$$ leading to the regret bound of order $O(\frac{m\log{n} \log{\frac{1}{\delta}}}{\Delta _{\min}^{\epsilon}})$ in Theorem 5.2 which removes the $\sqrt{n}$ factor in Theorem 5.1.
>
> **Q9. The additional term of $\log{m}$ in Theorem 5.2 is intriguing, can the authors comment on this?**\
> A9: Thanks for pointing this out. Our comments are two-fold:\
> $\mathbf{(1)}$ The regret bound in our Theorem 5.2 is (informal form) $$O(mn[\epsilon+\delta]+\frac{\log{\frac{1}{\delta}}}{\Delta _{\min}^{\epsilon}}m\log{n}),$$ and becomes $O(\log{(nm)}m\log{n})$ if we set $\delta = \frac{1}{nm}$, $\epsilon = \frac{1}{mn}$ and $\Delta _{\min} > \epsilon$ is large. Therefore, the additional term of $\log{m}$ is actually caused by setting the confidence factor $\delta = \frac{1}{mn}$, which to some extent is an unavoidable term when combining $m$ regret bounds for UCB-type algorithms.\
> $\mathbf{(2)}$ We can also remove the additional term of $\log{m}$ by setting $\delta = \frac{1}{n}$, then our Bayes regret bound in Theorem 5.2 becomes $O([mn\epsilon + m]+ \frac{\log{n}}{\Delta _{\min}^{\epsilon}}m\log{n})$, which is order of $O(m\log^{2}{n})$ if we set $\epsilon =\frac{1}{mn}$ and the gap $\Delta _{\min} > >\epsilon$ is large.

---

> ### Author Response · Authors · 2024-08-05
> **Response to the Review by Reviewer diEN (Part 3/4)**
>
> **Q10. Can the authors comment on the technical difference to remove the polynomial dependence of $K$ to logarithmic in Theorem 5.4 for the combinatorial case?.**\
> A10: Thanks. We need to admit that the improvement of the polynomial dependence of $K$ to logarithmic in Theorem 5.4 is actually attributed to the $d$-dimensional feature representation of each action. Detailed explanations lie in the two following aspects:\
> $\mathbf{(1)}$ If we use our Theorem 5.4 to derive regret bounds for multi-task $K$-armed bandit, then $d=K$ and we need to set the feature representation of the $k$-th arm ($k \in [K]$) as the one-hot vector whose $k$-th element is $1$ and others $0$. Then we need to replace $d$ in our Theorem 5.4 with $K$, and will still get a regret bound for multi-task $K$-armed bandit with polynomial dependence of $K$.\
> $\mathbf{(2)}$ Even though our Theorem 5.4 will lead to a regret bound with polynomial dependence of $K$ for traditional $K$-armed bandit problem (without feature representation), our Theorem 5.4 still reveals an insight that: if the size $K$ of arms is truly large (e.g. $K >1000$), representing each action/arm with a $d$-dimensional feature ($d < K$) will result in a sharper regret bound without the polynomial dependence of $K$.
>
> **Q11. The bound in Theorem 5.5 is stronger than that in [S. Basu, NeurIPS2021] in the case $\Delta _{\min}^{\epsilon}$ is large, otherwise the comment in line 266 would result in a bound worse, is this correct?**\
> A11: Thanks. It is true that if the gap $\Delta _{\min}^{\epsilon}$ is small and $\epsilon=\frac{1}{mn}$, our bounds in Theorems 5.2 and 5.5 become worse. However, if we choose $\epsilon = \frac{1}{\sqrt{n}}$ (or choose $\epsilon = \frac{1}{n}$ as explained in Answer 9 to **Question 9**), we can still obtain a good regret bound even if the gap $\Delta _{\min}^{\epsilon}$ is small. Detailed explanations are two-fold and will be added in the final version:\
> $\mathbf{(1)}$ If we set $\epsilon = \frac{1}{\sqrt{n}}$, $\delta = \frac{1}{n}$, and $\Delta _{\min} >> \epsilon$ (i.e. $\Delta _{\min}$ is large), our regret bound in Theorem 5.2 (similar regret bound in Theorem 5.5 can be derived in a similar way) is of order $O(m\sqrt{n} + m\log^{2}{n}) = O(m\sqrt{n})$ and is still improved over the latest bound $O(m\sqrt{n\log{n}\log{(mn)}})$ in [17, Theorem 3] in Table 1.\
> $\mathbf{(2)}$ If we set $\epsilon = \frac{1}{\sqrt{n}}$, $\delta = \frac{1}{n}$, and $\Delta _{\min} \leq \epsilon$ (i.e. $\Delta _{\min}$ is small), our regret bound in Theorem 5.2 is of order $O(m\sqrt{n}+\sqrt{n}\log{n} [m\log{n}])=O(m\sqrt{n}\log^{2}{n})$, which is still comparable with the latest one $O(m\sqrt{n\log{n}\log{(mn)}})$ in [17, Theorem 3].
>
> **Q12. Algorithms 1 and 2: Apart from $Q$, the update step for $Q _{t+1}$ also requires the various other likelihood and model parameters (assumed known) like $\mu _{q}, \Sigma _{q}, \sigma$, which should be included as input.**\
> A12: Sorry for the confusion. Actually Algorithms 1 and 2 are general forms of HierTS/HierBayesUCB algorithms (hence for any hierarchical distribution assumption), not the specific forms of HierTS/HierBayesUCB algorithms under hierarchical Gaussian assumption. Therefore, the input of general HierTS/HierBayesUCB algorithms does not necessarily include model parameters like $\mu _{q}, \Sigma _{q}, \sigma$. We will correct the step **Update $Q _{t+1}$ with Eq.(2)** in Algorithms 1 and 2 with **Update $Q _{t+1}$** to avoid confusion in the revised version.

---

> ### Author Response · Authors · 2024-08-05
> **Response to the Review by Reviewer diEN (Part 4/4)**
>
> **Q13. No simulation examples were shown to demonstrate the performance of *ORACLE*. operator and more importantly, the computational complexity, for this important step in both HierTS and HierBayesUCB algorithms.**\
> A13: Thanks for pointing this out. It is true that in the current work we did not conduct simulation experiments to demonstrate the performance of *ORACLE* operator and combinatorial HierTS/HiereBayesUCB. This paper mainly focuses on providing improved regret analysis and novel algorithms for hierarchical Bayesian bandit/semi-bandit settings. Practical implementation of the proposed hierarchical semi-bandit algorithms is left as one of our research directions.
>
> **Q14. What are the values of $m,n$ in Figure 1(f) -- it seems like HierTS surprisingly performs very poorly, in fact, with higher $T$, vanilla TS might have better regret than HierTS.**\
> A14: Thanks. As shown in Experiment section, we set $m=10, n=400$ in Figure 1(f). There are two insights behind the performance of HierTS in Figure 1(f): \
> $\mathbf{(1)}$ It is true that in Figure 1(f) (where we set $\sigma _{q}=1$) HierTS outperforms vanilla TS by a small margin. But in Figure 2(f) (where we set a smaller $\sigma _{q}=0.5$) HierTS outperforms vanilla TS much better. Therefore, the outperformance of HierTS over vanilla TS could vary in different hierarchical Bayesian bandit environments. \
> $\mathbf{(2)}$ The different outperformance of HierTS over vanilla TS is consistent with our regret bound in Theorem 5.1 for HierTS: our regret bound involves terms $\lambda _{1}(\Sigma _{q})$ and $tr(\Sigma _{q} \Sigma _{0}^{-1})$, which reveals that a larger variance $\sigma _{q}$ leads to a larger regret bound. Hence, the larger variance $\sigma _{q} =1.0$ in Figure 1(f) leads to larger regret of HierTS than that in Figure 2(f) (with a smaller variance $\sigma _{q} =0.5$).
>
> **Q15. In line 153, since $t$ is mentioned as a suffix in $\mathbb{P} _{s,t}$, it might be better to use $\mathbb{P}(\theta _{s,\star}=\theta | \mu _{*}=\mu _{t}, H _{s,t})$. Also, $\mu _{s,t}$ in line 160 is not defined till the following page (equation 3).**\
> A15: Thank you for the kind suggestion, we will use $\mathbb{P}(\theta _{s,\star}=\theta | \mu _{\star}=\mu _{t}, H _{s,t})$ in the final version. Besides, $\hat{\mu}$ and $\hat{\sigma}$ in line 160 are the expectation and covariance of the conditional distribution (given the history $H$) of the true task parameter $\theta _{*}$, and we will clarify them when introducing HierBayesUCB algorithm in the revised version.
>
> **References**\
> [1] Improved Algorithms for Linear Stochastic Bandits. NeurIPS 2011.\
> [7] No Regrets for Learning the Prior in Bandits. NeurIPS 2021.\
> [15] Improved Algorithms for Stochastic Linear Bandits Using Tail Bounds for Martingale Mixtures. NeurIPS 2023.\
> [17] Hierarchical Bayesian Bandits. AISTATS 2022.\
> [21] An Improved Regret Bound for Thompson Sampling in the Gaussian Linear Bandit Setting. ISIT 2021.\
> [25] Meta-Thompson Sampling. ICML 2021.\
> [30] Linearly Parameterized Bandits. MoOR 2010.\
> [31] Learning to Optimize via Posterior Sampling. MoOR 2011.\
> [32] An Information-Theoretic Analysis of Thompson Sampling. JMLR 2016.

---

> ### Comment · Reviewer_diEN · 2024-08-13
>
> Thanks for the response. I am overall satisfied with the responses and will raise my score to 6. However, based on the discussion, I feel there are a lot of interesting directions to pursue on various fronts within the current setup.

---

> > ### Author Response · Authors · 2024-08-13
> > **Thank you for the response.**
> >
> > Dear Reviewer diEN,\
> >        Many thanks for your response. We appreciate your support very much, and really benefit a lot from your detailed reviews. We will take your suggestions into the revision to improve the quality of this paper. Thank you!
> >
> > Best,\
> > Authors

---

### Official Review · Reviewer_yLZF · 2024-07-16

**Soundness:** 3
**Presentation:** 4
**Contribution:** 3
**Rating:** 8
**Confidence:** 4

**Summary:**

This paper studies the multi-task Gaussian linear bandit and semi-bandit problems. It uses a Bayesian approach to maintain a meta-distribution over the hyper-parameters of within-task parameters. It provides an improved regret bound for the for multi-task for HierTS algorithm in the case of infinite action set. For the same setting, it proposes HierBayesUCB which is the UCB counterpart of HierTS. The authors extend their algorithm to the concurrent multi-task and combinatorial semi-bandit settings as well.

**Strengths:**

Contribution: The paper improves upon prior works in several directions; it proposes tighter regret bounds for HierTS, extends multi-task bandit algorithms to combinatorial semi-bandit setting.

Presentation: the paper is very well-organized, it clearly defines the problem, pseudo-code of the algorithms are clearly presented, and the experimental results are extensive.

**Weaknesses:**

The paper is lacking real-world dataset experiments. This could help assess the algorithm's robustness w.r.t model misspecification and gauge the theoretical results consistency w.r.t assumptions violation.

**Questions:**

No questions.

**Limitations:**

Yes.

---

> ### Author Rebuttal · Authors · 2024-08-05
>
> **Q1. The paper improves upon prior works in several directions; it proposes tighter regret bounds for HierTS, extends multi-task bandit algorithms to combinatorial semi-bandit setting.**\
> A1: Thanks for your positive comments. we will continue to improve the quality of this paper.
>
> **Q2. The paper is lacking real-world dataset experiments. This could help assess the algorithm's robustness w.r.t model misspecification and gauge the theoretical results consistency w.r.t assumptions violation.**\
> A2: Thanks for your comments. It is true that in the current work we only conduct synthetic experiments to validate the effectiveness of our theoretical results and our proposed algorithms. Conducting real-world dataset experiments is also one of our future directions and we will try to add it in the final version.

---

### Official Review · Reviewer_wPds · 2024-07-16

**Soundness:** 4
**Presentation:** 4
**Contribution:** 3
**Rating:** 6
**Confidence:** 3

**Summary:**

This paper revisits the learning problems of (multi-task) Bayesian linear bandit/semi bandits, which is interesting. The improvement of the Bayesian regret bound for the multi-task Bayes regret bound of HierTS is very marginal. For the remaining presented upper bounds, there are still near-optimal up to some $\log$ factors from the regret lower bounds. I do not check the proofs in the appendix in detail, but it seems they are correct and the proofs are well written.

In the introduction section, the description
 “The gap between the cumulative reward of optimal actions in hindsight and the cumulative reward of agent 24 is defined as regret” seems not to be aligned with the learning problems to be solved. For me, I think we still use pseudo rewards. So, we may not care about the optimal actions in hindsight. Instead, we care about the action with the highest mean given a problem instance.

Questions:

I have a question, when using the notion of Bayesian regret, why is there an instance-dependent gap? For the UCB-based algorithms, I am feeling that it is also possible to have an instance-dependent regret bound using the notion of frequentist regret. Why not do it in that way?



In Line 53, there is a typo. I guess it should be “latest” instead of “latext”.

Can you rotate Table 4? It is hard for readers who use a desktop.

**Strengths:**

see the first box

**Weaknesses:**

see the first box

**Questions:**

see the first box

**Limitations:**

Frequentist analysis may be more interesting, but I kind of know the reasons why not to use frequentist regret as the combination of TS and frequentist regret needs to have anti-concentration bounds, which are always hard to develop.

---

> ### Author Rebuttal · Authors · 2024-08-05
>
> **Q1. The description “The gap between the cumulative reward of optimal actions in hindsight and the cumulative reward of agent 24 is defined as regret” seems not to be aligned with the learning problems to be solved. Instead, we care about the action with the highest mean given a problem instance.**\
> A1: Thanks for pointing this out. It is true that in Problem Setting we care about the action with the highest mean reward (not the highest reward and hence seems not aligned with the pseudo regret definition in Introduction), and we define the action with the highest mean reward as the optimal action. We will make a more rigorous statement of our introduction in the revised version.
>
> **Q2. When using the notion of Bayesian regret, why is there an instance-dependent gap?**\
> A2: Thanks. There are two reasons for the existence of instance-dependent gap:\
> $\mathbf{(1)}$ The first reason is that we use a new bounding technique to upper bound $\mathbb{E}\Delta _{s,t}$ (e.g. see Lines 549-550) as follows (informal form): $$ \mathbb{E}\Delta _{s,t} \mathbf{1}\lbrace\Delta _{s,t} \geq \epsilon\rbrace=\mathbb{E}\frac{\Delta _{s,t}^{2}}{\Delta _{s,t}} \mathbf{1}\lbrace\Delta _{s,t} \geq \epsilon\rbrace \leq \mathbb{E}\frac{C _{t,s, A _{s,t}}^{2}}{\Delta _{\min}^{\epsilon}}.$$ Therefore, there exists $\Delta _{\min}^{\epsilon}$ in the denominator in our bound.\
> $\mathbf{(2)}$ The second reason is that the instance-dependent gap is actually a random variable, and our bound needs to take the expectation of this random variable. Note that the instance $\theta _{s, \star}$ is random, and the gap $\Delta _{s,t} = \theta _{s,\star}^{\top}A _{s, \star}-  \theta _{s,\star}^{\top}A _{s, t}$ is a random variable. Finally, the term $\mathbb{E}{\frac{1}{\Delta _{\min}^{\epsilon}}}$ in our regret bound (e.g. see Lines 549-550) takes the expectation over the randomness of $\Delta _{s,t}$. Similar instance-dependent gap based regret bound can also be found in the latest work [3, Thm5].
>
> **Q3. For the UCB-based algorithms, I am feeling that it is also possible to have an instance-dependent regret bound using the notion of frequentist regret. Why not do it in that way?**\
> A3: Thanks for your suggestions, our explanations are two-fold:\
> $\mathbf{(1)}$ Using frequentist regret bound to derive Bayes regret bound (either instance-dependent or instance-independent) is also one of our future research directions, because the techniques for deriving frequentist regret bounds are more fruitful and more general. But in the current work, we focus on applying improved Bayes regret analysis (like improved matrix analysis in Lemma B.1 and novel regret decomposition strategy in Theorem C.1) to derive sharper Bayes regret bounds.\
> $\mathbf{(2)}$ It seems difficult to apply existing gap-dependent frequentist regret bound for UCB-type algorithm to derive instance-gap-dependent Bayes regret bound for multi-task hierarchical Bayesian bandit problems. To the best of our knowledge, the existing instance-dependent frequentist regret bound for UCB-type algorithm is in [1, Theorem 5], and is not suitable to be applied in our setting due to the following two reasons: $\mathbf{(i)}$ [1, Theorem 5] assumes the boundedness of the instance parameter $\theta _{s,\star}$, which hence could not be applied to analyze Gaussian bandit instance where the instance parameter $\theta _{s,\star}$ is not uniformly bounded. $\mathbf{(ii)}$ The frequentist regret bound in [1, Theorem 5] is more lengthy than the single-task Bayes regret bound for UCB in [3, Theorem 5], and hence could not lead to a concise multi-task Bayed bound as in our Theorem 5.2.
>
> **Q4. In Line 53, there is a typo. I guess it should be “latest” instead of “latext”.**\
> A4: Thanks for pointing this out. We will correct it in the final version.
>
> **Q5. Can you rotate Table 4? It is hard for readers who use a desktop.**\
> A5: Sorry for the inconvenience. Because Table 4 is much wider than it is tall, we choose Table 4 with horizontal layout.
>
> **References**\
> [1] Improved Algorithms for Linear Stochastic Bandits. NeurIPS 2011.\
> [3] Finite-Time Logarithmic Bayes Regret Upper Bounds. NeurIPS 2023.

---

### Author Rebuttal · Authors · 2024-08-05

# Response to All Reviewers

We sincerely thank all reviewers for their detailed reading and constructive comments. We address below one concern that is common among reviewers and then we address the concerns of each reviewer individually:

**Common Question 1. Discuss the novelty in the techniques beyond using the improved inequality from [21]**\
A1: Besides the application of the improved inequality from [21], our technical novelties lie in the following three aspects and we will clarify them in the revised version: \
$\mathbf{(1)}$ For the improved regret bound for HierTS in Theorem 5.1 in the sequential bandit setting: our proof has two novelties: $\mathbf{(i)}$ We use a more technical positive semi-definite matrix decomposition analysis (i.e. our Lemma B.1) to reduce the multiplicative factor $\kappa^{2}(\Sigma _{0})$ to $\kappa(\Sigma _{0})$. $\mathbf{(ii)}$ Define a new matrix $\tilde{X} _{s,t}$ such that the denominator in the regret is $\sigma^{2}+B^{2}\lambda _{1}(\Sigma _{0})$, not just $\sigma^{2}$, avoiding the case that the variance serves alone as the denominator. Such technical novelties are also listed in our Table 4.\
$\mathbf{(2)}$ For the improved regret bound for HierBayesUCB in Theorem 5.2 in the sequential bandit setting: our technical novelty lies in decomposing the Bayes regret $\mathcal{BR}(m,n)=\mathbb{E}\sum _{t\geq1}\sum _{s \in \mathcal{S} _{t}}\Delta _{s,t}$ into three terms:
$$\mathbb{E}\sum _{t\geq1}\sum _{s \in \mathcal{S} _{t}}\Delta _{s,t}=\mathbb{E}\sum _{t \geq 1,s \in \mathcal{S} _{t}}\Delta _{s,t}\big[\mathbf{1}\lbrace\Delta _{s,t}\geq \epsilon, E _{s,t}\rbrace + \mathbf{1}\lbrace\Delta _{s,t}< \epsilon,E _{s,t}\rbrace +\mathbf{1}\lbrace \bar{E} _{s,t}\rbrace\big],$$ and bounding the first term with a new method as well as the specific property of BayesUCB algorithm as follow: $$ \mathbb{E}\Delta _{s,t} \mathbf{1}\lbrace\Delta _{s,t} \geq \epsilon, E _{s,t}\rbrace=\mathbb{E}\frac{\Delta _{s,t}^{2}}{\Delta _{s,t}} \mathbf{1}\lbrace\Delta _{s,t} \geq \epsilon, E _{s,t}\rbrace \leq \mathbb{E}\frac{C _{t,s, A _{s,t}}^{2}}{\Delta _{\min}^{\epsilon}},$$ resulting in the final improved gap-dependent regret bound for HierBayesUCB as follows $$ \big(\sum _{t\geq1,s\in \mathcal{S} _{t}}\lVert A _{s,t}\rVert _{\hat{\Sigma} _{s,t}}^{2}\log{\frac{1}{\delta}}\big)/\Delta _{\min}^{\epsilon} \leq O\big(m\log{(n)}\log{\frac{1}{\delta}} \big),$$ which is of order $ O\big(m\log{(n)}\log{(mn)} \big)$ if we set $\delta = \frac{1}{mn}$.\
$\mathbf{(3)}$ For the improved regret bounds for HierTS and HierBayesUCB in the concurrent setting and in the sequential semi-bandit setting: besides the aforementioned technical novelties in $\mathbf{(1)}$ and $\mathbf{(2)}$, the additional technical novelty lies in leveraging more refined analysis (e.g. using Woodbury matrix identity cleverly) to bound the gap between the matrices $\bar{\Sigma} _{t+1}^{-1}$ and $\bar{\Sigma} _{t}^{-1}$ (more details can be found in Lemma D.1 and Equation (6) in Page 22).

Besides the concerns, we also thank gratefully reviewers for pointing out minor typos and we will fix them in the revised version.

---

### Decision · Program_Chairs · 2024-09-25

**Decision:**

Accept (poster)

**Comment:**

This paper examines the Hierarchical Bayesian multi-task bandit problem, improving the Bayes regret bound of an existing algorithm (HierTS), and proposes a new algorithm (HierBayesUCB) with an improved Bayes regret bound under additional assumptions.

Throughout the paper, the use of the term "gap-dependent/independent" analysis for Bayes regret is somewhat misleading. The information about the gap (in a frequentist sense) disappears once expectations are taken, as pointed out by Reviewer wPds. The Gap-dependent in this paper corresponds to the structure for the prior. If possible, it is recommended to rewrite Theorem 5.5, etc., using the margin conditions (see e.g., Chakraborty et al., ICML, 2023).

Nevertheless, a number of contributions are still noteworthy. Following the rebuttal and discussion, all reviewers have a favorable view of the paper's contributions and lean toward acceptance.